# A 7T fMRI dataset of synthetic images for out-of-distribution modeling of vision

Alessandro T. Gifford [1,2,3] ✉, Radoslaw M. Cichy [1,2,3,4], Thomas Naselaris[5,6] & Kendrick Kay [5] ✉

Large-scale visual neural datasets such as the Natural Scenes Dataset (NSD) are enabling models of the brain with performances beyond what was possible just a decade ago. However, because the stimuli of these datasets typically live within a common naturalistic visual distribution, they make it challenging to implement out-of-distribution (OOD) generalization tests crucial for the development of robust brain models. Here, we address this by releasing NSD-synthetic, a dataset of 7T fMRI responses from the same eight NSD participants for 284 synthetic images. We show that NSD-synthetic's fMRI responses reliably encode stimulus-related information and are OOD with respect to NSD; that OOD generalization tests on NSD-synthetic reveal differences between brain models that are not detected in-distribution; and that the degree of OOD (quantified as the test data distance from the training data) is predictive of the magnitude of model failures. Together, NSD-synthetic enables OOD generalization tests that facilitate the development of more robust models of visual processing.

Recently, there has been an increase in both collection and use of large-scale visual neural datasets that extensively sample brain responses to visual stimuli, suggesting that the field of vision science is entering a new era of big open data[1–7]. One example of this recent trend in the context of human fMRI is the Natural Scenes Dataset (NSD) which provides high-quality 7T fMRI responses to over 70,000 images of naturalistic scenes across 8 participants[1]. Its unprecedented size and quality have made NSD one of the most popular go-to datasets for research in computational neuroscience, where data-hungry machine- and deep-learning models are used to understand the functioning of the brain[8–13]. Within three years since its release, NSD has been used in hundreds of research projects, leading to the development of state-of-the-art predictive models of neural responses to visual stimulation[14–16] and to new theory formation[17–21]. Thus, large-scale visual neural datasets such as NSD are enabling new progress in computational models of the brain, leading to new theoretical insights.

However, it remains difficult to assess the level of out-of-distribution (OOD) generalization of brain models built using recent large-scale visual neural datasets, that is, whether model predictions generalize outside of the stimulus distribution on which they are trained[22]. This is because, despite the unprecedented size of these datasets, their stimuli typically live within the same visual distribution, which comprises only a fraction of the vast visual space that our brains process during our lifetime. For example, through clustering and reconstruction analyses, Shirakawa and colleagues showed that NSD's stimulus images have limited visuo-semantic diversity[23]. As a result, brain models are typically tested in-distribution (ID), that is, within the visual distribution on which they are trained. However, OOD tests are critical for visual neuroscience research in three significant ways. First, successful models should predict brain responses under a broad range of situations. In that sense, OOD generalization serves as a stricter but essential assessment of model robustness and validity than ID tests. Poor OOD generalization indicates that theoretical inferences of brain

[1]Institute of Psychology, Freie Universität Berlin, Berlin, Germany. [2]Einstein Center for Neurosciences Berlin, Charité—Universitätsmedizin Berlin, Berlin, Germany. [3]Bernstein Center for Computational Neuroscience, Humboldt-Universität zu Berlin, Berlin, Germany. [4]Berlin School of Mind and Brain, Humboldt-Universität zu Berlin, Berlin, Germany. [5]Center for Magnetic Resonance Research, Department of Radiology, University of Minnesota, Minneapolis, MN, USA. [6]Department of Neuroscience, University of Minnesota, Minneapolis, MN, USA. ✉e-mail: alessandro.gifford@gmail.com; kay@umn.edu

function might not apply beyond stimuli from the train distribution, suggesting that further development is needed to capture stimulus-to-brain-response relationships. Second, breaking down generalization scores across different types of OOD conditions can reveal stimulus properties that neural models fail to account for[24], hence providing explicit targets for model improvement. Third, OOD generalization scores serve to distinguish among models that may otherwise have similar ID test scores[25,26]. Finding that one of several similarly-performing models generalizes better OOD can reveal important properties that make models successful (e.g., architecture, training diet, learning objective), informing the engineering of more robust models and inspiring new hypotheses about brain function.

To address the lack of OOD components in large-scale visual neural datasets and enable crucial OOD generalization tests of brain models, here we release a companion dataset to NSD called *NSD-synthetic* (for the remainder of the paper, we refer to NSD as *NSD-core*, to distinguish it from NSD-synthetic). NSD-synthetic consists of fMRI responses for an additional scan session from the same eight participants of NSD-core. During this session, fMRI responses were measured to 284 synthetic (non-naturalistic) stimuli while the participant performed either a fixation task or a one-back task. Through computational analyses and modeling we show that NSD-synthetic's fMRI responses reliably encode stimulus-related information (Fig. 2) and that the responses are OOD with respect to NSD-core (Fig. 3). Furthermore, we provide proof of principle that encoding models trained on NSD-core generalize OOD on NSD-synthetic with lower prediction accuracies compared to ID generalization on NSD-core (Fig. 4); that OOD tests on NSD-synthetic reveal differences between encoding models not detected by ID tests on NSD-core (Fig. 5); that the degree of OOD is a useful indicator of the magnitude of model failures (Fig. 6); and that the concept of OOD can be usefully applied even within the domain of naturalistic stimuli (Fig. 7). Together, as the OOD companion of NSD-core, NSD-synthetic enables strict OOD generalization tests critical for development of more robust models of visual processing and the formulation of more accurate theories of human vision.

## Results

### NSD-synthetic stimuli and experimental design

To enable strict OOD generalization tests of computational models of visual processing, we created a set of 284 synthetic images that are far removed from the natural scene images in NSD-core (Fig. 1a). These synthetic images are divided into 8 classes, and every class contains multiple subclasses of 4 images each. The image classes include: various types of noise (4 subclasses, 16 images); natural scenes (2 subclasses, 8 images); manipulated version of natural scenes (3 subclasses, 12 images); contrast modulation (4 subclasses, 16 images); phase-coherence modulation (4 subclasses, 16 images); single words varying in position (10 subclasses, 40 images); spiral gratings varying in orientation and spatial frequency (28 subclasses, 112 images); and chromatic noise varying in hue (16 subclasses, 64 images) (Fig. 1b).

We presented these images in a rapid event-related design consisting of 4-s trials (2-s ON, 2-s OFF; instead of NSD-core's 3-s ON, 1-s OFF trials) while measuring fMRI responses (7T, 1.8-mm resolution, TR 1.6 s) from each of the eight NSD participants (Fig. 1c). To assess potential task dependence of neural responses, participants performed a fixation task and a one-back task in alternating runs (instead of NSD-core's long-term continuous recognition task) (Fig. 1d). In the fixation task, participants reported brightness increments and decrements of the fixation dot. In the one-back task, participants reported whether the current image was identical to the previous image. The participants exhibited good behavioral compliance: overall percent correct in the fixation task was 98%, 97%, 80%, 94%, 97%, 92%, 76%, and 83%, and overall $d'$ in the one-back task was 3.3, 2.6, 1.6, 2.7, 2.6, 3.2, 2.2, and 1.9. Since, for the purposes of this paper, our goal is to showcase

NSD-synthetic's out-of-distribution image properties, we pooled fMRI responses across tasks and analyzed fMRI responses with respect to the stimulus presented.

### NSD synthetic's fMRI responses reliably encode stimulus-related information

NSD-synthetic's main purpose is to facilitate out-of-distribution (OOD) generalization tests aimed at improving the robustness of computational models of the brain. A prerequisite of these tests is that NSD-synthetic's fMRI responses must reliably encode the different stimulus images. To assess this, we calculated NSD-synthetic's noise ceiling signal-to-noise ratio (NCSNR), a measure of stimulus-related signal in the fMRI responses, using methods introduced in previous work[1]. For all participants, we found that NCSNR scores are high over visual cortex, ranging approximately from 0.75 to 2 (Fig. 2a). This indicates that 36–80% variance in single-trial responses reflects signals driven by the stimulus. As a comparison, NSD-core's NCSNR scores range approximately from 0.5 to 1.25, indicating that 20–60% variance in single-trial responses reflects signals driven by the stimulus (Supplementary Fig. 1). Hence, stimulus-related information is indeed reliably encoded in NSD-synthetic's fMRI responses. Within visual cortex, NSD-synthetic's NCSNR is highest in early areas compared to intermediate, ventral, dorsal, and lateral areas. We believe this reflects the limited visual and semantic complexity of NSD-synthetic's stimuli, which leads to reduced signals in higher visual areas that preferentially respond to more complex visual features[27–32]. Supporting this intuition, the visually and semantically more complex naturalistic images from NSD-core lead to NCSNR scores that are more uniform across visual cortex (Supplementary Fig. 1).

To further assess the quality of the encoded visual information, we next analyzed NSD-synthetic's univariate and multivariate fMRI responses from both early visual cortex (EVC) and high-level visual cortex (HVC) regions of interest (ROIs). We chose ROIs based on their known responsiveness for the visual features defining NSD-synthetic's stimulus images, so as to best characterize the visual information encoded in NSD-synthetic's fMRI responses. The EVC ROIs consisted of V1, V2, V3, and hV4, due to their responsiveness to low- and mid-level visual features such as spatial frequency[33], contrast energy[34], edges[35], and color[36]. The HVC ROIs consisted of the parahippocampal place area (PPA) due to its responsiveness to visual scenes[29] and of the visual word form area (VWFA) due to its responsiveness to visual words[37].

We begin by showing that NSD-synthetic's univariate fMRI responses—defined as the average activity over all vertices within each ROI—for each image subclass reproduce known tunings of EVC and HVC ROIs. In Fig. 2b, we see that images of scenes activate PPA more than images consisting of noise, images with low contrast, and images with low phase-coherence, in line with PPA's tuning to visual environments[29]. Additionally, univariate responses increase substantially with increasing contrast for EVC ROIs[38]. Finally, images consisting of noise and images with low phase coherence activate EVC ROIs more than HVC ROIs, in line with EVC's selectivity for low-level visual features such as contrast energy[34]. In Fig. 2c, we find that chromatic noise activates EVC more than HVC, in line with EVC's role in the processing of color[36] and the fact that the chromatic noise patterns lack high-level structure. In Fig. 2d, we see that single words activate VWFA the most, especially when presented foveally (i.e., Word4 Pos3, Word6 Pos3), in line with VWFA's tuning to visual words[37]. In Fig. 2e, we see that spiral gratings activate EVC more than HVC, in line with EVC's general responsiveness to contrast energy[34]. Among all EVC ROIs, V1 is driven by spiral gratings the most, and shows preferential tuning for certain spatial frequencies within each spiral grating type (e.g., SpiralA SF4, SpiralB SF4, SpiralC SF4)[33,39,40].

Next, we analyze the visual information in NSD-synthetic's multivariate fMRI responses—i.e., population response patterns over all vertices within each ROI. Through representational similarity analysis

 

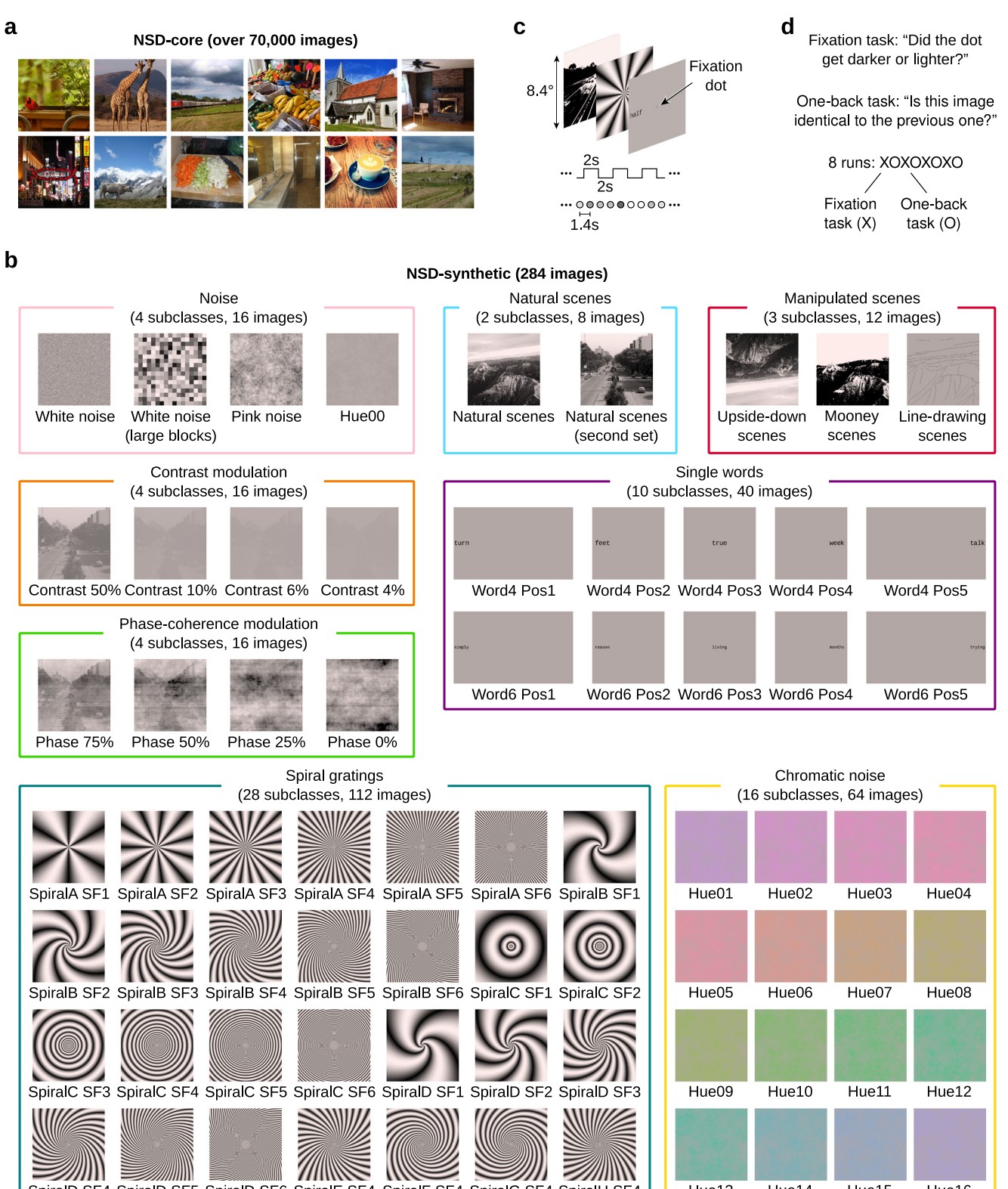

**Fig. 1 | Stimuli and experimental design. a** Example natural scenes from NSD-core. **b** Example stimulus images in NSD-synthetic. The images are divided into 8 classes, indicated by colored boxes. Every class is further divided into a varying number of subclasses, where each subclass consists of 4 images. For each subclass, we visualize one of the four images along with the subclass name. For visualization purposes, the images with the most peripheral words are not square cropped (Word4 Pos1, Word4 Pos5, Word6 Pos1, Word6 Pos5). **c** Trial design. Images were presented using a 2-s ON, 2-s OFF trial structure. A small central fixation dot changed luminance every 1.4 s. **d** Tasks. Participants performed two different tasks in alternating runs. In the fixation task, participants reported increments and decrements of the luminance of the fixation dot. In the one-back task, participants judged whether the currently presented image was identical to the previously presented image.

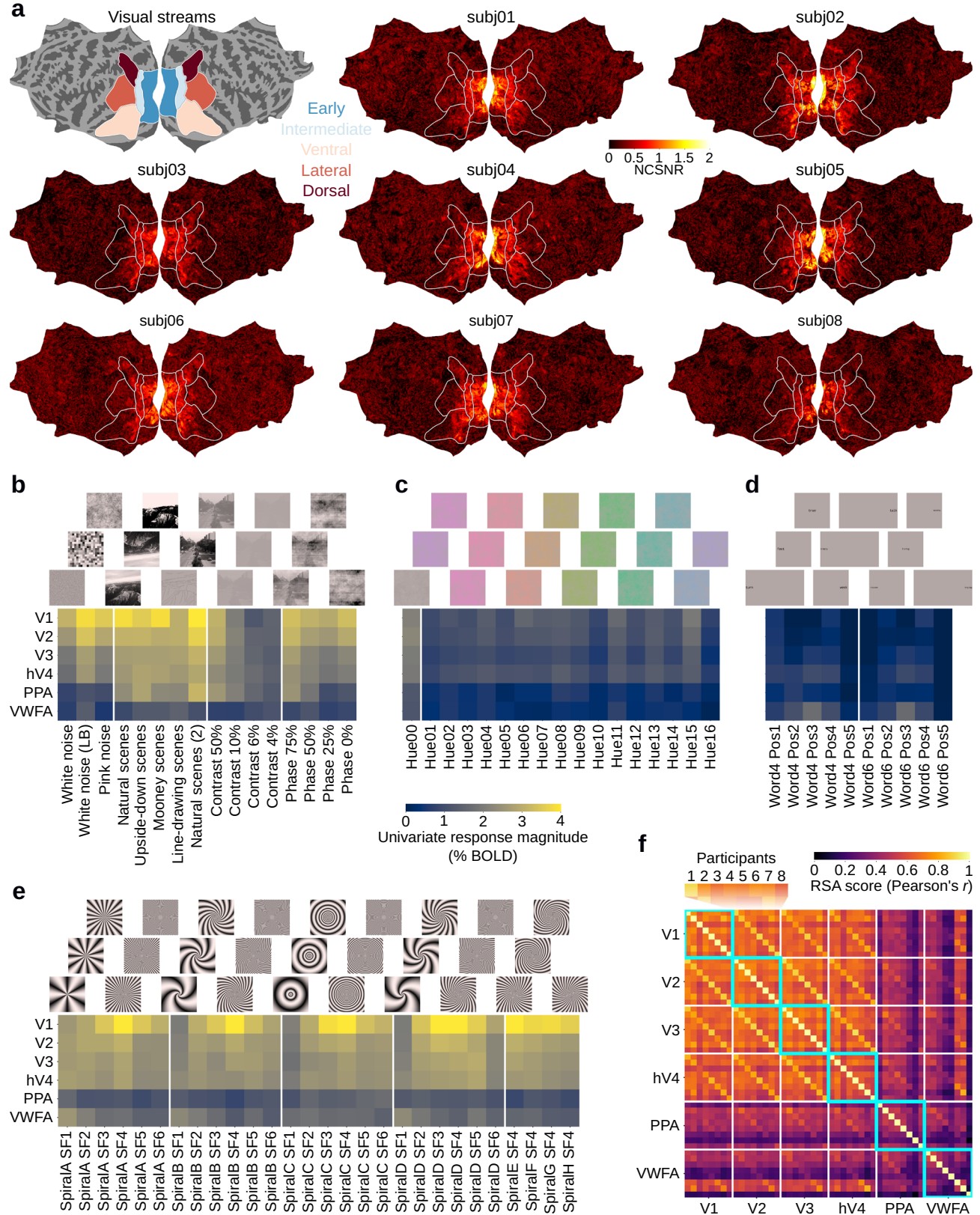

(RSA)[41], we assessed the similarity of multivariate fMRI responses between each participant and ROI. For each ROI, we found that NSD-synthetic's multivariate fMRI responses are correlated between participants (RSA scores within the cyan boxes in Fig. 2f), indicating that participants have shared visual representations. The cross-participant RSA scores are higher for EVC than for HVC, in line with the NCSNR and

univariate response results indicating EVC ROIs are more responsive to NSD-synthetic's stimulus images than HVC ROIs (Fig. 2a–e). We also found that the multivariate fMRI responses are correlated between ROIs (RSA scores outside the cyan boxes in Fig. 2f), again more so for EVC ROIs. In particular, RSA scores from ROIs in the same participant are elevated (k-th subdiagonals in Fig. 2f), which may be indicative of

**Fig. 2 | Noise ceiling, univariate, and multivariate analyses reveal robust visual signals in NSD-synthetic. a** Participant-wise noise ceiling signal-to-noise ratio (NCSNR), plotted on flattened cortical surfaces. White contours indicate regions based on the 'streams' region of interest (ROI) collection as provided in the NSD data release. **b** ROI-wise participant-average univariate responses for noise, natural/manipulated scenes, contrast modulation, and phase-coherence modulation (groupings indicated by vertical white lines). **c** ROI-wise responses for chromatic noise (Hue01–Hue16) and corresponding achromatic noise (Hue00). **d** ROI-wise

responses for single words. For visualization purposes, the images with the most peripheral words are not square cropped (Word4 Pos1, Word4 Pos5, Word6 Pos1, Word6 Pos5). **e** ROI-wise responses for spiral gratings. **f** Representational similarity analysis (RSA) scores indicating the similarity of multivariate fMRI responses between participants and ROIs. Thin white lines separate groups of the eight participants within the same ROI. Cyan boxes indicate cross-participant comparisons within the same ROI.

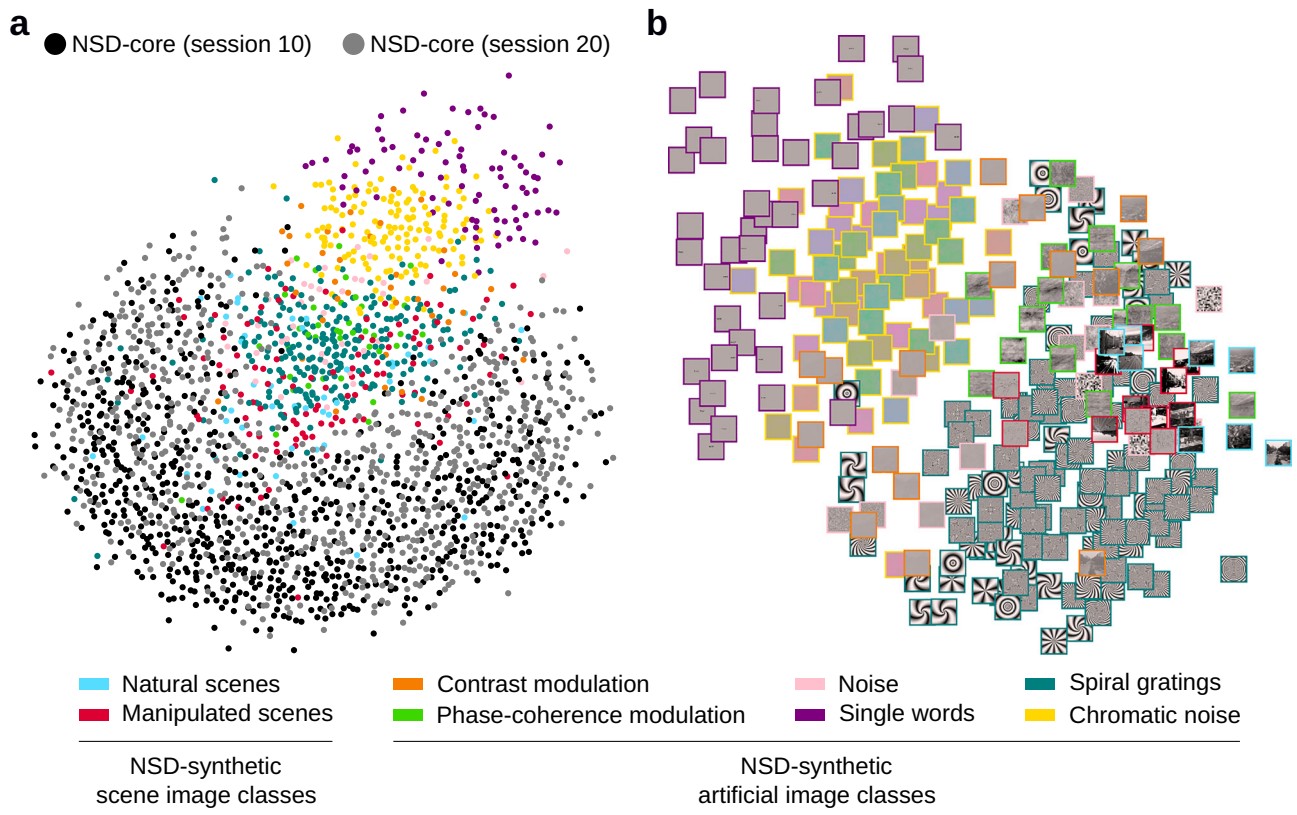

**Fig. 3 | Multidimensional scaling (MDS) confirms brain responses to NSD-synthetic's artificial stimuli are out-of-distribution with respect to natural scenes. a** MDS embedding of single-trial fMRI responses from the NSD-synthetic scan session and two NSD-core scan sessions. Colored dots represent trials from the

NSD-synthetic session, color-coded according to image class. Black and gray dots represent trials from two NSD-core sessions, respectively. **b** MDS embedding of trial-averaged fMRI responses to each of the 284 NSD-synthetic images. Colored borders indicate image class.

spatial noise correlations shared across areas. However, the multivariate responses from different ROIs are also highly similar across participants (off-subdiagonal entries in Fig. 2f). Since noise is uncorrelated between participants, this bypasses the effect of spatial noise correlations and indicates similarity of visual representations across areas.

In summary, NSD-synthetic's fMRI responses reliably encode stimulus-related information with the strongest stimulus-driven responses occurring in EVC ROIs. The encoded information conforms to known tuning properties of EVC and HVC ROIs, and is consistent across participants.

### NSD-synthetic's fMRI responses are out-of-distribution with respect to NSD-core

To be well suited for OOD generalization tests, NSD-synthetic's fMRI responses should not only reliably encode stimulus-related visual information, but should also exhibit a different distribution compared to NSD-core's responses. To ascertain this, we reduced NSD-synthetic and NSD-core's fMRI responses to two dimensions using multidimensional scaling (MDS)[42,43], and compared the resulting two-dimensional MDS embeddings. To evaluate whether differences in

distribution are due to unwanted session-specific effects, we applied MDS jointly on NSD-synthetic's single-trial fMRI responses and on the single-trial fMRI responses from two NSD-core scan sessions (sessions 10 and 20), aggregated across participants. We found that the single-trial fMRI responses for NSD-synthetic form a distinct cluster compared to the single-trial fMRI responses from NSD-core (Fig. 3a). The fact that the single-trial fMRI responses from the two NSD-core scan sessions overlap in the MDS embedding space suggests that the difference between NSD-synthetic and NSD-core's data distributions is not driven by session-specific effects, but rather by stimulus-related signals.

These two-dimensional MDS embeddings additionally revealed two clear patterns visible by eye. First, NSD-synthetic's responses for scene images are closer in the MDS embedding space to NSD-core (Fig. 3a), which is reasonable given that NSD-core consists of fMRI responses to scenes. Second, NSD-synthetic's fMRI responses cluster based on the image class to which they belong (Fig. 3a). To visualize this more effectively, we applied MDS on NSD-synthetic's trial-averaged fMRI responses for the 284 stimulus images only, and plotted these images positioned at the two-dimensional MDS embedding coordinates (Fig. 3b). Again, we see a clear clustering based on image

classes. This clustering is consistent with properties of NSD-synthetic's stimulus images that are visible by eye, and is geometrically similar to the class clustering from applying MDS jointly on NSD-synthetic and NSD-core's responses (Fig. 3a).

In summary, NSD-synthetic's fMRI responses are differently distributed (i.e., OOD) with respect to NSD-core. These distributional differences are driven by the distinctness of NSD-synthetic's visual stimuli, and the similarity structure of NSD-synthetic's fMRI responses appears to mirror what is visible in the stimulus images.

## NSD-synthetic enables out-of-distribution generalization tests for models of the brain

Having established that NSD-synthetic is OOD with respect to NSD-core, we turn to assessing whether NSD-synthetic is useful for OOD generalization tests of computational models of the brain. As computational models we used neural encoding models, that is, predictive algorithms that generate brain responses to arbitrary stimulus images[13,44–47]. We expected that encoding models trained on NSD-core would generalize OOD on NSD-synthetic with lower prediction accuracies than they would for ID generalization.

The encoding models consisted of linear regressions that map image features–downsampled to 250 principal components (PCs) using principal component analysis (PCA)[48]–onto the fMRI responses of each vertex. For image features we used the activations of AlexNet[49], a convolutional deep neural network pretrained on image classification that is a well-established standard in computational visual neuroscience research. We trained the linear regression weights using image features and fMRI responses from NSD-core, and tested their generalization performance both ID and OOD. For ID tests, we used a separate data partition of NSD-core not used for model training; for OOD tests, we used NSD-synthetic. We quantified generalization performance by correlating the predicted fMRI responses with their recorded (i.e., experimentally collected) counterparts, obtaining ID and OOD explained variance scores for each participant and vertex (Fig. 4a). Critically, explained variance scores are determined not only by the encoding model's generalization performance, but also by the level of noise in the recorded fMRI responses (which might differ between the ID and OOD fMRI responses). Thus, to make the ID and OOD explained variance scores more meaningful and directly comparable, we normalized them with the noise ceiling of the corresponding recorded fMRI responses (Fig. 4b). This resulted in ID and OOD noise-ceiling-normalized explained variance scores that indicate the amount of explainable variance accounted for by the models for each vertex and participant (Fig. 4c).

As expected, we found that the encoding models trained on NSD-core predict visual fMRI responses better ID than OOD (Fig. 4c,d) ($P = 10^{-7}$, paired sample $t$-test, one-sided, $N = 8$ participants). The fact that OOD generalization is lower is crucial, as it indicates that further improvement is needed to capture stimulus-to-brain-response relationships, thus validating NSD-synthetic's usefulness for model and theory development. We further observed that decreases in OOD performance compared to ID performance are consistent across both lower- and higher-level visual areas (see red regions in Fig. 4d). Importantly, these decreases in performance are not due to lack of explainable signal in NSD-synthetic's fMRI responses, but rather to model failures. When plotting the vertex-wise explained variance scores against their noise ceilings, we found that compared to ID tests on NSD-core, the encoding models' explained variance scores were consistently lower relative to the noise ceiling for OOD tests on NSD-synthetic, across both lower- and higher-level visual areas (Fig. 4e).

To ensure that the lower OOD performances are meaningful and robust, we performed three controls. First, to ensure that the lower performance on NSD-synthetic is not simply due to the fact that it was collected in a separate scan session, we trained and tested the encoding models using fMRI responses from separate NSD-core scan sessions (Supplementary Fig. 2). Second, to ensure that the specific regression method that one chooses to use does not change our conclusions, we trained and tested the encoding models using individual AlexNet layers, by either mapping each layer's first 250 PCs onto fMRI responses with linear regression, or by mapping the layer's full feature space using ridge regression (Supplementary Figs. 3-4). Third, to ensure that our results are not contingent on a specific choice of deep neural network, we trained and tested the encoding models on image features from deep neural networks other than AlexNet (Supplementary Figs. 5–7). In all three control analyses, we observed lower OOD performances for NSD-synthetic than for NSD-core. Importantly, controlling for session-specific effects led to a considerable decrease in ID (Δ mean noise-ceiling-normalized explained variance score of 6.5%), but not OOD (Δ mean noise-ceiling-normalized explained variance score of 0.18%) generalization. Therefore, for a clearer comparison between ID and OOD generalization performances using NSD-synthetic, we recommend researchers to train and test their models on data from independent scan sessions. Furthermore, neither of the two regression methods led to best OOD generalization performances for all AlexNet layers, suggesting that the optimal regression strategy might be contingent on the specific data scenario.

In summary, we showed that encoding models trained on NSD-core generalize ID and, to a lower extent, OOD on NSD-synthetic. The lower OOD generalization performance opens the door for using NSD-synthetic in combination with NSD-core to perform OOD generalization tests for model and theory improvement.

## Out-of-distribution generalization tests reveal differences between brain models not detected by in-distribution tests

After demonstrating that NSD-synthetic enables OOD generalization tests, we ask whether these tests reveal differences between brain models that are not detected by ID tests. If so, NSD-synthetic would provide unique insight essential for developing more robust models of the brain.

To assess this, we compared the ID and OOD noise-ceiling-normalized explained variance scores of AlexNet-based encoding models (Fig. 5a) with the scores of encoding models based on a more modern computer vision architecture, a vision transformer pretrained on image classification called vit_b_32[50] (Fig. 5b). When tested ID, we found that AlexNet outperformed vit_b_32 in lower-level visual areas (blue areas in Fig. 5c, left panel) and that vit_b_32 outperformed AlexNet in higher-level visual areas (red areas in Fig. 5c, left panel). This ID difference between the two models was modest, peaking at 10% of absolute explained variance difference (Fig. 5c, left panel). Surprisingly, when testing the models OOD we found that vit_b_32 instead outperformed AlexNet across both lower- and higher-level visual areas (red areas in Fig. 5c, right panel). This OOD difference between the two models was much stronger than their ID difference, peaking at 25% of absolute explained variance difference (Fig. 5c, right panel). Thus, OOD tests revealed differences between brain models not detected by ID tests. These differences suggest that vit_b_32 is a more robust model of visual cortex compared to AlexNet. Furthermore, the magnitudes of absolute explained variance difference scores indicate that OOD tests allow to better disentangle brain models compared to ID tests.

In a last comparison, we assessed whether OOD tests reveal unique differences also between brain encoding models based on highly similar deep neural networks. Specifically, we compared encoding models based on ResNet50[51], a convolutional neural network with skip connections pretrained on task-supervised image classification, with encoding models based on MoCo[52], consisting of the same ResNet50 architecture but trained with self-supervised contrastive learning[53]. When tested ID, we found that ResNet50 outperformed MoCo in higher-level visual areas (blue areas in Fig. 5d, left panel), and that MoCo outperformed ResNet50 in lower-level visual areas (red

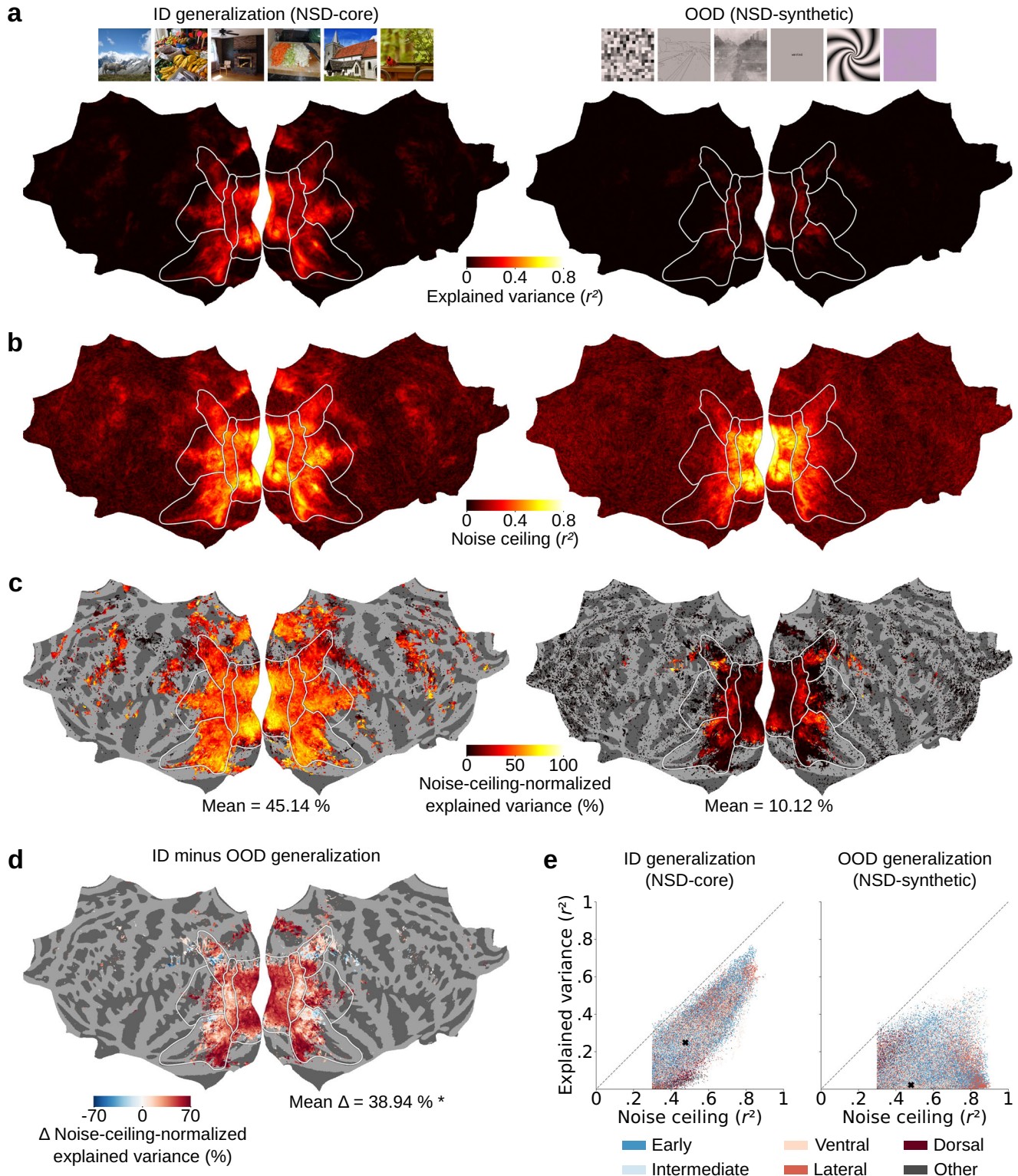

**Fig. 4 | Brain encoding models exhibit reduced performance when tested out-of-distribution on NSD-synthetic.** We built AlexNet-based encoding models and tested them in-distribution (ID) on NSD-core as well as out-of-distribution (OOD) on NSD-synthetic. This figure shows results averaged across participants on flattened cortical surfaces. **a** Model explained variance ($r^2$). **b** Noise ceiling ($r^2$). **c** Model explained variance normalized by the noise ceiling. To reduce false positives, for each vertex, we averaged results from only those participants with a noise ceiling greater than 0.3 (vertices for which no participant has a noise ceiling above 0.3 are not shown). **d** Difference between ID and OOD noise-ceiling-normalized explained variance. For each vertex, we averaged results from only those participants with

noise ceiling greater than 0.3 for both NSD-core and NSD-synthetic. The asterisk indicates a significant difference between the ID and OOD noise-ceiling-normalized explained variance scores ($P = 10^{-7}$, paired sample $t$-test, one-sided, $N = 8$ participants). **e** Scatterplots of vertex-wise explained variance scores against their noise ceilings for ID (NSD-core) and OOD (NSD-synthetic) generalization tests, color coded by the visual stream that each vertex belongs to. For each vertex, we plotted results from only those participants with noise ceiling greater than 0.3 for both NSD-core and NSD-synthetic (same as in **d**). Black crosses indicate median scores across all vertices.

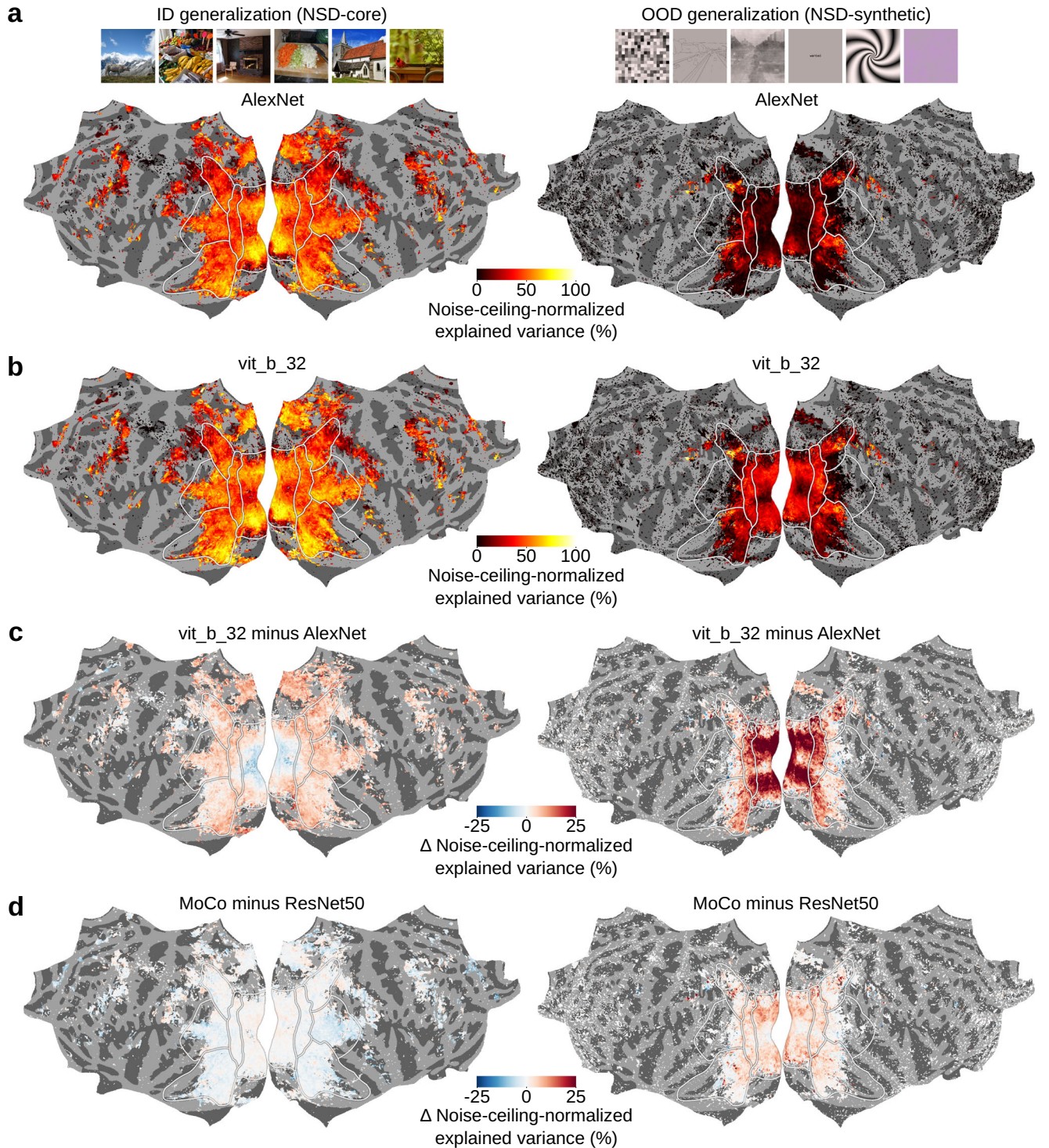

**Fig. 5 | Out-of-distribution generalization tests reveal differences between brain encoding models not detected by in-distribution tests.** We built deep-neural-network-based encoding models and tested them in-distribution (ID) on NSD-core as well as out-of-distribution (OOD) on NSD-synthetic. This figure shows results averaged across participants on flattened cortical surfaces. To reduce false positives, for each vertex we averaged results from only those participants with a noise ceiling greater than 0.3 (vertices for which no participant has a noise ceiling above 0.3 are not shown). **a** AlexNet encoding models' explained variance normalized by the noise ceiling. **b** vit_b_32 encoding models' explained variance normalized by the noise ceiling. **c** Difference between AlexNet and vit_b_32's noise-ceiling-normalized explained variance. **d** Difference between MoCo and ResNet50's noise-ceiling-normalized explained variance.

areas in Fig. 5d, left panel). When tested OOD, MoCo instead outperformed ResNet50 across both lower- and higher-level visual areas (red areas in Fig. 5d, right panel). The ID difference between the two models was smaller (peaking at 5% of absolute explained variance difference, Fig. 5d, left panel) compared to their OOD difference (peaking at 10% of absolute explained variance difference, Fig. 5d, right panel), again indicating that OOD tests allow to better disentangle brain models compared to ID tests. Thus, OOD tests reveal differences not detected by ID tests also for encoding models based on highly similar deep neural networks.

To ensure that the OOD differences not detected by ID tests are meaningful and robust, we performed two controls. First, to ensure that our results are not simply due to the fact that NSD-synthetic was collected in a separate scan session, we trained and tested the encoding models using fMRI responses from separate NSD-core scan sessions (Supplementary Fig. 8). Second, to ensure that the specific regression method that one chooses to use does not change our conclusions, we trained and tested the encoding models on image features from individual ResNet50 or MoCo layers, by either mapping each layer's first 250 PCs onto fMRI responses with linear regression, or by mapping the layer's full feature space using ridge regression (Supplementary Figs. 9–12). In both control analyses, OOD tests revealed differences between models not detected by ID tests.

In summary, these results offer a proof of principle that OOD tests reveal differences between models that are not detected ID. This indicates that the outcomes of OOD and ID generalization tests do not imply one another, and therefore that NSD-synthetic provides unique insight for model development.

### The degree of out-of-distribution is informative of brain model failures

The neural responses on which a brain model is tested can exhibit various degrees of OOD–that is, various degrees of distributional distance–with respect to the responses on which the model is trained. Therefore, we ask whether there is a relationship between the extent to which the fMRI responses used to test an encoding model are differently distributed from the train responses and the encoding model's generalization performance to these test responses. If so, the degree of OOD could serve as an indicator of the stimulus properties that encoding models might fail to account for, which may help guide the engineering of more robust models[24,54]. We operationally define the degree of OOD as the Euclidean distance between the fMRI response to a given stimulus of interest and the NSD-core train responses in two-dimensional MDS space.

Using MDS, we reduced the trial-averaged fMRI responses for NSD-synthetic (used to test the encoding models) and for NSD-core (used to train the models), to two dimensions. Within this two-dimensional embedding space, we then computed the Euclidean distance between the responses of each NSD-synthetic image and the responses of each NSD-core image, and averaged these distance scores across all NSD-core images, as well as across NSD-synthetic images belonging to the same image class. This resulted in distributional distance scores indicating, for each NSD-synthetic image class, the degree of OOD of its fMRI responses from NSD-core's responses (Fig. 6a).

Next, we compared these distributional distance scores with the noise-ceiling-normalized encoding accuracy scores of vision-transformer-based encoding models trained on NSD-core and tested on each NSD-synthetic image class (Fig. 6b). We found that the distributional distance scores are anticorrelated with the encoding accuracy scores (Spearman's $\rho = -0.36$, $P = 0.0004$, one-sample $t$-test, one-sided, $N = 8$ participants) (Fig. 6c), indicating that encoding models generalize better to fMRI responses that have lower distributional distance to the responses on which the models are trained.

As a complementary way of assessing the encoding models' generalization performance for the different NSD-synthetic image classes, we used the models' predicted fMRI responses to identify, in a zero-shot fashion, the stimulus images of NSD-synthetic's recorded fMRI responses[2,44,55,56] (choosing among 8 randomly selected images per NSD-synthetic image class, for a total of 64 images; the final identification scores reflect the average across 1000 analysis iterations, each with a different random selection of 8 images per image class) (Fig. 6d). We found that the ranks of the correct stimulus image (i.e., the position at which the correct image appears among the models' choices) correlate with the distributional distance scores (Spearman's $\rho = 0.65$, $P = 10^{-6}$, one-sample $t$-test, one-sided, $N = 8$ participants) (Fig. 6e),

indicating that encoding models more reliably identify stimulus images when the corresponding fMRI responses have lower distributional distance to the model's train responses.

We obtained quantitatively similar results when using encoding models based on other deep neural network architectures (Supplementary Figs. 13–15), indicating that the correlation between distributional distance scores and encoding model performance is not contingent on a specific choice of deep neural network.

In summary, these results show that the degree to which an encoding model's test fMRI responses are OOD with respect to the model's train responses is informative of the model's generalization performance to these test responses. Thus, the degree of OOD is a useful indicator of the stimulus properties that encoding models might fail to account for.

### NSD enables out-of-distribution tests of brain models across both naturalistic and synthetic stimulus images

Up to here, we implemented OOD generalization tests using NSD-synthetic. To highlight that OOD tests need not be restricted to neural responses for synthetic stimuli, in this last analysis, we assess the OOD generalization performance of vision-transformer-based encoding models to fMRI responses for both naturalistic and synthetic images.

We first used a combination of MDS and clustering algorithms to divide NSD-core's fMRI responses into train, ID test, and naturalistic OOD test splits[23]. For each participant, we reduced the trial-averaged fMRI responses for NSD-core's images to two dimensions through MDS, and partitioned this two-dimensional embedding space into 15 clusters using k-means clustering. For the naturalistic OOD test split we considered the fMRI responses from the cluster with highest Euclidean distance from the centroids of the remaining 14 clusters and, within it, selected fMRI responses for the 284 images with the highest Euclidean distance from the remaining clusters. For the ID test split we selected fMRI responses for 284 images randomly selected from the remaining 14 clusters. As the train split, we selected the images from the remaining 14 clusters not belonging to the ID test split. These procedures ensured that the naturalistic OOD test responses are maximally distant in MDS embedding space from the distribution of the train responses, and that the train and ID test responses come from the same distribution (Fig. 7a). We then trained encoding models using the train split and evaluated them ID (on NSD-core), as well as on naturalistic (on NSD-core) and synthetic (on NSD-synthetic) OOD test fMRI responses.

We obtained higher noise-ceiling-normalized encoding accuracy scores for ID testing on NSD-core, followed by naturalistic OOD testing on NSD-core, and finally by synthetic OOD testing on NSD-synthetic (Fig. 7b). We obtained the same ranking when using the encoding models' predicted fMRI responses to identify, in a zero-shot fashion, the stimulus images of the ID, naturalistic OOD, and synthetic OOD test fMRI responses (choosing among all 852 images from the three test splits) (Fig. 7c). This ranking was expected, as it reflects the fact that the ID, naturalistic OOD, and synthetic OOD test fMRI responses are increasingly distant from the train responses in MDS space (Fig. 7a).

To ensure that this ranking between test splits is meaningful and robust, we performed two controls. First, to ensure that the ranking is not contingent on a specific choice of deep neural network, we trained and tested the encoding models on image features from deep neural networks other than the vision transformer (Supplementary Figs. 16–18). Second, to ensure that the way in which the test splits are defined does not change our conclusions, we selected the images of the ID and naturalistic OOD test splits based, not on empirically observed fMRI responses, but rather on the features of a deep neural network (Supplementary Fig. 19). In both control analyses, we observed decreasing model generalisation performances for ID, naturalistic OOD, and synthetic OOD test splits.

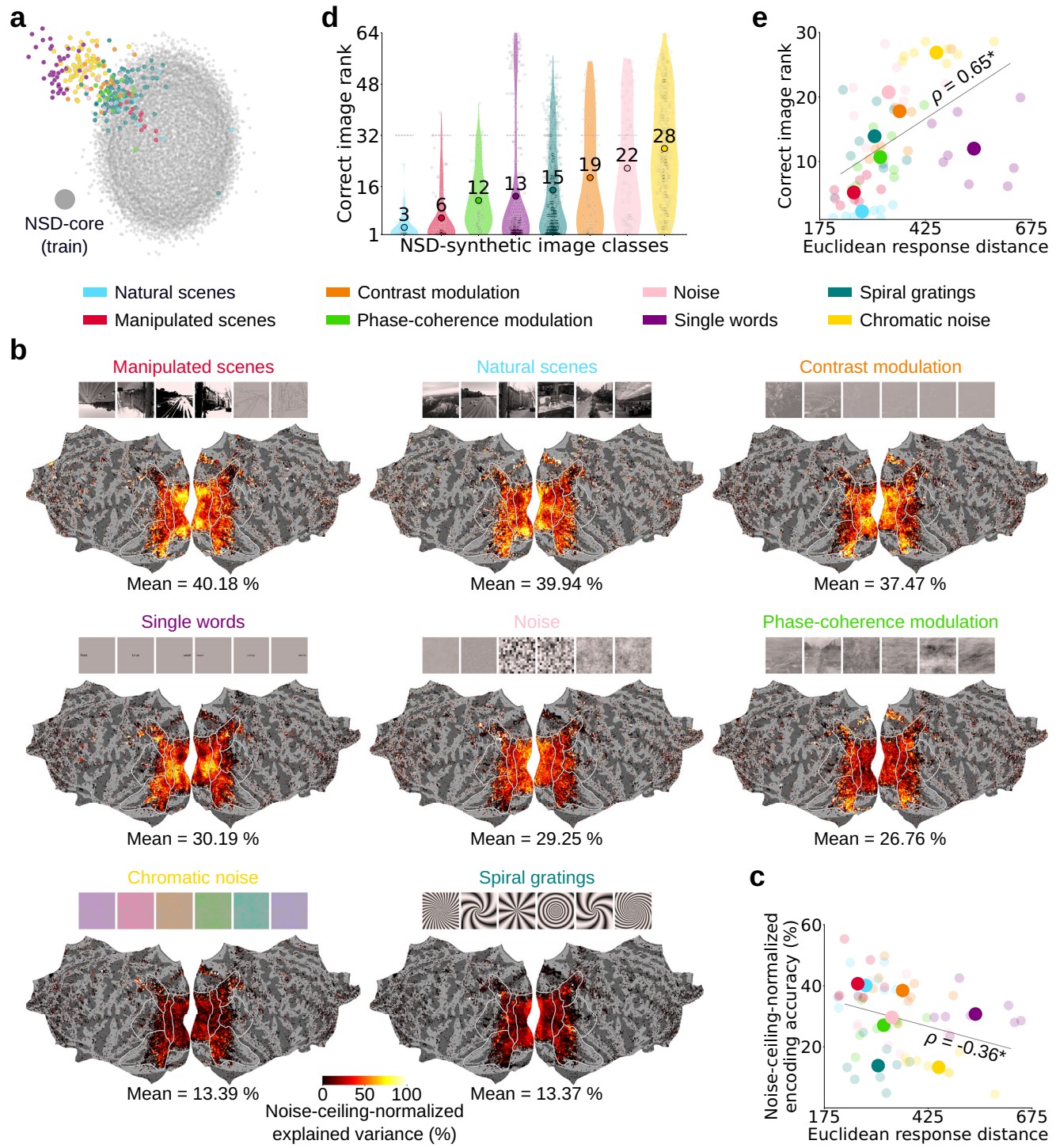

In conclusion, these results showcase the strength of combining NSD-core and NSD-synthetic to test the OOD robustness of brain models on neural responses. The results also demonstrate that the concept of OOD is not restricted to artificial stimuli, but that through carefully-designed train-test splits it can be usefully applied even within the domain of naturalistic stimuli.

## Discussion

Here we release NSD-synthetic, the out-of-distribution (OOD) companion dataset of NSD-core. We have shown that NSD-synthetic's fMRI responses have high signal-to-noise ratio and reliably encode properties of the presented visual stimuli (Fig. 2). Importantly, we have also

demonstrated that the distribution of visually evoked responses in NSD-synthetic is fundamentally different compared to the distribution of visually evoked responses in NSD-core (Fig. 3). Finally, we provided proof of principle that encoding models trained on NSD-core generalize to NSD-synthetic with lower prediction accuracies than they do to NSD-core (Fig. 4); that OOD tests on NSD-synthetic reveal differences between models that are not detected by in-distribution tests on NSD-core (Fig. 5); that the degree of OOD is a useful indicator of the magnitude of model failures (Fig. 6); and that the concept of OOD can be usefully applied even within the domain of naturalistic stimuli (Fig. 7). Together, these findings successfully establish NSD-synthetic as a useful out-of-distribution companion to NSD-core. All data are

**Fig. 6 | The degree of out-of-distribution is informative of brain model failures.** We trained vision-transformer-based encoding models on NSD-core, and tested them independently for each NSD-synthetic image class. **a** MDS embedding of trial-averaged fMRI responses (aggregated across participants) from NSD-core (gray dots) and from NSD-synthetic (colored dots, color-coded according to image class). **b** Encoding models' explained variance for each NSD-synthetic image class, normalized by the noise ceiling. This figure shows results averaged across participants on flattened cortical surfaces. To reduce false positives, for each vertex we averaged results from only those participants with a noise ceiling greater than 0.3 (vertices for which no participant has a noise ceiling above 0.3 are not shown). **c** Scatterplot indicating the relationship between the degree of out-of-distribution (OOD)–that is, the degree of distributional distance–of fMRI responses and the encoding accuracy scores, across NSD-synthetic image classes. The x-axis represents the Euclidean distance in MDS embedding space between fMRI responses for each NSD-synthetic image class and NSD-core (i.e., the distances in **a**). The y-axis represents the noise-ceiling normalized encoding accuracy of encoding models tested on each NSD-synthetic image class (i.e., the mean scores in **b**). **d** Encoding models' zero-shot identification scores of the recorded fMRI response stimulus images, for each NSD-synthetic image class, on violin plots. The y-axis represents the correct image rank, that is, the position at which the correct image appears

among the models' choices (choosing among 8 randomly selected images per NSD-synthetic image class, for a total of 64 images; the final identification scores reflect the average across 1000 analysis iterations, each with a different random selection of 8 images per image class), with lower ranks indicating a more robust image identification. The colored violin shapes indicate the distribution of observations, with wider sections corresponding to higher density of values. Small transparent dots indicate the correct image rank for single images and participants. Large opaque dots indicate the average correct image rank across all participants and images within each NSD-synthetic image class, with the corresponding score reported above the dot. Horizontal dashed lines indicate chance identification scores. **e** Scatterplot indicating the relationship between the degree of OOD of fMRI responses and the zero-shot image identification scores, across NSD-synthetic image classes. The x-axis represents the Euclidean distance in MDS embedding space between fMRI responses for each NSD-synthetic image class and NSD-core (i.e., the distances in **a**). The y-axis represents the average correct image rank across all images within each NSD-synthetic image class (i.e., the average scores in **d**). Large opaque and small transparent dots indicate participant-average and single participant scores, respectively. **c**, **e** The asterisk indicates a significant correlation between response distance scores and encoding model performance ($P < 0.001$, one-sample $t$-test, one-sided, $N = 8$ participants).

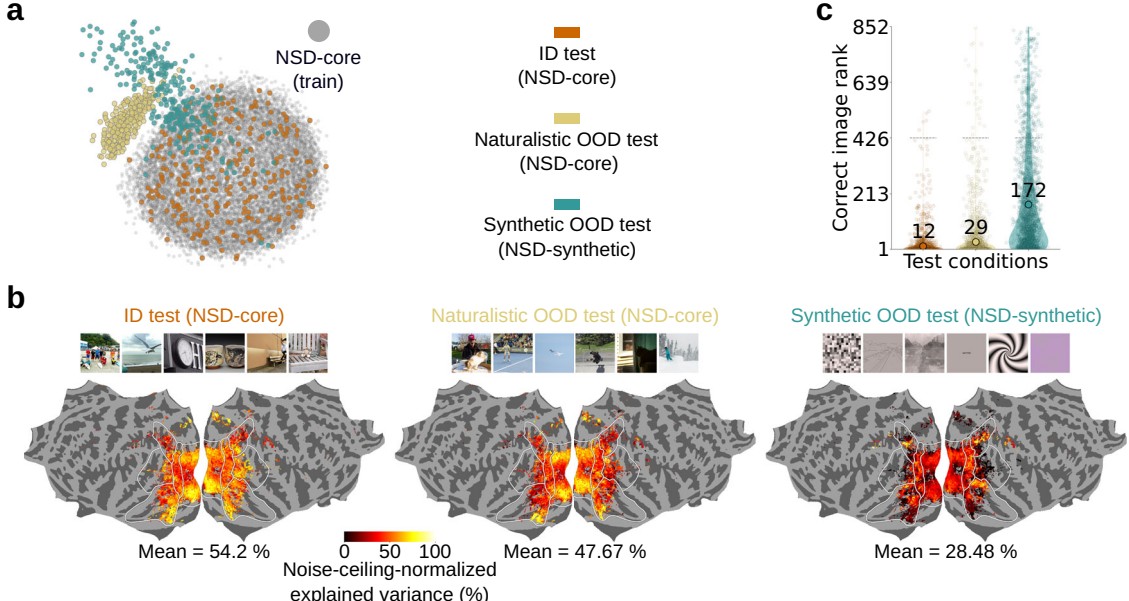

**Fig. 7 | NSD enables out-of-distribution tests of brain models across both naturalistic and synthetic stimulus images.** We trained vision-transformer-based encoding models on NSD-core, and tested their generalization across three conditions: in-distribution (ID) on NSD-core, naturalistic out-of-distribution (OOD) on NSD-core, and synthetic OOD on NSD-synthetic. **a** MDS embedding of trial-averaged fMRI responses (aggregated across participants) for the train images (gray dots), for the ID test images (orange dots), for the naturalistic OOD test images (yellow dots), and for the synthetic OOD test images (turquoise dots). **b** Encoding models' explained variance for the three test conditions, normalized by the noise ceiling. This figure shows results averaged across participants on flattened cortical surfaces. To reduce false positives, for each vertex we averaged results from only those participants with a noise ceiling greater than 0.3 for both

NSD-core and NSD-synthetic (vertices for which no participant has a noise ceiling above 0.3 are not shown). **c** Encoding models' zero-shot identification scores of the recorded fMRI response stimulus images, for each of the three test conditions, on violin plots. The y-axis represents the correct image rank, that is, the position at which the correct image appears among the models' choices (choosing among all 852 images from the three test splits), with lower ranks indicating a more robust image identification. The colored violin shapes indicate the distribution of observations, with wider sections corresponding to higher density of values. Small transparent dots indicate the correct image rank for single images and participants. Large opaque dots indicate the average correct image rank across all participants and images within each test condition, with the corresponding score reported above the dot. Horizontal dashed lines indicate chance identification scores.

publicly available, including data in ready-to-use preprocessed formats, at http://naturalscenesdataset.org. We hope researchers will take advantage of this new dataset in building more robust and accurate computational models of the brain, leading to better theories of vision.

OOD generalization refers to modeling scenarios where the test data distribution differs from the train data distribution, resulting in stricter tests of model robustness compared to in-distribution (ID) tests[22]. In line with this, we found that NSD-synthetic's fMRI responses are indeed distributed differently than NSD-core's responses, and that encoding models trained on NSD-core achieve lower generalization

performance when tested OOD on NSD-synthetic. Finding a lower OOD generalization performance is essential for establishing NSD-synthetic as a stricter testbed that highlights deficiencies of brain models not observable with ID tests.

Beyond indicating that further model improvement is required to accurately describe visual representations in the brain, NSD-synthetic also suggests directions for closing the gap between ID and OOD generalization performance. One direction of model improvement comes from consideration of the properties of NSD-synthetic's stimuli. Since the stimuli consist of a multitude of diverse and parameterized

images, they facilitate discovery of the specific visual features to which computational models fail to generalize. Notably, we found that the generalization performance for the different NSD-synthetic image classes varied considerably, from performances on par to the ones obtained ID (e.g., mean noise-ceiling-normalized encoding accuracy of 40.18% for manipulated scenes), to substantially lower performances (e.g., 13.37% for spiral gratings). Future research efforts could explore why certain visual features are modeled better than others, for example, why fMRI responses for the noise image class are better modeled than responses for the chromatic noise class. We additionally found that the generalization performances of the NSD-synthetic image classes negatively correlated with their degree of OOD. In light of these observations, we suggest that future modeling efforts may benefit from focusing on improving model robustness for stimulus properties with a high degree of OOD[24,54].

Another direction of model improvement comes from benchmarking the performance of different computational models on NSD-synthetic and isolating model properties leading to best OOD generalization[25,26]. Because each model embeds a different hypothesis of visual processing, these OOD generalization tests can help adjudicate between competing hypotheses, therefore facilitating theory formation. We found that encoding models based on a self-supervised deep neural network better generalize OOD than encoding models based on its task-supervised counterpart, in line with recent work proposing self-supervision as a more plausible account of coding in visual cortex[9,21,57]. The improved performance of self-supervised deep neural networks might derive from their training diet, which beyond colored naturalistic images also includes augmented images such as grayscale or blurred images. Compared to the fully naturalistic colored images typically used to train task-supervised deep neural networks, these augmented images are more similar to the artificial images from NSD-synthetic, therefore resulting in encoding models with better OOD generalization performances. Future efforts that systematically benchmark a larger variety of encoding models[16] might confirm this finding, and also isolate further model properties that improve OOD generalization performance. As another illustrative example, St-Yves and colleagues used NSD-synthetic to compare OOD generalization between encoding models with or without hierarchical visual representations. They found that non-hierarchical representations most accurately predict NSD-synthetic's fMRI responses, whereas the difference between non-hierarchical and hierarchical models was much less apparent when testing predictions ID on NSD-core. Based on these findings, the authors derived the theoretical conclusion that "*instead of forming a serial processing cascade that supports a single function, the diverse representations in V1–V4 may subserve diverse and independent functions that are distributed across the visual areas*"[20].

To obtain valid inferences, the details, quantifications, and interpretation of OOD tests require careful consideration. Here, we comment on four specific aspects that researchers should consider when performing OOD generalization tests.

First, there is a continuum of degrees to which the test data can be OOD with respect to the train data, which in turn affects the generalization performance of computational models of the brain. We showed this for test fMRI responses for different NSD-synthetic image classes (Fig. 6), as well as for test fMRI responses for naturalistic and synthetic images (Fig. 7). Our empirical results reinforce the fact that OOD generalization performance of brain models cannot be fully described by a simple summary number, but lives in a continuum of distances between the train and test data distributions. Combining NSD-core and NSD-synthetic provides a way to test several degrees of OOD generalization across this continuum, from lower degrees of OOD and thus less strict generalization tests using fMRI responses for NSD-core's naturalistic stimuli, to higher degrees of OOD and thus stricter tests using NSD-synthetic. (However, stricter OOD generalization tests are in principle also possible with NSD-core's naturalistic stimuli, through a more constrained division of train-test splits than the one employed in this work.) We envision that this will endow researchers with greater flexibility in model testing, resulting in a more nuanced account of the OOD generalization performance of brain models.

Second, there are several approaches to implement and test computational models of the brain, each with its unique strengths, and which approach should be best used towards a particular scientific goal is a current topic of debate[9,58,59]. Here, we used the linearizing encoding approach (Naselaris et al., 2011), which consists of two independently trainable components: the image-computable model (typically a deep neural network trained on a computer vision task) used to extract the image features, and the linear mapping trained to map these image features onto brain responses. Thus, the outcomes of our OOD tests reflect the combination of both the image-computable model and the linear mapping. We note that other researchers might wish to test only the OOD generalization performance of the image-computable model. For this modeling philosophy, a different analytic approach would be necessary, such as representational similarity analysis[41] (with no feature re-weighting), or determining a one-to-one mapping between model features and neural units[60].

Third, we interpreted the distributional differences between NSD-synthetic and NSD-core's fMRI responses as stemming from differences in the stimuli of the two datasets. However, in theory, distributional differences might also reflect other properties of the data, such as differences in task[61] or trial timing[62], session-specific effects, and fluctuations in cognitive state. All these data properties represent diverse dimensions across which different datasets might fall in- or out-of-distribution. Therefore, to improve transparency, reproducibility, and comparison across studies using NSD-synthetic, we recommend researchers to explicitly define the OOD operationalization used.

Fourth, properties of the data not relevant to the researcher's question of interest might introduce spurious distributional differences between NSD-synthetic and NSD-core's fMRI responses. For example, the trial timing differed between NSD-core (3-s ON, 1-s OFF) and NSD-synthetic (2-s ON, 2-s OFF), with longer image onscreen times typically leading to higher fMRI responses[62]. Similarly, NSD-core and NSD-synthetic differed in task, with tasks also affecting fMRI response magnitude[61]. Additionally, since NSD-synthetic and NSD-core's fMRI responses come from distinct scan sessions, they are affected by session-specific effects, that is, differences in fMRI responses across sessions due to incidental session-specific factors (e.g., time of day, arousal, cognitive state, hardware state). To eliminate spurious distributional differences, researchers might consider $z$-scoring fMRI responses within each scan session. However, by centering responses and transforming them to unit variance, $z$-scoring also alters the signal of interest, potentially eliminating distributional differences of interest. Our approach for compensating for differences in fMRI response gain due to trial timing was to quantify encoding model performance using Pearson correlation (which is insensitive to gain differences). Furthermore, to reduce session-specific effects we analyzed fMRI responses aggregated across participants, considered vertices with a signal-to-noise ratio score above a certain threshold (see Methods), and performed control analyses where we trained and tested encoding models on fMRI data from separate NSD-core scan sessions.

There are other large-scale visual fMRI datasets that have OOD components. One such OOD component comes from the Algonauts Project 2025 challenge dataset[63,64] which consists of fMRI responses to multimodal movies including visual, auditory and linguistic stimulation. In this dataset, the test split was chosen to be OOD compared to the train split based on different movie dimensions (e.g., emotional valence, movie genre, language, background music, visual style). NSD-synthetic complements this dataset by offering fMRI responses from a controlled stimulus set and experimental design (controlled artificial images presented sequentially while participants maintained central

fixation and performed two different tasks versus uncontrolled naturalistic multimodal movies presented while participants freely performed eye movements without an explicit task). In addition, NSD-synthetic provides fMRI responses at higher magnetic field strength (7T versus 3T) and with twice as many participants (8 versus 4), though the Algonauts 2025 dataset does include more hours of fMRI data per participant. Other large-scale fMRI datasets with OOD components include the Deep Image Reconstruction dataset[65] and the Visual Illusion Reconstruction dataset[66]. These datasets consist of fMRI responses to naturalistic images of objects and scenes (ID component), as well as to artificial images, letters, and illusory images (OOD component). NSD-synthetic complements these datasets with fMRI responses for a different and larger set of OOD stimulus images (284 images versus 50 or 38 for the Deep Image Reconstruction or Visual Illusion Reconstruction datasets, respectively). In addition, NSD-synthetic provides fMRI responses at higher magnetic field strength (7T versus 3T) and a larger ID component on which to train computational models of the brain (up to 30,000 trials per participant in NSD-core versus up to 7200 or 16,000 trials per participant for the Deep Image Reconstruction or Visual Illusion Reconstruction datasets, respectively). Overall, we emphasize that a key feature of NSD-synthetic is that it was conducted in concert with NSD-core. Over the last three years, NSD-core has become the most widely used dataset and a de-facto benchmark in computational visual neuroscience. Hence, there are exciting opportunities for enriching the insights derived from NSD.

While NSD-synthetic's main strength is to enable OOD generalization tests, the dataset is also suitable for other use cases. First, it contains several well-controlled image manipulations that enable characterization of how specific visual dimensions affect brain responses (independent of any computational model). For example, NSD-synthetic stimuli allow an examination of whether and how neural representations differ when viewing a natural scene upside-down[67], viewing a binarized version of the scene[68], or viewing a line drawing of the scene[69,70]. Second, it contains fMRI responses across two different visual tasks: a fixation task where participants reported increments and decrements of the luminance of the fixation dot and a one-back task where participants judged whether the currently presented image was identical to the previously presented image. Given that the same stimuli were presented across the two tasks, these data allow investigation of whether and how cognitive tasks influence neural representations[61].

The results of our analyses indicate that stimulus-related information in NSD-synthetic's fMRI responses is primarily encoded in early to intermediate visual cortical areas (i.e., V1 to hV4). This presumably is due to the fact that NSD-synthetic stimuli consist mostly of simple visual features that well activate early to intermediate visual cortex (e.g., various forms of noise and spiral gratings), but which are not well suited for driving activity in higher visual cortical areas responsive to perceptually and semantically more complex stimuli[27–32]. Because of this, the results and conclusions obtained with NSD-synthetic might be most robust when testing the OOD generalization of computational models of lower visual cortical areas.

## Methods

Procedures for data acquisition, data pre-processing, and GLM analysis were the same as in NSD-core. For full details of these procedures, please refer to NSD-core's paper[1]. Below, we provide a detailed summary of the procedures.

### Participants

The NSD-synthetic experiment was conducted with the same set of eight participants who took part in NSD-core. These participants (subj01–subj08) were right-handed with normal or corrected-to-normal visual acuity and no color blindness, and consisted of two males and six females, with age range 19–32 years. Informed written consent was obtained from all participants, and the experimental protocol was approved by the University of Minnesota Institutional Review Board. Participants were compensated at a rate of $30 per hour, plus performance bonuses. Additional participant information, including height, weight, handedness and visual acuity, was logged and is available online.

### MRI data acquisition

MRI data were collected at the Center for Magnetic Resonance Research at the University of Minnesota. Functional MRI data were collected using a 7T Siemens Magnetom passively shielded scanner and a single-channel-transmit, 32-channel-receive RF head coil (Nova Medical). Data for NSD-synthetic were collected in one scan session that took place after completion of the NSD-core experiment. The fMRI data were collected using gradient-echo EPI at 1.8 mm isotropic resolution with whole-brain coverage (84 axial slices, slice thickness 1.8 mm, phase encode direction anterior-to-posterior, TR 1600 ms, TE 22.0 ms, partial Fourier 7/8, iPAT 2, multi-band slice acceleration factor 3). In addition to the EPI scans, the scan session also included dual-echo fieldmaps for post hoc correction of EPI spatial distortion (same overall slice slab as the EPI data, 2.2 mm × 2.2 mm × 3.6 mm resolution, TE1 8.16 ms, TE2 9.18 ms). The acquisition structure for the NSD-synthetic scan session was [F XOXO F XOXO F], where F indicates a fieldmap, X indicates a fixation run (268 TRs), and O indicates a one-back run (268 TRs).

### Stimulus display and scanner peripherals

Stimuli were presented using a Cambridge Research Systems BOLDscreen 32 LCD monitor positioned at the head of the 7T scanner bed. The monitor operated at a resolution of 1920 pixels × 1080 pixels at 120 Hz. Participants viewed the monitor via a mirror mounted on the RF coil. The size of the monitor image was 69.84 cm (width) × 39.29 cm (height), and the viewing distance was 176.5 cm. Measurements of display spectral power density were obtained using a PR-655 spectroradiometer (Photo Research). A Mac Pro computer controlled stimulus presentation using code based on Psychophysics Toolbox 3.0.14. Behavioral responses were recorded using a button box (Current Designs). Ear plugs were used to reduce acoustic noise, and head motion was mitigated using headcases.

Eye-tracking was performed using an EyeLink 1000 system (SR Research). We caution that the eye-tracking data are variable in quality due to difficulties achieving sufficient pupil contrast in some participants. In addition, eye-tracking data for the first NSD-synthetic run for subj05 were corrupted and are therefore not available. For full details of eye-tracking acquisition and pre-processing, see NSD-core's paper[1]. Overall, we found that participants were generally able to maintain good central fixation during our various fMRI experiments, with the possible exception of subj05 (see NSD-core's Extended Data Fig. 4[1]).

### The NSD-synthetic experiment

**Stimuli.** Stimuli consisted of 284 distinct images. All images were presented under two different tasks, a fixation task and a one-back task. A high-level description of the images is as follows (the number of images is indicated in parentheses): white noise (4), white noise with a large block size (4), pink noise (4), natural scenes (4), upside-down versions of these scenes (4), Mooney versions of these scenes (4), line-drawing versions of these scenes (4), contrast-modulated natural scenes (4 scenes × 5 contrast levels (100%, 50%, 10%, 6%, 4%) = 20), phase-coherence-modulated natural scenes (4 scenes × 4 coherence levels (75%, 50%, 25%, 0%) = 16), single words (2 word lengths × 5 positions × 4 words = 40), spiral gratings varying in orientation and spatial frequency (112), and chromatic pink noise varying in hue (68).

Images were prepared at a resolution of 1360 pixels (width) × 714 pixels (height), corresponding to a visual field extent of 16° × 8.4°.

Most images occupy the central 8.4° × 8.4°, which is the stimulus size used in NSD-core; a few word stimuli extend beyond this central region (details below). Of the 284 images, the first 220 images are achromatic. Pixel luminance values for these images were prepared in the range 0–1 and then transformed using a color lookup table that increases linearly in luminance (L + M) with the chromaticity of D65. In this lookup table, the RGB values corresponding to the minimum and maximum luminance are (0,0,0) and (252,220,216), respectively. The remaining 64 images are chromatic, and their design is described in further detail below.

The images are hierarchically organized: there are 284 distinct images belonging to 71 image subclasses, and these 71 image subclasses belong to 8 image classes (Noise, Natural scenes, Manipulated scenes, Contrast modulation, Phase-coherence modulation, Single words, Spiral gratings, and Chromatic noise). Below, we describe each of the image subclasses. The parentheses at the end of each description indicate the associated image class.

**White noise: Image subclass 1 (images 1–4).** Each image consists of pixels whose luminance values were randomly drawn from a uniform distribution between 0 and 1. Four distinct images were generated (Noise).

**White noise (large block): Image subclass 2 (images 5–8).** Pixels are grouped into block sizes of 42 pixels × 42 pixels (0.49° × 0.49°). Luminance values for each block were randomly drawn from a uniform distribution between 0 and 1. Four distinct images were generated (Noise).

**Pink noise: Image subclass 3 (images 9–12).** Images with $1/f$ amplitude spectra were generated. This was achieved by multiplying white Gaussian noise by the desired amplitude spectrum. For each image, pixel values from 3.5 standard deviations below and above the mean were linearly mapped to the range 0–1. Four distinct images were generated (Noise).

**Natural scenes: Image subclass 4 (images 13–16).** Four natural scenes were chosen from the stimulus set used in a previous study[69]; the motivation for choosing from this stimulus set is that the natural scenes have associated line drawings (which are used in image subclass 7). The four chosen scenes depict mountains, a highway, a street, and a desk. Each natural scene was converted to grayscale, square cropped, resized to 714 pixels × 714 pixels, converted to luminance values assuming a display gamma of 2.0, and then contrast-normalized such that the 0.1 and 99.9 percentiles were linearly mapped to the range 0–1 (Natural scenes).

**Upside-down scenes: Image subclass 5 (images 17–20).** The prepared images from image subclass 4 were rotated by 180° (Manipulated scenes).

**Mooney scenes: Image subclass 6 (images 21–24).** We binarized the prepared images from image subclass 4 by thresholding each image at the median pixel value. The resulting images have pixel values of either 0 (black) or 1 (white), and are akin to Mooney images[68]. Participants were not explicitly pre-exposed to the intact versions of the images, but were presumably familiarized with the intact images over the course of the scan session (Manipulated scenes).

**Line-drawing scenes: Image subclass 7 (images 25–28).** Line drawings were obtained from a previous study[69]. In brief, trained artists at the Lotus Hill Research Institute created line drawings by tracing the contours of color photographs using a custom interface. The sequence and coordinates of each line stroke were digitally recorded. The line drawings associated with the natural scenes used in image subclass 4 were square cropped, resized to 714 pixels × 714 pixels, and then

prepared as black lines (0) on a gray background (0.5) (Manipulated scenes).

**Natural scenes (second set): Image subclass 8 (images 29–32).** A second set of four natural scenes were chosen from the stimulus set used in the previous study[69]. The four chosen scenes depict a forest, a superhighway, a boulevard, and cubicles. The same image preparation procedures were applied as in image subclass 4 (Natural scenes).

**Contrast 50%, Contrast 10%, Contrast 6%, Contrast 4%: Image subclasses 9–12 (images 33–48).** The prepared images from image subclass 8 were manipulated to have contrast levels of 50%, 10%, 6%, and 4%. This was achieved by designating the prepared images as having a contrast level of 100% and by scaling pixel luminance values towards 0.5 by the appropriate amount (Contrast modulation).

**Phase 75%, Phase 50%, Phase 25%, Phase 0%: Image subclasses 13–16 (images 49–64).** We took the prepared images from image subclass 9 (the 'Contrast 50%' subclass) and manipulated the phase coherence level. Specifically, the phase spectrum of each image was blended with a random phase spectrum (e.g., 'Phase 75%' indicates a blend of 75% original and 25% random). A different random phase spectrum was used for each image and each coherence level. Pixel luminance values were then truncated to fit the range 0–1 (only a very small amount of clipping occurred) (Phase-coherence modulation).

**Word4 Pos1–5, Word6 Pos1–5: Image subclasses 17–26 (images 65–104).** Each image contains a single word (black on a gray background) written in a sans-serif, monospaced font (Liberation Mono). Half of the words are 4 letters long and presented in a relatively large font size (x-height 0.4°, center-to-center spacing 0.43°, word bounding box height 0.69°), and half of the words are 6 letters long in a smaller font size (x-height 0.27°, center-to-center spacing 0.28°, word bounding box height 0.46°). The 4-letter words roughly match the 6-letter words in horizontal extent (average word width 1.64°). The 4- and 6-letter words (20 in each set) are closely matched in lexical frequency. The words span a wide range of syntactic categories (parts of speech). Each word was placed at one of five possible positions along the horizontal meridian (−6°, −3°, 0°, 3°, 6°). For each combination of word length and position, four images were generated, yielding a total of 2 word lengths × 5 positions × 4 words = 40 images. Note that the words positioned at −6° and 6° are the only images (out of the 284 NSD-synthetic images) with content that lies outside of the central 8.4° × 8.4° region. We caution that models trained on NSD-core may generalize poorly to these images due to their relatively peripheral image content (Single words).

**SpiralA–D SF1–6, SpiralE–H SF4: Image subclasses 27–54 (images 105–216).** These images are sinusoidal gratings in log-polar coordinates (we refer to these as 'spiral gratings' but they can also be called 'log-polar gratings'). The primary advantage of spiral gratings over conventional Cartesian sinusoidal gratings is that for spiral gratings, spatial frequency varies inversely with eccentricity, thereby enabling more efficient sampling of neural tuning preferences (neurons at higher eccentricities are tuned for lower spatial frequencies)[71]. We created spiral gratings at full contrast. We prepared four types of spirals, SpiralA–D (pinwheels, forward spirals, annuli, reverse spirals), at six spatial frequency levels ($L$ = 6.0, 11.0, 20.0, 37.0, 69.0, 128.0), and we prepared four additional types of spirals, SpiralE–H (various intermediate spirals), at one spatial frequency level ($L$ = 37.0). For a given point in the visual field with eccentricity $E$ (in degrees), the local spatial frequency (in cycles per degree) is given by $L/(2\pi E)$; hence, for a fixed spatial frequency level $L$, different spirals have identical local spatial frequency content (but differ in local orientation). To avoid aliasing, a central circle was cut out of each image. The diameter of this

circle was 4, 8, 14, 26, 48, and 86 pixels (0.05°, 0.09°, 0.16°, 0.31°, 0.56°, and 1.01°) for the six spatial frequency levels, respectively. We generated a total of 4 types of spirals (SpiralA–D) × 6 spatial frequency levels + 4 types of spirals (SpiralE–H) × 1 spatial frequency level = 28 image subclasses. For each image subclass, four images were generated, corresponding to four equally spaced grating phases. Hence, the total number of images was 28 image subclasses × 4 phases = 112 images. For general information about the design of spiral gratings, please see this previous study[71] (Spiral gratings).

**Hue00: Image subclass 55 (images 217–220).** These images are the same as the prepared images from image subclass 3 (pink noise with $1/f$ amplitude spectra) but lowered in contrast. The images are metameric in chromaticity with standard illuminant D65. Four images were generated (Noise).

**Hue01–16: Image subclasses 56–71 (images 221–284).** These 64 images are the only chromatic images in NSD-synthetic. To create these images, we selected 16 different hues by choosing evenly spaced angles in a version of the MacLeod-Boynton chromaticity diagram[72] that is based on the Stockman, MacLeod, and Johnson cone fundamentals[73]. We then determined the maximum saturation by fitting a circle with the maximum available radius within the gamut of the BOLDscreen display. Finally, for each hue, we created isoluminant images by taking the 4 pink noise patterns from image subclass 55 and using these patterns to modulate between zero saturation (the D65 gray point) and maximum saturation (within the determined circle). To ensure isoluminance for individual observers, we created participant-specific versions of the 64 images based on heterochromatic flicker photometry results obtained from each participant (Chromatic noise).

For most image subclasses (comprising a total of 268 images), images naturally come in groups of four for which we do not expect much variation in brain activity. For example, for the 'White noise' subclass, the four generated images are quite similar to each other (they are just different samples of noise). As another example, for the 'SpiralA SF1' subclass, the four generated images differ only with respect to the specific phase of the underlying spiral grating. Hence, for these image subclasses, we present images only once per task (with the four presentations serving as quasi-repetitions). In contrast, for the 'Natural scenes', 'Upside-down scenes', 'Mooney scenes', and 'Line-drawing scenes' subclasses (comprising a total of 16 images), each distinct scene might be expected to produce a substantially different brain activity pattern. Hence, for these image subclasses, we present images four times per task. Overall, for a complete set of trials for one task, we require 268 images × 1 presentation + 16 images × 4 presentations = 332 stimulus trials.

**Trial and run design.** Each trial lasted 4 s and consisted of the presentation of an image for 2 s, followed by a 2 s gap. (The motivation for the shorter 2 s duration compared to the 3 s duration in NSD-core was to reduce the strong adaptation effects caused by gratings.) In total, 107 trials were conducted in a run; thus, each run lasted 428 s. The first three trials (12 s) and the last four trials (16 s) were blank trials. The remaining 100 trials were divided into 93 stimulus trials and seven blank trials. The blank trials were randomly positioned in each run such that the minimum and maximum number of continuous stimulus trials was nine trials (36 s) and 14 trials (56 s), respectively.

We allocated four runs to complete the data for the fixation task. To determine trial ordering for these four runs, we first created a random ordering of the complete set of 332 stimulus trials, splitting these trials into 4 runs (83 stimulus trials each). This random ordering was subject to the constraint that no image was repeated back-to-back

within a run. Then, for each run, we randomly selected 10 of the stimulus trials to be repeated as one-back trials. This yielded, for each run, a total of 83 regular stimulus trials + 10 inserted one-back trials = 93 stimulus trials.

A total of eight runs were collected in the NSD-synthetic scan session, with participants alternating between the fixation task and the one-back task. To ensure that differences in brain activity across tasks are not due to stimulus-related differences, we took the run design for the fixation task and used it identically for the one-back task (albeit changing the run order). Specifically, the tasks performed in the eight runs were XOXOXOXO, where X indicates the fixation task and O indicates the one-back task, and the run designs for the eight runs were ACBDCADB, where A, B, C, and D correspond to the four runs designed for the fixation task (as described above). All stimulus and experiment characteristics (e.g., trial ordering, one-back events, dot color changes) were kept identical across participants.

**Stimulus presentation and task.** The BOLDscreen monitor was configured to behave as a linear display device, and all stimuli were delivered using a linear color lookup table. (For natural scene stimuli, we prepared the luminance of these stimuli to simulate standard display gamma; see details above.) Stimuli occupied 16.0° (width) × 8.4° (height). Stimulus presentation was locked to the refresh rate of the BOLDscreen monitor, which was nearly exactly 120 Hz. Empirically, we confirmed that the durations of runs were highly reliable, ranging from 427.94 s to 427.96 s. Throughout each run (including blank trials), a small semi-transparent fixation dot with a black border (0.2° × 0.2°, 50% opacity) was present at the center of the stimuli. Every 1.4 s, the luminance of the dot was randomly set to one of five different linearly spaced levels (0.17, 0.37, 0.58, 0.79, 1), with consecutive repetitions allowed. Stimuli were shown against a gray background (luminance level 0.5).

The fixation task (odd runs) was intended to be a challenging task that draws cognitive resources away from the stimuli and towards the small fixation dot[74]. In the fixation task, participants were instructed to fixate the central dot and to press button 1 using the index finger of their right hand when the dot darkens and button 2 using the middle finger of their right hand when the dot lightens. Participants were informed that the dot brightness sometimes stays constant. Participants were encouraged to ignore the stimulus images and focus on the color of the dot.

The one-back task (even runs) was intended to encourage perception of the stimuli. In the one-back task, participants were instructed to fixate the central dot (ignoring the dot color changes) and to press button 1 if the presented image is different from the previous image and button 2 if the presented image was identical to the previous image. In a sense, this task is similar to the continuous recognition task performed in NSD-core where participants were asked to judge whether each presented image is new (not yet seen) or old (presented before). Participants were warned that some images may look degraded, noisy, or abstract, that some images may be positioned off to the side, and that some images may be quite similar but not identical to others. Participants were additionally instructed to continue to fixate and wait for the next image in the event of blank trials.

**Summary of image repetitions and trial counts.** Across the entire NSD-synthetic experiment, there are a total of 93 stimulus trials × 8 runs = 744 stimulus trials. Of these stimulus trials, 80 trials are one-back trials (10.8%). Considering all trials except the one-back trials, 268 images are presented 1 time per task and 16 images are presented 4 times per task. Considering all trials, including the one-back trials, 236 images are presented 1 time per task, 32 images are presented 2 times per task, 8 images are presented 4 times per task, and 8 images are presented 5 times per task.

## Pre-processing of MRI data

Details on MRI pre-processing are provided in the Supplementary Information of NSD-core's paper[1]. In brief, T1-weighted anatomical data were processed using FreeSurfer to create cortical surface representations. Functional data were pre-processed using one temporal resampling to correct for slice time differences and one spatial resampling to correct for head motion within and across scan sessions, EPI distortion, and gradient nonlinearities. Two versions of the functional data were prepared: a 1.8 mm standard-resolution preparation (temporal resolution 1.333 s) and an upsampled 1.0-mm high-resolution preparation (temporal resolution 1.000 s). Population receptive field and functional localizer experiments included in NSD were used to define retinotopic and category-selective regions of interest (ROIs), respectively. The defined ROIs include retinotopic visual areas (V1, V2, V3, hV4) as well as category-selective regions such as extrastriate body area (EBA), fusiform face area (FFA), parahippocampal place area (PPA), and visual word form area (VWFA).

## GLM analysis of fMRI data

We performed a GLM analysis of NSD-synthetic's pre-processed time-series data. This GLM analysis was the same as performed for NSD-core, and provides single-trial BOLD response amplitude estimates ('betas') in units of percent signal change. Our GLM approach combines three analysis components: a library of hemodynamic response functions (HRFs), adaptation of the GLMdenoise technique[75,76] to a single-trial GLM framework, and application of ridge regression[77] as a method for dampening the noise inflation caused by correlated single-trial GLM regressors. The GLM approach has been implemented in a software tool called GLMsingle[78].

In implementing the GLM analysis for NSD-synthetic, we made the following design choices. First, the fixation and one-back tasks were treated separately. Hence, each image was associated with two distinct conditions: one condition reflects the presentation of the image during the fixation task and the other condition reflects the presentation of the image during the one-back task. Second, because the GLMdenoise and ridge regression techniques require condition repeats (so that cross-validated optimization of hyper-parameters can be performed), we designated certain groups of images as repeats of a single condition. This choice reflects the expectation that the images within each group are likely to give very similar evoked brain responses. We specifically designated each of image subclasses 1, 2, 3, and 27–71 as consisting of four images that are repeats of a single condition. Third, to minimize assumptions and avoid complications stemming from any repetition suppression (adaptation) effects, we separately coded the one-back trials such that each one-back trial was assigned its own single-trial regressor. As such, the responses evoked by the one-back trials have minimal influence on the optimization of the GLMdenoise and ridge regression analysis components.

## Analysis of behavioral data

For the fixation task, we extracted the first button response obtained within a time window extending 0–1400 ms after each dot color change. We recorded whether the button response was correct (i.e., button 1 for brightness decrements, button 2 for brightness increments). We counted any case in which no button was pressed as an incorrect response. Percent correct was used to summarize performance.

For the one-back task, for each stimulus trial (excluding the first), we extracted the first button response obtained within a time window extending 250–4250 ms after the onset of the stimulus image. We then interpreted the responses as reflecting a simple signal detection experiment. Trials in which the image was identical to the previously presented image (one-back trials) were treated as 'signal present' trials. For these trials, button 2 ("old") corresponds to a hit and button 1

("new") corresponds to a miss. All other trials were treated as 'signal absent' trials. For these trials, button 2 ("old") corresponds to a false alarm and button 1 ("new") corresponds to a correct rejection. We counted any trial in which no button was pressed as an incorrect response (either a miss or a false alarm). Across trials, we computed the hit rate and the false alarm rate. We then converted these rates into the $d'$ sensitivity index based on the inverse cumulative distribution function of the standard normal distribution. In this calculation, to prevent numerical explosion, we set the minimum and maximum possible values for the hit rate and false alarm rate to be 0.01 and 0.99, respectively.

## Analysis of fMRI data

Below, we describe the specific analyses of NSD demonstrated in this paper. These analyses go beyond the data preparation as described above. For all analyses we used the pre-processed fMRI betas version 3 ('nsdsyntheticbetas_fithrf_GLMdenoise_RR' for NSD-synthetic and 'betas_fithrf_GLMdenoise_RR' for NSD-core), prepared in FreeSurfer's fsaverage space. (Beta version 3 reflects all GLM analysis components as described above.) For the analyses shown in this paper, we interpret responses based on the presented stimuli, aggregating across the tasks performed by the participant.

To reduce the effect of noise on results, for each participant, we utilized vertices with noise ceiling signal-to-noise ratio (NCSNR) scores above a threshold (except for the NCSNR visualization and for the encoding modeling, where we analyzed all available vertices). We quantified the NCSNR following the method proposed in NSD-core's data release paper[1], and selected 0.6 as the NCSNR threshold. We estimated the NCSNR independently for NSD-synthetic and NSD-core. Note that in the context of NSD-synthetic, we computed a single NCSNR estimate using fMRI responses from both tasks. Hence, our NCSNR estimate regards genuine task-related variability as noise and may therefore underestimate signals in the data.

**Univariate fMRI response analysis.** To compute NSD-synthetic's univariate fMRI responses, we sequentially averaged the fMRI responses across four dimensions: first across repeated trials (resulting in one trial-averaged response for each of the 284 NSD-synthetic stimulus images); second across vertices within each ROI (V1, V2, V3, hV4, PPA, VWFA); third across all images from the same subclass; and fourth across all participants. This resulted in one univariate response for each image subclass and ROI, indicating the extent to which a given ROI is activated by each image subclass.

**Representational similarity analysis.** Using representational similarity analysis (RSA)[41], we assessed the similarity of NSD-synthetic's multivariate fMRI responses between the ROIs of all participants (V1, V2, V3, hV4, PPA, VWFA). We began by averaging the multivariate fMRI responses across repeated trials, resulting in one trial-averaged response for each of the 284 NSD-synthetic stimulus images. Next, for each ROI and participant, we built representational similarity matrices (RSMs), symmetric matrices indicating the similarity (Pearson's $r$) of multivariate responses for each pair of stimulus images. Finally, for each pair of RSMs, we correlated (Pearson's $r$) the lower triangle of the two RSMs. This resulted in one RSA score for each combination of participant and ROI, indicating the similarity of visual information encoded in their multivariate fMRI responses.

**Multidimensional scaling.** To compare NSD-synthetic's and NSD-core's data distributions, we applied multidimensional scaling (MDS)[42,43] to single-trial fMRI responses aggregated across the two datasets (Fig. 3a). This MDS analysis determines a low-dimensional space that best approximates the similarity of responses between all stimulus image pairs. To improve stability of results, we appended

fMRI vertex responses across participants before applying MDS. Furthermore, to assess potential differences in distributions due to session-specific effects (e.g., incidental differences in overall fMRI response magnitude across sessions), we aggregated the 744 single-trial fMRI responses from NSD-synthetic to the first 744 (out of 750) single-trial fMRI responses from two NSD-core scan sessions (sessions 10 and 20). The MDS analysis ultimately produced a two-dimensional array of shape (2,232 trials × 2 embedding dimensions).

We also applied MDS on only NSD-synthetic's fMRI responses (without NSD-core) (Fig. 3b). To improve stability of results, for each of the 284 images, we averaged the fMRI responses across repeated trials and appended vertex responses across participants. This MDS analysis resulted in a two-dimensional array of shape (284 images × 2 embedding dimensions).

**Encoding models.** We trained and tested encoding models that linearly map stimulus image features onto fMRI responses. The image features consisted of the layer-wise activations of four deep neural networks, all trained on the ILSVRC-2012 image set[79]. The first deep neural network was AlexNet, a convolutional neural network pretrained on image classification[49], for which we used the activations from the output of the last sublayer of each of the eight model layers. The second deep neural network was vit_b_32, a vision transformer pretrained on image classification[50], for which we used the activations from the output of the last sublayer of each of the twelve model layers. The third deep neural network was ResNet50[51], a convolutional neural network with skip connections pretrained on task-supervised image classification, for which we used the activations from the output of the last sublayer of each of the four convolutional model layers. The fourth deep neural network was MoCo[52], consisting of the same ResNet50 architecture but trained with self-supervised contrastive learning, for which we used the activations from the output of the last sublayer of each of the four model layers.

We trained an independent encoding model using the image features of each deep neural network. The train data split consisted of NSD-core's 9000 participant-unique images and corresponding fMRI responses (i.e., the images uniquely seen by individual participants). During training, we first pre-processed each image by center-cropping it to square size using as size the image's smallest dimension; resized it to 224 × 224 pixels; applied a square-root transformation to the RGB values if necessary (we used NSD-core images as-is and applied a square root transformation to NSD-synthetic images to match the NSD-core images); scaled its values to the range [0, 1]; and normalized the scaled RGB values using mean = [0.485, 0.456, 0.406] and standard deviation = [0.229, 0.224, 0.225]. We then fed each of the pre-processed train images to one of the four deep neural networks described above, extracted the corresponding image features, and concatenated these features across layers. Next, to reduce computational costs while retaining the most important axes of variation, we reduced the dimensionality of the concatenated features to 250 principal components using principal components analysis (PCA)[48]. Finally, we trained independent linear regressions that mapped these PCA-reduced features onto the fMRI responses of each vertex in each participant.

We additionally trained encoding models independently for each deep neural network layer, using either linear or cross-validated ridge regression. In the case of linear regression, we reduced the dimensionality of the image features from each deep neural network layer to 250 principal components. In the case of ridge regression, we used the full image features of each deep neural network layer without dimensionality reduction through PCA.

We tested these encoding models both in-distribution (ID) and out-of-distribution (OOD). For ID, we tested models on 284 of the 515 NSD-core shared images that all participants saw three times. For OOD, we tested models on NSD-synthetic's 284 stimulus images. For both the ID and OOD testing images, we used the same procedure described above for obtaining PCA-reduced features, and then mapped these features onto fMRI responses using the trained encoding models weights. This generated predicted responses for ID and OOD testing images for all vertices and participants.

To assess the encoding models' ID and OOD generalization performance, we compared the predicted fMRI responses with their recorded (i.e., experimentally collected) counterparts, using Pearson's correlation. We correlated predicted and recorded fMRI responses independently for each vertex and participant, across the 284 ID or 284 OOD images, resulting in both ID and OOD correlation scores for each vertex and participant. These correlation scores reflect encoding model generalization performance, but are also affected by the level of noise in the recorded fMRI responses (which might not be equal between ID and OOD fMRI responses). Thus, to make the ID and OOD explained variance scores directly comparable, we normalized the ID and OOD generalization performance by the noise ceiling of the corresponding recorded fMRI responses. First, we squared all correlation scores after setting negative scores to zero, thus obtaining explained variance ($r^2$) for each vertex and participant. Second, following the method proposed in NSD-core's data release paper[1], we computed each vertex's noise ceiling signal-to-noise ratio (NCSNR), independently for NSD-synthetic and NSD-core's test fMRI responses. We converted these NCSNR scores into noise ceiling scores, indicating the amount of variance contributed by the signal expressed as a percentage of the total amount of variance in the data. These noise ceilings indicate the upper bound of model performance, given the noise in the data. Third, we divided the $r^2$ ID and OOD scores with the corresponding noise ceilings. This resulted in noise-ceiling-normalized explained variance scores that indicate the amount of explainable variance that is accounted for by the models, for each vertex and participant.

We applied the same encoding model procedure when controlling for session-specific effects (Supplementary Figs. 2 and 8), when testing the models on individual NSD-synthetic image classes (Fig. 6), and when testing the models on fMRI responses for both naturalistic and synthetic stimulus images (Fig. 7), with the only difference being the train and test data used.

**Zero-shot image identification.** As a complementary way of assessing the encoding models' generalization performance, we used the models' predicted fMRI responses to identify, in a zero-shot fashion, the stimulus images of the trial-averaged recorded test fMRI responses[2,44,55,56]. We started by correlating (Pearson's $r$) the vertex activity pattern of the recorded fMRI response for each test image, with the encoding-model-predicted vertex activity patterns for the candidate images (including for the target image of interest). For the recorded vertex activity pattern for each test image we then ranked, from highest to lowest, its correlation scores with the predicted vertex activity patterns for all candidate images. Finally, we stored the rank of the correct stimulus image (i.e., the position at which the vertex activity pattern of the recorded fMRI response is correlated with the vertex activity pattern of the predicted fMRI response for the same image). This resulted in one rank score for each test image, where lower ranks indicate a more robust identification of the recorded fMRI response stimulus images, using predicted fMRI responses from encoding models.

## Software
Data analyses were carried out in Python 3, using the following libraries (version in parenthesis): GLMsingle (v1.2); h5py (v3.1.0); Matplotlib (v3.9.4); NiBabel (v4.0.2); Nilearn (v0.9.2); Numpy (v1.26.4); nsdcode (v1.0.0); Pandas (v1.5.1); Pillow (v9.2.0); pycortex (v1.2.10); PyTorch (v1.13.0); scikit-learn (v1.1.1); SciPy (v1.12.0); Torchvision (v0.14.0); tqdm (v4.64.1).

fMRI data collection and preprocessing were carried out in MATLAB 2020a, using Psychtoolbox-3 FreeSurfer, and SPM12.

## Reporting summary

Further information on research design is available in the Nature Portfolio Reporting Summary linked to this article.

## Data availability

The NSD-synthetic dataset is freely available at http://naturalscenesdataset.org. We provide both raw data in BIDS format and prepared data files, along with extensive technical documentation in the NSD Data Manual (https://cvnlab.slite.page/p/CT9Fwl4_hc/NSD-Data-Manual).

## Code availability

We provide an archive of code used for creating the NSD dataset (https://github.com/cvnlab/nsddatapaper/). We also provide utility functions for working with the prepared NSD data (https://github.com/cvnlab/nsdcode/). Finally, we provide code to reproduce the analyses of the NSD data shown in this paper (https://github.com/gifale95/NSD-synthetic).

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

## Acknowledgements

We thank J. Bosten, B. Broderick, and A. White for consultation regarding stimulus design. We thank E. Allen and Y. Wu for collecting the neuroimaging data. We thank Martin Hebart for helpful discussions. We thank the HPC Service of FUB-IT, Freie Universität Berlin for computing time (https://doi.org/10.17169/refubium-26754). Collection of the NSD dataset was supported by NSF IIS-1822683 (K.K.) and NSF IIS-1822929 (T.N.). This work was supported by NIH grant R01EY034118 (K.K.), German Research Council (DFG) grants (CI 241/1-3, CI 241/1-7, INST 272/297-1) (R.M.C.), and European Research Council (ERC) consolidator grant (ERC-CoG-2024101123101) (R.M.C.).

## Author contributions

K.K. designed the experiment and pre-processed the data. A.T.G. modeled and analyzed the data. K.K. and A.T.G. prepared figures. A.T.G., K.K.,

T.N., and R.M.C. interpreted results. A.T.G. and K.K. wrote the manuscript. All authors discussed and edited the manuscript.

## Funding

## Competing interests
The authors declare no competing interests.
