## [Transparent Peer Review file · Nature Communications]

A 7T fMRI dataset of synthetic images for out-of-distribution modeling of vision

Corresponding Author: Dr Alessandro Gifford

Version 0:

Reviewer comments:

Reviewer #1

(Remarks to the Author)

This manuscript introduces a valuable fMRI dataset collected from subjects who also participated in the NSD(-core) project, with neural responses recorded for a carefully curated set of synthetic stimuli spanning eight distinct families (e.g., noise images, Mooney images, spiral gratings). The authors perform univariate and multivariate analyses and provide evidence of reliable neural responses—especially in early visual areas. They further position these stimuli as out-of-distribution relative to NSD-core and report a significant drop in encoding performance, suggesting the potential utility of this dataset for developing more robust vision models.

The release of this dataset and accompanying analysis code is a noteworthy contribution, particularly because it extends the widely used NSD-core dataset with a diverse set of synthetic stimuli collected from the same participants. The public availability of the data, along with a well-documented and user-friendly codebase, strongly supports reproducibility.

However, significant concerns exist regarding the analysis and interpretation of the collected data. Key aspects, such as the conceptualization of out-of-distribution (OOD) conditions, the control for potential experimental confounds, and the rigor of the analytical validation, need further attention to firmly establish the manuscript's conclusions. As a result, while the dataset is a valuable contribution, the current analyses may not yet fully substantiate claims of major advances significant to the field and the journal's focus.

Major comments

1

The most significant concern pertains to the treatment of out-of-distribution (OOD) stimuli in this study. In the Introduction (around line 45), the authors define in-distribution (ID) stimuli as those derived from the training distribution and OOD stimuli as those from a different distribution. I agree with this definition, but the study exclusively treats OOD as synthetic images and generalizes results from these synthetic stimuli as representative of OOD performance more broadly. This operationalization is problematic.

The implicit assumption would be that NSD-core sufficiently spans the natural image distribution. However, this assumption has been challenged in recent work. For example, Shirakawa et al. (2025, *_Neural Networks_*) pointed out that NSD-core exhibits limited diversity in natural images to just around 40 semantic categories, though it contains 30,000 brain samples per subject. Moreover, the same study demonstrated that failure of decoding OOD stimulus feature by redesigning training-test split of NSD-core. These results highlight that generalization to OOD is already quite challenging even within the domain of natural images, and that the notion of "OOD" should not be limited to synthetic stimuli alone.

In the present study, however, no effort has been made to evaluate model generalization to natural images beyond the training distribution. As a result, the observed drop in encoding performance for synthetic stimuli is difficult to interpret: it remains unclear whether the observed degradation is due to the stimuli being OOD, or additionally due to their synthetic images' nature.

To properly assess generalization to OOD stimuli, the authors should consider incorporating natural images that are semantically OOD with respect to the training data, not just synthetic images. Without a more rigorous treatment of the concept of OOD, the study does not yet provide the kind of clear and generalizable insight and falls short of establishing the broader significance of its findings to the field.

2

The encoding analysis in this paper aims to assess encoding performance differences caused by stimulus distributions (NSD-core vs NSD-synthetic). However, the data split designs between the two differ in an important way: NSD-core training and test data come from the same fMRI runs, while NSD-synthetic test data come from entirely separate sessions. Hence, the performance drop for NSD-synthetic may not solely reflect OOD difficulty, but also such confounding factors.

Although the manuscript briefly acknowledges this point in the Discussion (around L374), a more thorough examination of these confounds is essential to support the claim that NSD-synthetic is a useful benchmark for model and theory improvement. Additional analyses are required to quantify and mitigate the potential effects of session-specific variability. For example, the authors could perform an additional encoding analysis on NSD-core in which the model is trained using only runs that do not overlap with the test data—i.e., by explicitly excluding any training data from the same run as each test sample. This would allow a cleaner comparison focused solely on the effect of stimulus distribution, minimizing potential confounds due to run- or session-level dependencies. While implementing such an analysis may require additional effort, it would significantly reduce the influence of these confounds, and provide a clearer assessment of how stimulus distribution impacts performance.

Without such additional controls analysis, the observed performance gap between NSD-core and NSD-synthetic cannot be conclusively attributed to distributional differences, and thus the validity of the core claim remains in question. In its current form, the manuscript does not meet the rigor expected within this field.

3

In the encoding analysis, image features are derived from PCA-reduced vectors (top 250 components) based on DNN activations from multiple layers, extracted from NSD-core training images. It is possible that this PCA reduction can emphasize effective information in ID components while discarding low-level visual features that are potentially useful for predicting responses in early visual areas—particularly for synthetic stimuli that lack naturalistic structure.

To evaluate this possibility, the authors can analyze the explained variance ratio or PCA reconstruction error for both NSD-core test images and NSD-synthetic images. This would help determine whether the PCA disproportionately suppresses information from OOD stimuli. Additionally, running encoding analyses using features from either lower or higher DNN layers independently, rather than combining them, would be better to isolate the contributions of low- versus high-level features clearly.

Without such analysis, it remains uncertain whether the observed encoding performance—and comparisons across models—truly reflect differences in model generalization, or are instead confounded by limitations in the feature representation itself. In its current form, the work does not sufficiently demonstrate that the observed effects are attributable to the models rather than artifacts of feature construction. As it stands, the work lacks the evidential clarity required to support the key claims.

4

A further important concern is that, although the stated goal of this study is to advance the modeling of visual processing under out-of-distribution (OOD) conditions, the analyses rely solely on simple linear encoding models.

Recent work has increasingly raised concerns about the validity and interpretability of encoding analyses. For instance, Nonaka et al. reported that in feature-selection-based encoding models, only around 40% of model units contributed to prediction, suggesting that much of the model's representational richness may be ignored in the encoding analysis. Moreover, there is growing skepticism about the overreliance on encoding performance metrics alone, particularly in the context of benchmarks such as brain-score. Malhotra and Bowers (2024), for example, showed that encoding accuracy can sometimes depend more on image background than on object content, raising questions about what these models are actually learning to predict. In the current study, it also remains unclear what aspects of the image features are being used for prediction—whether they reflect ideal image content, semantic content, trivial properties such as average luminance, session-specific noise, or other unexamined factors. While the recent work by Schaeffer et al. (2024) has several problems, their claim about the fundamental concerns of frameworks that focus primarily on maximizing encoding scores should not be neglected.

Taking these recent discussions into account, this study that relies exclusively on standard encoding performance—without addressing potential confounding factors or validating the interpretability of the predicted responses—falls short of demonstrating meaningful progress in the field. The authors should consider conducting more multifaceted evaluations—such as testing whether the predicted responses can identify the presented stimuli—to better assess whether the predictions genuinely reflect stimulus related information.

Minor comments

5

The current OOD set seems to be heavily skewed toward spiral gratings (~40% of stimuli). As such, there is a possibility that aggregate performance in Figures 4 and 5 are dominated by this single stimulus type. To assess the encoding performance on OOD stimuli more clearly, I recommend reporting encoding performance separately by OOD category (e.g., spiral gratings, hue-manipulated scenes, words, line drawings, etc.). Even as supplementary material, such an analysis would help users understand whether performance drops generalize across stimulus types or are specific to particular OOD classes.

6

L 300: The phrase “ResNet-50 is a recurrent convolutional neural network” is inaccurate. ResNet-50 is not a recurrent model and should be rephrased.

7

L 524: The method for generating line-drawing stimuli is unclear. Compared to other stimulus types, the description feels incomplete. Please provide more detail on the algorithm or pipeline used.

Although the manuscript introduces a useful dataset, it does not currently meet the journal’s standards for presenting conceptually novel and methodologically rigorous work that offers significant advances for specialists in the field. For these reasons, I do not recommend this manuscript for publication in this journal.

****Reference:****

Malhotra, G. & Bowers, J. Predicting brain activation does not license conclusions regarding DNN-brain alignment: The case of Brain-Score. in *Cognitive Computational Neuroscience* (Boston, USA, 2024). [https://2024.ccneuro.org/pdf/309_Paper_authored_Neural_Predictivity_CCN2024.pdf] [https://2024.ccneuro.org/pdf/309_Paper_authored_Neural_Predictivity_CCN2024.pdf].

Nonaka, S., Majima, K., Aoki, S. C. & Kamitani, Y. Brain hierarchy score: Which deep neural networks are hierarchically brain-like? *iScience* ****24****, 103013 (2021).

Schaeffer, R. *et al.* Position: Maximizing Neural Regression Scores May Not Identify Good Models of the Brain. in *UniReps workshop* (Vancouver, Canada, 2024). [<https://openreview.net/pdf?id=vbtj05J68r>] [<https://openreview.net/pdf?id=vbtj05J68r>]

Shirakawa *et al.* Spurious reconstruction from brain activity. *Neural Networks* (in press) (2025). [<https://doi.org/10.48550/arXiv.2405.10078>] [<https://doi.org/10.48550/arXiv.2405.10078>]

(Remarks on code availability)

Reviewer #2

(Remarks to the Author)

(Remarks on code availability)

Reviewer #3

(Remarks to the Author)

A very valuable resource for the visual neuroscience community. The original Natural Scenes Dataset of 7T fMRI responses to over 9,000 scenes (Nature Neuroscience, 2021) is one of the largest and most used open datasets in visual neuroscience (557 citations). The current paper announces and describes a supplement to this dataset, which seems to have been collected around the same time, but only released in March of this year. The supplement consists of 7T fMRI responses from the same individual subjects, to an additional 284 synthetic images (either manipulated, e.g. via contrast reduction or phase scrambling; or artificially generated e.g. colour fields or spatial-frequency-controlled spiral patterns). Although the new data supplement is much smaller than the original core dataset, it will be useful both for testing encoding models outside the distribution of natural images (the main purpose advocated by the authors) and for probing the cortical representation of colour, orientation, and spatial frequency. The stimuli in the dataset are probably not as exhaustively parameterised for this second purpose as a researcher would desire (e.g. colour hue space is sampled, but not the saturation and lightness axes), but the dataset will nevertheless be helpful for some carefully-chosen research questions (as well as serving as the intended out-of-distribution test set).

The paper is very clearly written and figures and methods are excellently presented. The analyses of the paper go beyond the "minimum required" to describe and quality-check the data, and also evaluate a number of deep neural network based encoding models.

MINOR COMMENTS

- line 46 "intelligence" seems like an odd way to characterise the research that would be done with this dataset. Why not "visual neuroscience" or similar?
- Figure 1B "Manipulates scenes" should read "Manipulated scenes"
- lines 96-101: It would be good to already acknowledge in the main text two important differences between the NSD-core and NSD-synthetic datasets - the temporal design (2s vs 3s on), and the task (mixture of fixation and same/different tasks, vs continuous old/new task). These differences do limit the comparability of the main and supplemental dataset and should be acknowledged early.
- line 128: It would be helpful to mention how the noise ceilings measured in this dataset compare to those in the NSD-core dataset.
- line 300: ResNet is a feedforward architecture with skip connections, in my understanding, not a recurrent network (i.e. it has no temporal dynamics or memory over stimuli).

(Remarks on code availability)

I have skimmed but not attempted to use any of the code. It appears well documented and thorough. Code is provided to reproduce all figures in the paper, as well as sharing the images and fMRI data.

Reviewer #4

(Remarks to the Author)

The authors present NSD-synthetic, a companion dataset to the Natural Scenes Dataset (NSD), consisting of 7T fMRI responses from 8 subjects viewing 284 synthetic images. This dataset is designed to enable out-of-distribution (OOD) generalization tests for computational models of visual processing. The authors demonstrate that neural responses to these synthetic stimuli are reliably encoded and are indeed OOD relative to NSD-core. They also present encoding model comparisons showing differences between self-supervised and task-supervised models in OOD settings, limited to early/mid visual cortex where the synthetic stimuli drive robust responses.

Strengths

This is an exceptionally valuable resource for the computational neuroscience community. The dataset addresses a critical gap in the field - namely, the ability to test whether models of visual processing generalize beyond the natural image distributions they are typically trained on. The experimental design is rigorous, using the same 8 subjects from NSD-core and maintaining high data quality standards with 7T fMRI at 1.8mm resolution. The open release of this data will enable researchers to develop more robust models and better understand the limitations of current approaches.

The technical quality is, as always from Ale, excellent. Comprehensive validation shows that the synthetic stimuli reliably drive neural responses, particularly in early visual areas. The use of multidimensional scaling to confirm the OOD nature of the responses is convincing, and the variety of synthetic stimuli (noise patterns, manipulated scenes, gratings, words) provides good coverage of different visual features.

Of particular interest, but also a potential confound, is the choice of artificial stimuli. As deep neural networks are (often) surprisingly bad using these type of stimuli. Using stimuli like spiral gratings, phase-scrambled images, etc. etc., the dataset probes a known weakness of current computational models. This makes NSD-synthetic an excellent diagnostic tool for understanding where and why DNNs fail to capture human visual processing.

Major/Medium point

* The authors show that self-supervised models (MoCo) outperform task-supervised models (ResNet-50) in OOD generalization, and that vision transformers outperform CNNs. For early visual cortex (V1-V4), where the synthetic stimuli drive robust responses, these findings appear valid and represent an important discovery about differential model architectures. The high noise ceiling values in these areas (Figure 2A) support meaningful model comparisons. We do not really know what happens in other parts of cortex, or if the models would generalise better towards OOD stimuli for areas in higher cortex (e.g. OOD stimuli that the cortex would respond to). Not sure what to do about this, you could look at OOD effects vs SNR effects. It could also really be that they are restricted to early cortex. But in any case I think it is important to reframe the conclusion a little bit to reflect that the OOD generalization findings are strongest for early visual cortex, with important caveats about interpretability in higher areas.

* The paper lacks discussion of why this might occur. and why this occurs for early cortex. Is the dictionary of these networks just better? Or because contrastive learning explicitly trains models to be invariant to various image transformations, and that this generalises towards these artificial stimuli?

In this cortex it might be an idea to look at layers in MOCO in relation to effects?

Some mechanistic insight would really help.

(Remarks on code availability)

I perused the code. It seems complete and is intelligible.

Version 1:

Reviewer comments:

Reviewer #1

(Remarks to the Author)

The authors have responded sincerely to the previous comments, and through extensive additional analyses and clarifications, the manuscript has clearly improved, addressing many of the initial concerns. At the same time, I think some of the new analyses and findings would benefit from more careful consideration and interpretation. I highly value the contribution of this dataset and recognize its significant potential impact on the field. Precisely for this reason, I believe that ensuring the analyses and interpretations are handled as rigorously as possible will further strengthen the manuscript.

[Related to my previous comments "1. The most significant concern ..."]

My first concern in the previous review was the treatment of OOD. In response, the authors redefined the training–test split in Figure 7 and prepared a naturalistic OOD test condition, which has clearly improved the manuscript. However, it remains somewhat unclear why the naturalistic OOD condition was prepared based on fMRI activity patterns rather than on stimuli. In both encoding and decoding approaches, it is generally more common to construct OOD conditions based on stimuli (Mitchell et al., 2008; Brouwer and Heeger, 2009; Madan et al., 2024; Shirakawa et al., 2025).

The manuscript would further benefit from considering reanalyses that use stimulus-based splits when preparing the naturalistic OOD test condition. The semantic feature–based data split in Shirakawa et al. (2025) could serve as an appropriate direction. Moreover, given the potential influence of session/run confounds (discussed later), it also strengthens the conclusions if the training–test split in Figure 7 further excluded session-related overlap.

Additionally, the motivation for the analysis in Figure 7 could be explained more clearly. In the Introduction, the statement that "(L76–77) the concept of OOD can be usefully applied even within the domain of naturalistic stimuli (Fig. 7)" feels somewhat abrupt in relation to the preceding paragraph (L45–51). Referring to Shirakawa et al. (2025) and their concerns about the diversity of NSD stimuli, as well as their decoding analyses, could help clarify the rationale. Finally, the procedure described in L450–L462 (low-dimensional projection, k-means clustering, and creating split) appears very similar to the procedure in Figure 6a of Shirakawa et al. (2025), and citing that paper here would strengthen the context.

[Related to my previous comments "2. The encoding analysis in this paper..."]

The second concern in my previous review was the potential confound of using the same runs/sessions for both training and test data. The authors addressed this with additional analyses such as Supplementary Figure 2, which control for session (and run) confounds by separating the last five sessions for testing. This addition is valuable and reassures me that the authors have considered the issue carefully.

One important point that could be emphasized more clearly, however, is that ID generalization drops substantially once session confounds are removed (with confounds: 45.14% in Fig. 4c; without confounds: 38.64% in Sup. Fig. 2a). Since OOD performance decreases only minimally (with confounds: 10.12%; without confounds: 9.94%), this suggests that session confounds strongly affect encoding performance.

It may strengthen the manuscript if the default encoding analyses (Fig. 4 ~ 7) are presented in a version that excludes session confounds, allowing for a clearer comparison focused on effects of interest. Explicitly acknowledging the impact of session confounds in the main text would also help readers better interpret the results. Highlighting this point would provide value not only for this dataset but also for future large-scale data collection paradigms and open-data splitting strategies.

[Related to my previous comments "3. In the encoding analysis,..."]

The third concern in my previous review was that including PCA might emphasize ID-relevant information while discarding OOD-relevant features. The new analyses in Supplementary Figures 3 and 4, comparing linear regression with PCA against ridge regression without PCA across layers, are a meaningful revision.

While this is a valuable addition and certainly reduces my concern, I believe the results reveal something more important

than is currently mentioned in the main text (L285). Specifically, in many layers, OOD prediction performance under ridge regression (without PCA) exceeds that under both linear regression (with PCA) and the main results (Fig. 4), a pattern not observed for ID performance. This implies that applying PCA to training features may not be the optimal approach for OOD modeling, as I pointed out in the previous review.

Explicitly acknowledging this point in the main text could further strengthen the paper, especially given its central focus on this paper is out-of-distribution modeling.

****[Related to my previous comments “4. A further important concern...”]****

The fourth concern in my previous review was that that OOD performance was evaluated only with encoding analyses. The authors addressed this with image-identification analyses, which is a valuable addition. However, interpretation remains somewhat complicated by (i) class imbalance candidate set in Fig. 6d (e.g., 112 spiral gratings vs. 68 chromatic noise vs. 40 word stimuli vs. 8–16 stimuli in other classes), and (ii) different candidate sets across conditions in Fig. 7c.

It may be helpful to consider using balanced candidate sets (e.g., N images per class) to eliminate the possibility that imbalance drives differences in Fig. 6d. Similarly, for Fig. 7c, using a common candidate set across conditions (e.g., pooling all test images, or drawing balanced samples from ID and OOD splits, or randomly selecting N training images) could make the interpretation clearer. Even if these stricter analyses do not substantially change the results, their inclusion would demonstrate rigor and increase the impact and value of this paper.

****[Related to my previous comments “5. The current OOD set seems...”]****

The fifth point was in my previous review to see the performance differences across each OOD category. In the revised manuscript, Figure 6 presents these results, and the Discussion briefly notes them (L508–L513). This is an important revision, though I believe it could be highlighted more concretely. For example, although aggregate NSD-synthetic performance (Supplementary Fig. 5 / Fig. 4) appears low, class-level performance varies substantially (40.18% for manipulated scenes vs. 13.37% for spiral gratings).

It would also be helpful to explain why the noise class shows relatively high encoding performance (29.25%), even exceeding that of some OOD classes (e.g., chromatic noise, spiral gratings). An additional exploration of which aspects of stimulus-related information are being actually captured by the encoding model could provide useful insight. In this context, rather than focusing solely on encoding performance, a more multifaceted analysis (e.g., preparing appropriate candidate sets for the image-identification analysis) could offer a promising direction.

****Minor concerns****

- In the updated Abstract, the authors write, “(L16) However, these datasets [NSD] lack out-of-distribution (OOD) components ... Here, we address this limitation by releasing NSD-synthetic ...,” yet later they state, “(L27) the concept of OOD is not restricted to artificial stimuli but can be usefully applied even within the domain of naturalistic stimuli.” These statements may appear contradictory; a brief clarification would help.
- At L389, the authors state that the degree of OOD is measured by Euclidean distance between training and test fMRI responses, but in practice it is computed in MDS space (L392–396). Modifying this inconsistency would improve precision.
- At L545, the phrase “across this continuum” is unclear. The results suggest discrete stages of OOD generalization rather than a continuous spectrum; clarification would be helpful.
- At L292, “Fig. 7” should be corrected to “Fig. 5.”
- At L535, the phrase “~ across the visual areas.” might be better written as “~ across the visual areas.”

Overall, this revised version is a substantial improvement over the previous manuscript. I appreciate the authors’ efforts in addressing the concerns raised in the first round of review and for performing extensive additional analyses. With the further clarifications and refinements suggested above, I believe this work will provide a particularly valuable contribution to the community and will have significant impact in advancing research on large-scale visual neural datasets.

****[Reference]****

Mitchell, T. M., Shinkareva, S. V., Carlson, A., Chang, K.-M., Malave, V. L., Mason, R. A., & Just, M. A. (2008). Predicting Human Brain Activity Associated with the Meanings of Nouns. *Science*, *320*(5880), 1191–1195. <https://doi.org/10.1126/science.1152876>

Brouwer, G. J., & Heeger, D. J. (2009). Decoding and Reconstructing Color from Responses in Human Visual Cortex. *Journal of Neuroscience*, *29*(44), 13992–14003. <https://doi.org/10.1523/JNEUROSCI.3577-09.2009>

Shirakawa, K., Nagano, Y., Tanaka, M., Aoki, S. C., Muraki, Y., Majima, K., & Kamitani, Y. (2025). Spurious reconstruction from brain activity. *Neural Networks*, *190*, 107515. <https://doi.org/10.1016/j.neunet.2025.107515>

Madan, S., Xiao, W., & Cao, M. (2024). Benchmarking Out-of-Distribution Generalization Capabilities of DNN-based Encoding Models for the Ventral Visual Cortex. *NeurIPS*, *2832*, 89249–89277. <https://dl.acm.org/doi/10.5555/3737916.3740748>

(Remarks on code availability)

Reviewer #2

(Remarks to the Author)

(Remarks on code availability)

I have just skimmed but not attempted to run. It appears to have well-documented and readable code. Those well-structured codes will be useful for reproducing the results.

Reviewer #3

(Remarks to the Author)

The authors have amply addressed my already minor comments. They have also gone above and beyond in their responses to other reviewers' comments, substantially extending the analyses in the paper. It's good to see that the new analyses and framing treat being "out of distribution" as a continuum, and as something separable from whether an image is natural/artificial. I continue to find the paper a valuable contribution to datasets and model evaluations in visual neuroscience.

(Remarks on code availability)

I haven't tried to run the code, but have read the ReadMe and skimmed through the code files. They appear well-organised and well-documented, and to provide everything needed to reproduce analyses underlying each figure in the paper.

Reviewer #4

(Remarks to the Author)

The authors have replied extensively to mine, but also the concerns of the other reviewers. I in particular appreciate the plimitation expressed in Discussion, page 25, lines 628-635 and the clear ideas expressed in lines 514-528.

I have no further comments.

(Remarks on code availability)

The code seems to be complete and also updated to the last version. I have sampled some files. Looks good.

Version 2:

Reviewer comments:

Reviewer #1

(Remarks to the Author)

The authors have responded sincerely to the previous comments, and through clarifications additional analyses, the manuscript has surely improved. I have only one remaining concern regarding the expression "less strict OOD generalization tests" used to describe the naturalistic OOD split. As currently written, this phrasing may unintentionally undervalue the potential of NSD-core.

I understood that preparing OOD test splits that additionally control for session-specific effects is challenging in the present analysis. Considering that the main purpose of this study is a proof-of-principle demonstration using NSD-synthetic, I will not request further additional analyses. However, I would encourage the authors to refine the manuscript by more clearly acknowledging the limitations of the current naturalistic OOD split and by avoiding statements that might be interpreted as fundamental limitations of NSD-core itself.

In particular, referring to the NSD-core-based OOD evaluation as a "less strict OOD generalization test" could be read as implying that NSD-core inherently permits only weak OOD tests. In practice, with more carefully designed train-test splits, stricter naturalistic OOD evaluations are possible even within NSD-core. For example, following the conventional stimulus-based OOD splitting approaches (Mitchell et al., 2008; Brouwer & Heeger, 2009; Madan et al., 2024; Shirakawa et al., 2025), we could construct a naturalistic OOD test set consisting of a specific semantic category (e.g., "zebra" images) presented only in the last few sessions. The training set could then be built from earlier sessions while excluding all zebra-containing stimuli. This would allow direct comparison between naturalistic OOD tests (inherent to NSD-core) and synthetic OOD tests (inherent to NSD-synthetic) while simultaneously removing session confounds existing in the current naturalistic OOD.

A similar logic can also be applied to the fMRI-based splits. For instance, given the distribution shown in Revision Fig. 1, we could design the red samples at the right side as a test set and construct the training set from the remaining black points

while excluding the corresponding region of the MDS space.

I am not requesting that the authors implement such additional analyses. Rather, I suggest that the manuscript be revised to (i) clarify that the “less strict” nature of the current NSD-core OOD tests reflects the specific data split used in this study, not a fundamental limitation of the dataset itself, and (ii) acknowledge more explicitly that stricter OOD tests within NSD-core are, in principle, possible with suitably designed splits.

Specifically, the following revised sections may merit careful reconsideration:

L20-L22:

“because the stimuli of these datasets typically live within a common naturalistic visual distribution, they do not allow for strict OOD generalization tests”

→ Suggest clarifying that strict tests are possible but require more deliberate split design.

L31-33

“less strict OOD generalization tests can be usefully applied even within the domain of naturalistic stimuli.”

L555-556

“although resulting in less strict OOD generalization tests due to the lower degree of OOD of fMRI responses for naturalistic compared to artificial stimuli.”

L633- L635

“from lower degrees of OOD and thus less strict generalization tests using fMRI responses for NSD-core’s naturalistic stimuli, to higher degrees of OOD and thus stricter tests using NSD-synthetic”

This revisions have considerably strengthened the manuscript. With the minor clarifications suggested above, I am confident that the manuscript will be even clearer and more informative, and that it will serve as a valuable resource for future studies on large-scale visual neural datasets.

(Remarks on code availability)

Reviewer #2

(Remarks to the Author)

(Remarks on code availability)

Dear Reviewers,

Please find the detailed point-by-point response below. Reviewer comments are highlighted in gray, and the corresponding responses indented. Within the responses, extracts from the manuscript are “*quoted in italics*”, and extracts reflecting changes to the manuscript are “*quoted in italics and red font*”. Manuscript line/page/figure numbers are indicated in **bold font**. Blue underscored text denotes hyperlinks.

Reviewer #1

Remarks to the Author

This manuscript introduces a valuable fMRI dataset collected from subjects who also participated in the NSD(-core) project, with neural responses recorded for a carefully curated set of synthetic stimuli spanning eight distinct families (e.g., noise images, Mooney images, spiral gratings). The authors perform univariate and multivariate analyses and provide evidence of reliable neural responses—especially in early visual areas. They further position these stimuli as out-of-distribution relative to NSD-core and report a significant drop in encoding performance, suggesting the potential utility of this dataset for developing more robust vision models.

The release of this dataset and accompanying analysis code is a noteworthy contribution, particularly because it extends the widely used NSD-core dataset with a diverse set of synthetic stimuli collected from the same participants. The public availability of the data, along with a well-documented and user-friendly codebase, strongly supports reproducibility.

However, significant concerns exist regarding the analysis and interpretation of the collected data. Key aspects, such as the conceptualization of out-of-distribution (OOD) conditions, the control for potential experimental confounds, and the rigor of the analytical validation, need further attention to firmly establish the manuscript's conclusions. As a result, while the dataset is a valuable contribution, the current analyses may not yet fully substantiate claims of major advances significant to the field and the journal's focus.

We thank Reviewer #1 for the positive evaluation and helpful comments, which resulted in valuable additional analyses and discussions. In brief, we:

(1) Added a new Results section and a Discussion paragraph on OOD generalization tests using fMRI responses for both naturalistic and synthetic images.

(2) Added a new Results section and a Discussion paragraph on the OOD generalization performance for individual NSD-synthetic image classes.

(3) Added new analyses and a Discussion paragraph related to the interpretability of OOD generalisation tests using the encoding modeling approach.

(4) Added several control analyses to ensure that our encoding results are not a spurious product of session-specific noise, nor of the type of image features used.

(5) Added clarifications as requested throughout the

manuscript. **Major Comments:**

1. The most significant concern pertains to the treatment of out-of-distribution (OOD) stimuli in this study. In the Introduction (around line 45), the authors define in-distribution (ID) stimuli as those derived from the training distribution and OOD stimuli as those from a different distribution. I agree with this definition, but the study exclusively treats OOD as synthetic images and generalizes results from these synthetic stimuli as representative of OOD performance more broadly. This operationalization is problematic.

The implicit assumption would be that NSD-core sufficiently spans the natural image distribution. However, this assumption has been challenged in recent work. For example, Shirakawa et al. (2025, *_Neural Networks_*) pointed out that NSD-core exhibits limited diversity in natural images to just around 40 semantic categories, though it contains 30,000 brain samples per subject. Moreover, the same study demonstrated that failure of decoding OOD stimulus feature by redesigning training-test split of NSD-core. These results highlight that generalization to OOD is already quite challenging even within the domain of natural images, and that the notion of “OOD” should not be limited to synthetic stimuli alone.

In the present study, however, no effort has been made to evaluate model generalization to natural images beyond the training distribution. As a result, the observed drop in encoding performance for synthetic stimuli is difficult to interpret: it remains unclear whether the observed degradation is due to the stimuli being OOD, or additionally due to their synthetic images' nature.

To properly assess generalization to OOD stimuli, the authors should consider incorporating natural images that are semantically OOD with respect to the training data, not just synthetic images. Without a more rigorous treatment of the concept of OOD, the study does not yet provide the kind of clear and generalizable insight and falls short of establishing the broader significance of its findings to the field.

We thank the Reviewer for the suggestion, which led to a new Results section where we tested the OOD generalization of encoding models for both naturalistic (from NSD-core) and synthetic (from NSD-synthetic) images. We obtained highest encoding accuracy scores for ID testing on NSD-core, followed by lower accuracy scores for

naturalistic OOD testing on NSD-core, and finally by synthetic OOD testing on NSD-synthetic. Together, this new analysis showcases the potential of combining NSD-core and NSD-synthetic to test the OOD robustness of brain models on neural responses for both naturalistic and synthetic stimuli, and that the concept of OOD is not restricted to artificial stimuli but can be usefully applied even within the domain of naturalistic stimuli (**Results (Fig. 7), page 20, lines 423-477**):

“NSD enables out-of-distribution tests of brain models across both naturalistic and synthetic stimulus images

Fig. 7 | NSD enables out-of-distribution tests of brain models across both naturalistic and synthetic stimulus images. We trained vision-transformer-based encoding models on NSD-core, and tested their generalization across three conditions: ID on NSD-core, naturalistic OOD on NSD-core, and synthetic OOD on NSD-synthetic. **a**, MDS embedding of trial-averaged fMRI responses (aggregated across participants) for the train images (gray dots), for the ID test images (orange dots), for the naturalistic OOD test images (yellow dots), and for the synthetic OOD test images (turquoise dots). **b**, Encoding models’ explained variance for the three test conditions, normalized by the noise ceiling. This figure shows results averaged across participants on flattened cortical surfaces. To reduce false positives, for each vertex we averaged results from only those participants with a noise ceiling greater than 0.3 for both NSD-core and NSD-synthetic (vertices for which no participant has a noise ceiling above 0.3 are not shown). **c**, Encoding models’ zero-shot identification scores of the recorded fMRI response stimulus images, for each of the three test conditions, on violin plots. The y-axis represents the correct image rank, that is, the position at which the correct image appears among the models’ choices (choosing among all 284 images of the corresponding test split), with lower ranks indicating a more robust image identification. The colored violin shapes indicate the distribution of observations, with wider sections corresponding to higher density of values. Small transparent dots indicate the correct image rank for single images and participants. Large black dots indicate the average correct image rank across all participants and images within each test condition, with the corresponding score reported above the dot. Horizontal dashed lines indicate chance identification scores.

Up to here we implemented OOD generalization tests using NSD-synthetic. To highlight that OOD tests need not be restricted to neural responses for synthetic stimuli, in this last analysis we assess the OOD generalization performance of vision-transformer-based encoding models to fMRI responses for both naturalistic and synthetic images.

*We first used a combination of MDS and clustering algorithms to divide NSD-core's fMRI responses into train, ID test, and naturalistic OOD test splits. For each participant, we reduced the trial-averaged fMRI responses for NSD-core's images to two dimensions through MDS, and partitioned this two-dimensional embedding space into 15 clusters using k-means clustering. For the naturalistic OOD test split we considered the fMRI responses from the cluster with highest Euclidean distance from the centroids of the remaining 14 clusters and, within it, selected fMRI responses for the 284 images with the highest Euclidean distance from the remaining clusters. For the ID test split we selected fMRI responses for 284 images randomly selected from the remaining 14 clusters. As the train split we selected the images from the remaining 9 clusters not belonging to the ID test split. These procedures ensured that the naturalistic OOD test responses are maximally distant in MDS embedding space from the distribution of the train responses, and that the train and ID test responses come from the same distribution (**Fig. 7a**). We then trained encoding models using the train split and evaluated them ID (on NSD-core), as well as on naturalistic (on NSD-core) and synthetic (on NSD-synthetic) OOD test fMRI responses.*

*We obtained higher noise-ceiling-normalized encoding accuracy scores for ID testing on NSD-core, followed by naturalistic OOD testing on NSD-core, and finally by synthetic OOD testing on NSD-synthetic (**Fig. 7b**). We obtained the same ranking when using the encoding models' predicted fMRI responses to identify, in a zero-shot fashion, the stimulus images of the ID, naturalistic OOD, and synthetic OOD test fMRI responses (choosing among all images of the corresponding test split) (**Fig. 7c**). This ranking was expected, as it reflects the fact that the ID, naturalistic OOD, and synthetic OOD test fMRI responses are increasingly distant from the train responses in MDS space (**Fig. 7a**). We obtained quantitatively similar results when using encoding models based on other deep neural network architectures (**Supplementary Figs. 16-18**), indicating that these findings are not contingent on a specific choice of deep neural network.*

In conclusion, these results showcase the strength of combining NSD-core and NSD-synthetic to test the OOD robustness of brain models on neural responses. The results also demonstrate that the concept of OOD is not restricted to artificial stimuli but can be usefully applied even within the domain of naturalistic stimuli."

Furthermore, to reflect that OOD generalization tests can involve both naturalistic and synthetic stimulus images, we also added the following paragraph to the Discussion (**Discussion, page 23, lines 537-549**):

*"To obtain valid inferences, the details, quantifications, and interpretation of OOD tests require careful consideration. Here, we comment on **four** specific aspects that researchers should consider when performing OOD generalization tests.*

First, there is a continuum of degrees to which the test data can be OOD with respect to the train data, which in turn affects the generalization performance of computational models of the brain. We showed this for test fMRI responses for different NSD-synthetic image classes (Fig. 6), as well as for test fMRI responses for naturalistic and synthetic images (Fig. 7). Our empirical results reinforce the fact that OOD generalization performance of brain models cannot be fully described by a simple summary number, but lives in a continuum. Combining NSD-core and NSD-synthetic provides a way to test several degrees of OOD generalization across this continuum, using fMRI responses for both naturalistic and synthetic stimuli. We envision that this will endow researchers with greater flexibility in model testing, resulting in a more nuanced account of the OOD generalization performance of brain models.”

Finally, we present the new Method section relative to the zero-shot identification analysis.

(Methods, page 35, lines 1026-1037): *“Zero-shot image identification. As a complementary way of assessing the encoding models’ generalization performance, we used the models’ predicted fMRI responses to identify, in a zero-shot fashion, the stimulus images of the trial-averaged recorded test fMRI responses (Kay et al., 2008; Horikawa & Kamitani, 2017; Seeliger et al., 2018). We started by correlating (Pearson’s r) the vertex activity pattern of the recorded fMRI response for each test image, with the encoding-model-predicted vertex activity patterns for all test images. For the recorded vertex activity pattern for each test image we then ranked, from highest to lowest, its correlation scores with the predicted vertex activity patterns for all test images. Finally, we stored the rank of the correct stimulus image (i.e., the position at which the vertex activity pattern of the recorded fMRI response is correlated with the vertex activity pattern of the predicted fMRI response for the same image). This resulted in one rank score for each test image, where lower ranks indicate a more robust identification of the recorded fMRI response stimulus images, using predicted fMRI responses from encoding models.”*

2. The encoding analysis in this paper aims to assess encoding performance differences caused by stimulus distributions (NSD-core vs NSD-synthetic). However, the data split designs between the two differ in an important way: NSD-core training and test data come from the same fMRI runs, while NSD-synthetic test data come from entirely separate sessions. Hence, the performance drop for NSD-synthetic may not solely reflect OOD difficulty, but also such confounding factors.

Although the manuscript briefly acknowledges this point in the Discussion (around L374), a more thorough examination of these confounds is essential to support the claim that NSD-synthetic is a useful benchmark for model and theory improvement. Additional analyses are required to quantify and mitigate the potential effects of session-specific variability. For example, the authors could perform an additional encoding analysis on NSD-core in which the model is trained using only runs that do not overlap with the test data—i.e., by explicitly excluding any training data from the same run as each test sample. This would allow a cleaner comparison focused solely on the effect of stimulus distribution,

minimizing potential confounds due to run- or session-level dependencies. While implementing such an analysis may require additional effort, it would significantly reduce the influence of these confounds, and provide a clearer assessment of how stimulus distribution impacts performance.

Without such additional controls analysis, the observed performance gap between NSD-core and NSD-synthetic cannot be conclusively attributed to distributional differences, and thus the validity of the core claim remains in question. In its current form, the manuscript does not meet the rigor expected within this field.

We fully agree that session-specific effects might artificially cause reduced performance for models generalizing to the scan session with synthetic stimuli responses. To control for session-specific effects in our encoding analyses, as suggested by the Reviewer, we trained and tested encoding models using fMRI responses from non-overlapping NSD-core scan sessions. This did not change our findings, namely that out-of-distribution (OOD) tests on NSD-synthetic result in lower generalization performances than in-distribution (ID) tests on NSD-core (**Fig. 4**), and that NSD-synthetic reveals differences between models that are not detected from ID tests (**Fig. 5**). Therefore, this excludes session-specific effects as a confounding factor in our analyses.

(Results (Fig. 4), page 13, lines 276-285): *“To ensure that the lower OOD performances are meaningful and robust, we performed three controls. First, to ensure that the lower performance on NSD-synthetic is not simply due to the fact that it was collected in a separate scan session, we trained and tested the encoding models using fMRI responses from separate NSD-core scan sessions (Supplementary Fig. 2). Second, to ensure that the specific regression method that one chooses to use does not change our conclusions, we trained and tested the encoding models on image features from individual AlexNet layers using linear or ridge regression (Supplementary Figs. 3-4). Third, to ensure that our results are not contingent on a specific choice of deep neural network, we trained and tested the encoding models on image features from deep neural networks other than AlexNet (Supplementary Figs. 5-7). In all three cases, we observed lower OOD performances for NSD-synthetic than for NSD-core.”*

“Supplementary Fig. 2 | In-distribution and out-of-distribution generalization comparison of AlexNet-based encoding models, controlling for session-specific effects. To control for session-specific effects, we trained and tested AlexNet-based encoding models using fMRI responses from separate NSD-core scan sessions. For each participant, we trained the encoding models using image features and fMRI responses from all but the last 5 NSD-core scan sessions. We then tested these models’ in distribution (ID) generalization on 284 image conditions that were only presented during NSD-core’s last 5 scan sessions, and their out-of-distribution (OOD) generalization on the 284 image conditions from NSD-synthetic. This figure shows results averaged across participants on flattened cortical surfaces. **a**, Model explained variance normalized by the noise ceiling. To reduce false positives, for each vertex we averaged results from only those participants with a noise ceiling greater than 0.3 (vertices for which no participant has a noise ceiling above 0.3 are not shown). **b**, Difference between ID and OOD noise-ceiling-normalized explained variance. For each vertex, we averaged results from only those participants with noise ceiling greater than 0.3 for both NSD-core and NSD-synthetic. The asterisk indicates a significant difference between the ID and OOD noise-ceiling-normalized explained variance scores ($P = 10^{-7}$, paired sample t-test, one-sided, $N = 8$ participants). **c**, Scatterplots of vertex-wise explained variance scores against their noise ceilings for ID (NSD-core) and OOD (NSD-synthetic) generalization tests, color coded by the visual stream that each vertex belongs to. For each vertex, we plotted results from only those participants with noise ceiling greater than 0.3 for both NSD-core and NSD-synthetic (same as in panel **b**). Black crosses indicate median scores across all vertices.”

(Results (Fig. 5), page 15, lines 335-342): “To ensure that the OOD differences not detected by ID tests are meaningful and robust, we performed two controls. First, to ensure that our results are not simply due to the fact that NSD-synthetic was collected in a separate scan session, we trained and tested the encoding models using fMRI responses from separate NSD-core scan sessions (**Supplementary Fig. 8**). Second,

to ensure that the specific regression method that one chooses to use does not change our conclusions, we trained and tested the encoding models on image features from individual ResNet50 or MoCo layers using either linear or ridge regressions (**Supplementary Figs. 9-12**). In both cases, OOD tests revealed differences between models not detected by ID tests.”

“Supplementary Fig. 8 | In-distribution and out-of-distribution generalization differences between encoding models, controlling for session-specific effects. To control for session-specific effects, we trained and tested encoding models using fMRI responses from separate NSD-core scan sessions. For each participant, we trained the encoding models using image features and fMRI responses from all but the last 5 NSD-core scan sessions. We then tested these models’ in distribution (ID) generalization on 284 image conditions that were only presented during NSD-core’s last 5 scan sessions, and their out-of-distribution (OOD) generalization on the 284 image conditions from NSD-synthetic. This figure shows results averaged across participants on flattened cortical surfaces. To reduce false positives, for each vertex we averaged results from only those participants with a noise ceiling greater than 0.3 (vertices for which no participant has a noise ceiling above 0.3 are not shown). **a**, Difference between AlexNet and vit_b_32’s noise-ceiling-normalized explained variance. **b**, Difference between MoCo and ResNet50’s noise-ceiling-normalized explained variance.”

3. In the encoding analysis, image features are derived from PCA-reduced vectors (top 250 components) based on DNN activations from multiple layers, extracted from NSD-core training images. It is possible that this PCA reduction can emphasize effective information in ID components while discarding low-level visual features that are potentially useful for predicting responses in early visual areas—particularly for synthetic stimuli that lack naturalistic structure.

To evaluate this possibility, the authors can analyze the explained variance ratio or PCA reconstruction error for both NSD-core test images and NSD-synthetic images. This would help determine whether the PCA disproportionately suppresses information from OOD

stimuli. Additionally, running encoding analyses using features from either lower or higher DNN layers independently, rather than combining them, would be better to isolate the contributions of low- versus high-level features clearly.

Without such analysis, it remains uncertain whether the observed encoding performance—and comparisons across models—truly reflect differences in model generalization, or are instead confounded by limitations in the feature representation itself. In its current form, the work does not sufficiently demonstrate that the observed effects are attributable to the models rather than artifacts of feature construction. As it stands, the work lacks the evidential clarity required to support the key claims.

We agree with the Reviewer that the current results might be influenced by the specific type of image features used, or from the dimensionality of the image features being reduced through PCA. To confirm that our findings are not a spurious product of the type of image features used or the downsampling of image features with PCA, we carried out two additional analyses. First, we trained and tested the encoding models using image features from individual deep neural network (DNN) layers, and mapped these image features onto the brain using either linear regression (after downsampling the feature space to 250 principal components), or ridge regression (on the full feature space, that is, not reduced with PCA). Second, we trained and tested the encoding models on image features from several DNNs. In both cases this did not change our findings, namely that out-of-distribution (OOD) tests on NSD-synthetic result in lower generalization performances than in-distribution (ID) tests on NSD-core (**Fig. 4**), and that NSD-synthetic reveals differences between models that are not detected ID (**Fig. 5**).

(Results (Fig. 4), page 13, lines 276-285): *“To ensure that the lower OOD performances are meaningful and robust, we performed three controls. First, to ensure that the lower performance on NSD-synthetic is not simply due to the fact that it was collected in a separate scan session, we trained and tested the encoding models using fMRI responses from separate NSD-core scan sessions (Supplementary Fig. 2). Second, to ensure that the specific regression method that one chooses to use does not change our conclusions, we trained and tested the encoding models on image features from individual AlexNet layers using linear or ridge regression (Supplementary Figs. 3-4). Third, to ensure that our results are not contingent on a specific choice of deep neural network, we trained and tested the encoding models on image features from deep neural networks other than AlexNet (Supplementary Figs. 5-7). In all three cases, we observed lower OOD performances for NSD-synthetic than for NSD-core.”*

"Supplementary Fig. 3 | In-distribution and out-of-distribution generalization comparison for encoding models based on individual AlexNet layers, trained with linear regression. We trained separate encoding models for each of 4 AlexNet layers: convolutional layer 1, convolutional layer 3, convolutional layer 5, and fully connected layer 7. The encoding models consisted of linear regressions that map each layer's image features onto the fMRI responses of each vertex. We tested these models in-distribution (ID) on NSD-core as well as out-of-distribution (OOD) on NSD-synthetic. This figure shows results averaged across participants on flattened cortical surfaces. a, ID explained variance normalized by the noise ceiling. To reduce false positives, for each vertex we averaged results from only those participants with a noise ceiling greater than 0.3 for NSD-core (vertices for which no participant has a noise ceiling above 0.3 are not shown). b, OOD explained variance normalized by the noise ceiling. For each vertex we averaged results from only those participants with a noise ceiling greater than 0.3 for NSD-synthetic. c, Difference between ID and OOD noise-ceiling-normalized explained variance. For each vertex, we averaged results from only those participants with noise ceiling greater than 0.3 for both NSD-core and NSD-synthetic. The asterisk indicates a significant difference between the ID and OOD noise-ceiling-normalized explained variance scores ($P < 10^{-6}$, paired sample t-test, one-sided, $N = 8$ participants)."

”Supplementary Fig. 4 | In-distribution and out-of-distribution generalization comparison for encoding models based on individual AlexNet layers, trained with ridge regression. We trained separate encoding models for each of 4 AlexNet layers: convolutional layer 1, convolutional layer 3, convolutional layer 5, and fully connected layer 7. The encoding models consisted of cross-validated ridge regressions that map each layer’s image features onto the fMRI responses of each vertex. We tested these models in-distribution (ID) on NSD-core as well as out-of-distribution (OOD) on NSD-synthetic. This figure shows results averaged across participants on flattened cortical surfaces. **a, ID explained variance normalized by the noise ceiling. To reduce false positives, for each vertex we averaged results from only those participants with a noise ceiling greater than 0.3 for NSD-core (vertices for which no participant has a noise ceiling above 0.3 are not shown). **b**, OOD explained variance normalized by the noise ceiling. For each vertex we averaged results from only those participants with a noise ceiling greater than 0.3 for NSD-synthetic. **c**, Difference between ID and OOD noise-ceiling-normalized explained variance. For each vertex, we averaged results from only those participants with noise ceiling greater than 0.3 for both NSD-core and NSD-synthetic. The asterisk indicates a significant difference between the ID and OOD noise-ceiling-normalized explained variance scores ($P < 10^{-6}$, paired sample t-test, one-sided, $N = 8$ participants).”**

”Supplementary Fig. 5 | In-distribution and out-of-distribution generalization comparison of vision-transformer-based encoding models. We built vision-transformer-based encoding models and tested them in-distribution (ID) on NSD-core as well as out-of-distribution (OOD) on NSD-synthetic. This figure shows results averaged across participants on flattened cortical surfaces. a, Model explained variance normalized by the noise ceiling. To reduce false positives, for each vertex we averaged results from only those participants with a noise ceiling greater than 0.3 (vertices for which no participant has a noise ceiling above 0.3 are not shown). b, Difference between ID and OOD noise-ceiling-normalized explained variance. For each vertex, we averaged results from only those participants with noise ceiling greater than 0.3 for both NSD-core and NSD-synthetic. The asterisk indicates a significant difference between the ID and OOD noise-ceiling-normalized explained variance scores ($P = 10^{-7}$, paired sample t -test, one-sided, $N = 8$ participants). c, Scatterplots of vertex-wise explained variance scores against their noise ceilings for ID (NSD-core) and OOD (NSD-synthetic) generalization tests, color coded by the visual stream that each vertex belongs to. For each vertex, we plotted results from only those participants with noise ceiling greater than 0.3 for both NSD-core and NSD-synthetic (same as in panel b). Black crosses indicate median scores across all vertices.”

”Supplementary Fig. 6 | In-distribution and out-of-distribution generalization comparison of ResNet50-based encoding models. We built ResNet50-based encoding models and tested them in-distribution (ID) on NSD-core as well as out-of-distribution (OOD) on NSD-synthetic. This figure shows results averaged across participants on flattened cortical surfaces. a, Model explained variance normalized by the noise ceiling. To reduce false positives, for each vertex we averaged results from only those participants with a noise ceiling greater than 0.3 (vertices for which no participant has a noise ceiling above 0.3 are not shown). b, Difference between ID and OOD noise-ceiling-normalized explained variance. For each vertex, we averaged results from only those participants with noise ceiling greater than 0.3 for both NSD-core and NSD-synthetic. The asterisk indicates a significant difference between the ID and OOD noise-ceiling-normalized explained variance scores ($P = 10^{-8}$, paired sample t-test, one-sided, $N = 8$ participants). c, Scatterplots of vertex-wise explained variance scores against their noise ceilings for ID (NSD-core) and OOD (NSD-synthetic) generalization tests, color coded by the visual stream that each vertex belongs to. For each vertex, we plotted results from only those participants with noise ceiling greater than 0.3 for both NSD-core and NSD-synthetic (same as in panel b). Black crosses indicate median scores across all vertices.”

Supplementary Fig. 7 | In-distribution and out-of-distribution generalization comparison of MoCo-based encoding models. We built MoCo-based encoding models and tested them in-distribution (ID) on NSD-core as well as out-of-distribution (OOD) on NSD-synthetic. This figure shows results averaged across participants on flattened cortical surfaces. **a**, Model explained variance normalized by the noise ceiling. To reduce false positives, for each vertex we averaged results from only those participants with a noise ceiling greater than 0.3 (vertices for which no participant has a noise ceiling above 0.3 are not shown). **b**, Difference between ID and OOD noise-ceiling-normalized explained variance. For each vertex, we averaged results from only those participants with noise ceiling greater than 0.3 for both NSD-core and NSD-synthetic. The asterisk indicates a significant difference between the ID and OOD noise-ceiling-normalized explained variance scores ($P = 10^{-7}$, paired sample t -test, one-sided, $N = 8$ participants). **c**, Scatterplots of vertex-wise explained variance scores against their noise ceilings for ID (NSD-core) and OOD (NSD-synthetic) generalization tests, color coded by the visual stream that each vertex belongs to. For each vertex, we plotted results from only those participants with noise ceiling greater than 0.3 for both NSD-core and NSD-synthetic (same as in panel **b**). Black crosses indicate median scores across all vertices.”

(Results (Fig. 5), page 15, lines 335-342): “To ensure that the OOD differences not detected by ID tests are meaningful and robust, we performed two controls. First, to ensure that our results are not simply due to the fact that NSD-synthetic was collected in a separate scan session, we trained and tested the encoding models using fMRI responses from separate NSD-core scan sessions (Supplementary Fig. 8). Second, to ensure that the specific regression method that one chooses to use does not change our conclusions, we trained and tested the encoding models on image features from individual ResNet50 or MoCo layers using either linear or ridge regressions

(Supplementary Figs. 9-12). In both cases, OOD tests revealed differences between models not detected by ID tests.”

”Supplementary Fig. 9 | In-distribution generalization differences between encoding models based on individual MoCo and ResNet50 layers, trained with linear regression. We trained separate encoding models for each of the 4 MoCo and ResNet50 layers: convolutional layer 1, convolutional layer 2, convolutional layer 3, and convolutional layer 4. The encoding models consisted of linear regressions that map each layer’s image features onto the fMRI responses of each vertex. We then tested these models in-distribution (ID) on NSD-core. This figure shows results averaged across participants on flattened cortical surfaces. **a**, MoCo’s ID explained variance normalized by the noise ceiling. To reduce false positives, for each vertex we averaged results from only those participants with a noise ceiling greater than 0.3 for NSD-core (vertices for which no participant has a noise ceiling above 0.3 are not shown). **b**, ResNet50’s ID explained variance normalized by the noise ceiling. For each vertex we averaged results from only those participants with a noise ceiling greater than 0.3 for NSD-synthetic. **c**, Difference between MoCo and ResNet50’s ID noise-ceiling-normalized explained variance. For each vertex, we averaged results from only those participants with noise ceiling greater than 0.3 for both NSD-core and NSD-synthetic.”

Supplementary Fig. 10 | Out-of-distribution generalization differences between encoding models based on individual MoCo and ResNet50 layers, trained with linear regression. We trained separate encoding models for each of the 4 MoCo and ResNet50 layers: convolutional layer 1, convolutional layer 2, convolutional layer 3, and convolutional layer 4. The encoding models consisted of linear regressions that map each layer's image features onto the fMRI responses of each vertex. We then tested these models out-of-distribution (OOD) on NSD-synthetic. This figure shows results averaged across participants on flattened cortical surfaces. **a**, MoCo's OOD explained variance normalized by the noise ceiling. To reduce false positives, for each vertex we averaged results from only those participants with a noise ceiling greater than 0.3 for NSD-core (vertices for which no participant has a noise ceiling above 0.3 are not shown). **b**, ResNet50's OOD explained variance normalized by the noise ceiling. For each vertex we averaged results from only those participants with a noise ceiling greater than 0.3 for NSD-synthetic. **c**, Difference between MoCo and ResNet50's OOD noise-ceiling-normalized explained variance. For each vertex, we averaged results from only those participants with noise ceiling greater than 0.3 for both NSD-core and NSD-synthetic."

Supplementary Fig. 11 | In-distribution generalization differences between encoding models based on individual MoCo and ResNet50 layers, trained with ridge regression. We trained separate encoding models for each of the 4 MoCo and ResNet50 layers: convolutional layer 1, convolutional layer 2, convolutional layer 3, and convolutional layer 4. The encoding models consisted of cross-validated ridge regressions that map each layer's image features onto the fMRI responses of each vertex. We then tested these models in-distribution (ID) on NSD-core. This figure shows results averaged across participants on flattened cortical surfaces. **a**, MoCo's ID explained variance normalized by the noise ceiling. To reduce false positives, for each vertex we averaged results from only those participants with a noise ceiling greater than 0.3 for NSD-core (vertices for which no participant has a noise ceiling above 0.3 are not shown). **b**, ResNet50's ID explained variance normalized by the noise ceiling. For each vertex we averaged results from only those participants with a noise ceiling greater than 0.3 for NSD-synthetic. **c**, Difference between MoCo and ResNet50's ID noise-ceiling-normalized explained variance. For each vertex, we averaged results from only those participants with noise ceiling greater than 0.3 for both NSD-core and NSD-synthetic."

Supplementary Fig. 12 | Out-of-distribution generalization differences between encoding models based on individual MoCo and ResNet50 layers, trained with ridge regression. We trained separate encoding models for each of the 4 MoCo and ResNet50 layers: convolutional layer 1, convolutional layer 2, convolutional layer 3, and convolutional layer 4. The encoding models consisted of cross-validated ridge regressions that map each layer's image features onto the fMRI responses of each vertex. We then tested these models out-of-distribution (OOD) on NSD-synthetic. This figure shows results averaged across participants on flattened cortical surfaces. **a**, MoCo's OOD explained variance normalized by the noise ceiling. To reduce false positives, for each vertex we averaged results from only those participants with a noise ceiling greater than 0.3 for NSD-core (vertices for which no participant has a noise ceiling above 0.3 are not shown). **b**, ResNet50's OOD explained variance normalized by the noise ceiling. For each vertex we averaged results from only those participants with a noise ceiling greater than 0.3 for NSD-synthetic. **c**, Difference between MoCo and ResNet50's OOD noise-ceiling-normalized explained variance. For each vertex, we averaged results from only those participants with noise ceiling greater than 0.3 for both NSD-core and NSD-synthetic."

4. A further important concern is that, although the stated goal of this study is to advance the modeling of visual processing under out-of-distribution (OOD) conditions, the analyses rely solely on simple linear encoding models.

Recent work has increasingly raised concerns about the validity and interpretability of encoding analyses. For instance, Nonaka et al. reported that in feature-selection-based encoding models, only around 40% of model units contributed to prediction, suggesting that

much of the model's representational richness may be ignored in the encoding analysis. Moreover, there is growing skepticism about the overreliance on encoding performance metrics alone, particularly in the context of benchmarks such as brain-score. Malhotra and Bowers (2024), for example, showed that encoding accuracy can sometimes depend more on image background than on object content, raising questions about what these models are actually learning to predict. In the current study, it also remains unclear what aspects of the image features are being used for prediction—whether they reflect ideal image content, semantic content, trivial properties such as average luminance, session-specific noise, or other unexamined factors. While the recent work by Schaeffer et al. (2024) has several problems, their claim about the fundamental concerns of frameworks that focus primarily on maximizing encoding scores should not be neglected.

Taking these recent discussions into account, this study that relies exclusively on standard encoding performance—without addressing potential confounding factors or validating the interpretability of the predicted responses—falls short of demonstrating meaningful progress in the field. The authors should consider conducting more multifaceted evaluations—such as testing whether the predicted responses can identify the presented stimuli—to better assess whether the predictions genuinely reflect stimulus related information.

We appreciate (and agree) with the Reviewer's recognition of the potential limitations/pitfalls of encoding model metrics. Following the Reviewer's suggestion, as a complementary way of assessing the encoding models' generalization performance we used the encoding models' predicted fMRI responses to identify, in a zero-shot fashion, the stimulus images of the recorded test fMRI responses (for the corresponding results text and figures, see Reviewer #1 comment 1, and Reviewer #1 comment 5). Reassuringly, we found that the zero-shot identification yielded similar result patterns to the standard encoding performance (i.e., NSD-synthetic image classes resulting in lower encoding accuracies, such as chromatic noise, also resulted in lower identification performances). Together, this suggests that the model predictions genuinely reflect stimulus related information, and that the reported result pattern is not a bias of the evaluation metric used.

Beyond this, to rule out several confounding factors related to our encoding modeling results, we carried out extensive additional analyses (reported above) which show that our encoding results are not a spurious product of session-specific noise (see Reviewer #1 comment 2), nor of the type of image features or regression method used (see Reviewer #1 comment 3).

Finally, based on the Reviewer's comments concerning the interpretability of results from encoding models, together with several helpful discussions on this topic with other researchers at the 2025 Cognitive Computational Neuroscience conference, we also added a new Discussion paragraph where we elaborate on the interpretation of OOD generalization performances of computational models of the brain implemented through the linear encoding approach (**Discussion, page 23, lines 537-561**):

*“To obtain valid inferences, the details, quantifications, and interpretation of OOD tests require careful consideration. Here, we comment on **four** specific aspects that researchers should consider when performing OOD generalization tests.*

[...]

Second, there are several approaches to implement and test computational models of the brain, each with its unique strengths, and which approach should be best used towards a particular scientific goal is a current topic of debate (Diedrichsen & Kriegeskorte, 2017; Doerig et al., 2023; Schaeffer et al., 2024). Here we used the linearizing encoding approach (Naselaris et al., 2011), which consists of two independently trainable components: the image-computable model (typically a deep neural network trained on a computer vision task) used to extract the image features, and the linear mapping trained to map these image features onto brain responses. Thus, the outcomes of our OOD tests reflect the combination of both the image-computable model and the linear mapping. We note that other researchers might wish to test only the OOD generalization performance of the image-computable model. For this modeling philosophy, a different analytic approach would be necessary, such as representational similarity analysis (with no feature re-weighting) (Kriegeskorte et al., 2008), or determining a one-to-one mapping between model features and neural units (Finzi et al., 2025).

Minor Comments:

5. The current OOD set seems to be heavily skewed toward spiral gratings (~40% of stimuli). As such, there is a possibility that aggregate performance in Figures 4 and 5 are dominated by this single stimulus type. To assess the encoding performance on OOD stimuli more clearly, I recommend reporting encoding performance separately by OOD category (e.g., spiral gratings, hue-manipulated scenes, words, line drawings, etc.). Even as supplementary material, such an analysis would help users understand whether performance drops generalize across stimulus types or are specific to particular OOD classes.

We thank the Reviewer for the suggestion, which led to a new Results section where we tested the OOD generalization of encoding models independently for each NSD-synthetic image class. Together, this new analysis showed that the degree of OOD—that is, the distance between the test and train fMRI responses—is informative of the model’s generalization performance to these test responses. Thus, the degree of OOD is a useful index of the stimulus properties an encoding model might fail to generalize to, in turn providing explicit targets for model improvement (**Results (Fig. 6), page 17, lines 347-422**):

“The degree of out-of-distribution is informative of brain model failures

Fig. 6 | The degree of out-of-distribution is informative of brain model failures.

We trained vision-transformer-based encoding models on NSD-core, and tested them independently for each NSD-synthetic image class. **a**, MDS embedding of trial-averaged fMRI responses (aggregated across participants) from NSD-core (gray dots) and from NSD-synthetic (colored dots, color-coded according to image class). **b**, Encoding models' explained variance for each NSD-synthetic image class, normalized by the noise ceiling. This figure shows results averaged across participants on flattened cortical surfaces. To reduce false positives, for each vertex we averaged results from only those participants with a noise ceiling greater than 0.3 (vertices for which no participant has a noise ceiling above 0.3 are not shown). **c**, Scatterplot indicating the relationship between the degree of OOD—that is, the degree of distributional distance—of fMRI responses and the encoding accuracy scores, across NSD-synthetic image classes. The x-axis represents the Euclidean distance in MDS embedding space between fMRI responses for each NSD-synthetic image class and NSD-core (i.e., the distances in panel a). The y-axis represents the noise-ceiling normalized encoding accuracy of encoding models tested on each NSD-synthetic image class (i.e., the mean

scores in panel **b**). **d**, Encoding models' zero-shot identification scores of the recorded fMRI response stimulus images, for each NSD-synthetic image class, on violin plots. The y-axis represents the correct image rank, that is, the position at which the correct image appears among the models' choices (choosing among all 284 NSD-synthetic images), with lower ranks indicating a more robust image identification. The colored violin shapes indicate the distribution of observations, with wider sections corresponding to higher density of values. Small transparent dots indicate the correct image rank for single images and participants. Large black dots indicate the average correct image rank across all participants and images within each NSD-synthetic image class, with the corresponding score reported above the dot. Horizontal dashed lines indicate chance identification scores. **e**, Scatterplot indicating the relationship between the degree of OOD of fMRI responses and the zero-shot image identification scores, across NSD-synthetic image classes. The x-axis represents the Euclidean distance in MDS embedding space between fMRI responses for each NSD-synthetic image class and NSD-core (i.e., the distances in panel **a**). The y-axis represents the average correct image rank across all participants and images within each NSD-synthetic image class (i.e., the average scores in panel **d**). **c,e**, Large opaque and small transparent dots indicate participant-average and single participant scores, respectively. The asterisk indicates a significant correlation between response distance scores and encoding model performance ($P < 0.001$, one-sample t-test, one-sided, $N = 8$ participants).

The neural responses on which a brain model is tested can exhibit various degrees of OOD—that is, various degrees of distributional distance—with respect to the responses on which the model is trained. Therefore, we ask whether there is a relationship between the extent to which the fMRI responses used to test an encoding model are differently distributed from the train responses and the encoding model's generalization performance to these test responses. If so, the degree of OOD could serve as an indicator of the stimulus properties that encoding models might fail to account for, which may help guide the engineering of more robust models (Madan et al., 2024; Kay 2018). We operationally define the degree of OOD as the Euclidean distance between the fMRI response to a given stimulus of interest and the NSD-core train responses.

Using MDS we reduced the trial-averaged fMRI responses for NSD-synthetic (used to test the encoding models) and for NSD-core (used to train the models), to two dimensions. Within this two-dimensional embedding space we then computed the Euclidean distance between the responses of each NSD-synthetic image and the responses of each NSD-core image, and averaged these distance scores across all NSD-core images, as well as across NSD-synthetic images belonging to the same image class. This resulted in distributional distance scores indicating, for each NSD-synthetic image class, the degree of OOD of its fMRI responses from NSD-core's responses (**Fig. 6a**).

Next, we compared these distributional distance scores with the noise-ceiling-normalized encoding accuracy scores of vision-transformer-based encoding models trained on NSD-core and tested on each NSD-synthetic image class (**Fig. 6b**). We found that the distributional distance scores are anticorrelated with the encoding accuracy scores (Spearman's $\rho = -0.36$, $P = 0.0004$, one-sample t-test, one-sided, $N = 8$ participants) (**Fig. 6c**), indicating that encoding models generalize

better to fMRI responses that have lower distributional distance to the responses on which the models are trained.

*As a complementary way of assessing the encoding models' generalization performance for the different NSD-synthetic image classes, we used the models' predicted fMRI responses to identify, in a zero-shot fashion, the stimulus images of NSD-synthetic's recorded fMRI responses (Kay et al., 2008; Horikawa & Kamitani, 2017; Seeliger et al., 2018), choosing among all 284 NSD-synthetic images (**Fig. 6d**). We found that the ranks of the correct stimulus image (i.e., the position at which the correct image appears among the models' choices) correlate with the distributional distance scores (Spearman's $\rho = 0.63$, $P = 10^{-6}$, one-sample t -test, one-sided, $N = 8$ participants) (**Fig. 6e**), indicating that encoding models more reliably identify stimulus images when the corresponding fMRI responses have lower distributional distance to the model's train responses.*

*We obtained quantitatively similar results when using encoding models based on other deep neural network architectures (**Supplementary Figs. 13-15**), indicating that the correlation between distributional distance scores and encoding model performance is not contingent on a specific choice of deep neural network.*

In summary, these results show that the degree to which an encoding model's test fMRI responses are OOD with respect to the model's train responses is informative of the model's generalization performance to these test responses. Thus, the degree of OOD is a useful indicator of the stimulus properties that encoding models might fail to account for.

In light of these new results, we made the following addition to the Discussion (**Discussion, page 22, lines 503-513**):

“Beyond indicating that further model improvement is required to accurately describe visual representations in the brain, NSD-synthetic also suggests directions for closing the gap between ID and OOD generalization performance. One direction of model improvement comes from consideration of the properties of NSD-synthetic's stimuli. Since the stimuli consist of a multitude of diverse and parameterized images, they facilitate discovery of the specific visual features to which computational models fail to generalize. Accordingly, we found that encoding models well generalize to test fMRI responses that have lower degree of OOD (i.e., responses for naturalistic or manipulated scenes), whereas they fail to generalize to fMRI responses with a higher degree of OOD (i.e., responses for chromatic noise or spiral gratings). To optimize modeling performance, it may be fruitful for future research efforts to improve robustness for stimulus properties with a high degree of OOD (Madan et al., 2024; Kay, 2018).”

6. L 300: The phrase “ResNet-50 is a recurrent convolutional neural network” is inaccurate. ResNet-50 is not a recurrent model and should be rephrased.

We thank the Reviewer for pointing out this incorrect model description, which we amended (**Results, page 15, lines 322-324**):

“Specifically, we compared encoding models based on ResNet-50, a convolutional neural network *with skip connections* pretrained on task-supervised image classification ...”

7. L 524: The method for generating line-drawing stimuli is unclear. Compared to other stimulus types, the description feels incomplete. Please provide more detail on the algorithm or pipeline used.

We added more detail on the approach used to generate the line-drawing stimuli (**Methods, page 28, lines 725-730**):

“Line-drawing scenes: Image subclass 7 (images 25–28). Line drawings were obtained from a previous study (Walther et al., 2011). In brief, trained artists at the Lotus Hill Research Institute created line drawings by tracing the contours of color photographs using a custom interface. The sequence and coordinates of each line stroke were digitally recorded. The line drawings associated with the natural scenes used in image subclass 4 were square cropped, resized to 714 pixels × 714 pixels, and then prepared as black lines (0) on a gray background (0.5). (Manipulated scenes)”

Although the manuscript introduces a useful dataset, it does not currently meet the journal’s standards for presenting conceptually novel and methodologically rigorous work that offers significant advances for specialists in the field. For these reasons, I do not recommend this manuscript for publication in this journal.

We thank the Reviewer for raising these important concerns. We believe that addressing these review comments strengthened the rigour, novelty, and usefulness of NSD-synthetic.

****Reference:****

Malhotra, G. & Bowers, J. Predicting brain activation does not license conclusions regarding DNN-brain alignment: The case of Brain-Score. in *Cognitive Computational Neuroscience* (Boston, USA, 2024).

https://2024.ccneuro.org/pdf/309_Paper_authored_Neural_Predictivity_CCN2024.pdf.

Nonaka, S., Majima, K., Aoki, S. C. & Kamitani, Y. Brain hierarchy score: Which deep neural networks are hierarchically brain-like? *iScience* 24, 103013 (2021).

Schaeffer, R. et al. Position: Maximizing Neural Regression Scores May Not Identify Good Models of the Brain. in *UniReps workshop* (Vancouver, Canada, 2024).

<https://openreview.net/pdf?id=vbtj05J68r>

Shirakawa _et al._ Spurious reconstruction from brain activity. *Neural Networks (in press)*

We thank the Reviewer for these references.

(2025). <https://doi.org/10.48550/arXiv.2405.10078>

Reviewer #2

Remarks to the Author

We thank Reviewer #2 for co-reviewing the manuscripts and for the helpful comments.

Reviewer #3

Remarks to the Author

A very valuable resource for the visual neuroscience community. The original Natural Scenes Dataset of 7T fMRI responses to over 9,000 scenes (Nature Neuroscience, 2021) is one of the largest and most used open datasets in visual neuroscience (557 citations). The current paper announces and describes a supplement to this dataset, which seems to have been collected around the same time, but only released in March of this year. The supplement consists of 7T fMRI responses from the same individual subjects, to an additional 284 synthetic images (either manipulated, e.g. via contrast reduction or phase scrambling; or artificially generated e.g. colour fields or spatial-frequency-controlled spiral patterns). Although the new data supplement is much smaller than the original core dataset, it will be useful both for testing encoding models outside the distribution of natural images (the main purpose advocated by the authors) and for probing the cortical representation of colour, orientation, and spatial frequency. The stimuli in the dataset are probably not as exhaustively parameterised for this second purpose as a researcher would desire (e.g. colour hue space is sampled, but not the saturation and lightness axes), but the dataset will nevertheless be helpful for some carefully-chosen research questions (as well as serving as the intended out-of-distribution test set).

The paper is very clearly written and figures and methods are excellently presented. The analyses of the paper go beyond the "minimum required" to describe and quality-check the data, and also evaluate a number of deep neural network based encoding models

We thank Reviewer #3 for the positive evaluation and helpful comments, which resulted in valuable edits and additions to the manuscript.

Minor comments:

1. line 46 "intelligence" seems like an odd way to characterise the research that would be done with this dataset. Why not "visual neuroscience" or similar?

Based on the Reviewer's suggestion, we changed "intelligence" to "visual neuroscience" (**Introduction, page 2, lines 51-52**):

*"However, OOD tests are critical for **visual neuroscience** research in three significant ways."*

2. Figure 1B "Manipulates scenes" should read "Manipulated scenes".

We thank the Reviewer for spotting this typo, which we fixed.

3. lines 96-101: It would be good to already acknowledge in the main text two important differences between the NSD-core and NSD-synthetic datasets - the temporal design (2s vs 3s on), and the task (mixture of fixation and same/different tasks, vs continuous old/new task). These differences do limit the comparability of the main and supplemental dataset and should be acknowledged early.

We added NSD-core's differing trial and task information to the Introduction (**Introduction, page 5, lines 103-107**):

*"We presented these images in a rapid event-related design consisting of 4-s trials (2-s ON, 2-s OFF; **instead of NSD-core's 3s ON, 1-s OFF trials**) while measuring fMRI responses (7T, 1.8-mm resolution, TR 1.6 s) from each of the eight NSD participants (**Figure 1C**). To assess potential task dependence of neural responses, participants performed a fixation task and a one-back task in alternating runs (**instead of NSD-core's long-term continuous recognition task**) (**Figure 1D**)."*

Furthermore, we acknowledged the spurious distributional differences introduced by distinct task and temporal design choices between NSD-core and NSD-synthetic in a dedicated Discussion section, together with suggestions on how to mitigate possible unwanted effects of these differences on modeling analyses (**Discussion, page 23, lines 537-587**):

*"To obtain valid inferences, the details, quantifications, and interpretation of OOD tests require careful consideration. Here, we comment on **four** specific aspects that researchers should consider when performing OOD generalization tests.*

[...]

***Third**, we interpreted the distributional differences between NSD-synthetic and NSD-core's fMRI responses as stemming from differences in the stimuli of the two datasets. However, in theory, distributional differences might also reflect other properties of the data, such as differences in task (Kay et al., 2023) or trial timing (Zhou et al., 2017), session-specific effects, and fluctuations in cognitive state. All these data properties represent diverse dimensions across which different datasets might fall in- or out-of-distribution. Therefore, to improve transparency, reproducibility,*

and comparison across studies using NSD-synthetic, we recommend researchers to explicitly define the OOD operationalization used.

Fourth, properties of the data not relevant to the researcher’s question of interest might introduce spurious distributional differences between NSD-synthetic and NSD-core’s fMRI responses. For example, the trial timing differed between NSD-core (3-s ON, 1-s OFF) and NSD-synthetic (2-s ON, 2-s OFF), with longer image onscreen times typically leading to higher fMRI responses (Zhou et al., 2017). Similarly, NSD-core and NSD-synthetic differed in task, with tasks also affecting fMRI response magnitude (Kay et al., 2023). Additionally, since NSD-synthetic and NSD-core’s fMRI responses come from distinct scan sessions, they are affected by session-specific effects, that is, differences in fMRI responses across sessions due to incidental session-specific factors (e.g., time of day, arousal, cognitive state, hardware state). To eliminate spurious distributional differences, researchers might consider z-scoring fMRI responses within each scan session. However, by centering responses and transforming them to unit variance, z-scoring also alters the signal of interest, potentially eliminating distributional differences of interest. Our approach for compensating for differences in fMRI response gain due to trial timing was to quantify encoding model performance using Pearson correlation (which is insensitive to gain differences). Furthermore, to reduce session-specific effects we analyzed fMRI responses aggregated across participants, considered vertices with a signal-to-noise ratio score above a certain threshold (see Methods), and performed control analyses where we trained and tested encoding models on fMRI data from separate NSD-core scan sessions.”

4. line 128: It would be helpful to mention how the noise ceilings measured in this dataset compare to those in the NSD-core dataset.

We elaborated on the comparison between NSD-synthetic and NSD-core’s noise ceiling signal-to-noise ratio (NCSNR) through text and a new Supplementary Figure (**Results, page 7, lines 129-144**):

“NSD-synthetic’s main purpose is to facilitate out-of-distribution (OOD) generalization tests aimed at improving the robustness of computational models of the brain. A prerequisite of these tests is that NSD-synthetic’s fMRI responses must reliably encode the different stimulus images. To assess this, we calculated NSD-synthetic’s noise ceiling signal-to-noise ratio (NCSNR), a measure of stimulus-related signal in the fMRI responses, using methods introduced in previous work (Allen et al., 2022). For all participants, we found that NCSNR scores are high over visual cortex, ranging approximately from 0.75–2 (Fig. 2a). This indicates that 36–80% variance in single-trial responses reflects signals driven by the stimulus. As a comparison, NSD-core’s NCSNR scores range approximately from 0.5–1.25, indicating that 20–60% variance in single-trial responses reflects signals driven by the stimulus (Supplementary Fig. 1). Hence, stimulus-related information is indeed reliably encoded in NSD-synthetic’s fMRI responses. Within visual cortex, NSD-synthetic’s NCSNR is highest in early areas compared to intermediate, ventral, dorsal, and lateral areas. We believe this reflects the limited visual and semantic complexity of NSD-synthetic’s stimuli, which leads to reduced signals in higher visual areas that

preferentially respond to more complex visual features. Supporting this intuition, the visually and semantically more complex naturalistic images from NSD-core lead to NCSNR scores that are more uniform across visual cortex (**Supplementary Fig. 1**).”

“Supplementary Fig. 1 | NSD-synthetic and NSD-core’s noise ceiling signal-to-noise ratio (NCSNR). a, NSD-synthetic’s participant-wise noise ceiling signal-to-noise ratio (NCSNR), plotted on flattened cortical surfaces. b, NSD-core’s participant-wise NCSNR, plotted on flattened cortical surfaces. a-b, White contours indicate regions based on the ‘streams’ ROI collection as provided in the NSD data release.”

5. line 300: ResNet is a feedforward architecture with skip connections, in my understanding, not a recurrent network (i.e. it has no temporal dynamics or memory over stimuli).

We thank the Reviewer for pointing out this incorrect model description, which we amended (**Results, page 15, lines 322-324**):

“Specifically, we compared encoding models based on ResNet-50, a convolutional neural network *with skip connections* pretrained on task-supervised image classification ...”

Remarks on code availability

I have skimmed but not attempted to use any of the code. It appears well documented and thorough. Code is provided to reproduce all figures in the paper, as well as sharing the images and fMRI data.

We thank the Reviewer for checking the code.

Reviewer #4

Remarks to the Author

The authors present NSD-synthetic, a companion dataset to the Natural Scenes Dataset (NSD), consisting of 7T fMRI responses from 8 subjects viewing 284 synthetic images. This dataset is designed to enable out-of-distribution (OOD) generalization tests for computational models of visual processing. The authors demonstrate that neural responses to these synthetic stimuli are reliably encoded and are indeed OOD relative to NSD-core. They also present encoding model comparisons showing differences between self-supervised and task-supervised models in OOD settings, limited to early/mid visual cortex where the synthetic stimuli drive robust responses.

Strengths

This is an exceptionally valuable resource for the computational neuroscience community. The dataset addresses a critical gap in the field - namely, the ability to test whether models of visual processing generalize beyond the natural image distributions they are typically trained on. The experimental design is rigorous, using the same 8 subjects from NSD-core and maintaining high data quality standards with 7T fMRI at 1.8mm resolution. The open release of this data will enable researchers to develop more robust models and better understand the limitations of current approaches.

The technical quality is, as always from Ale, excellent. Comprehensive validation shows that the synthetic stimuli reliably drive neural responses, particularly in early visual areas. The use of multidimensional scaling to confirm the OOD nature of the responses is convincing, and the variety of synthetic stimuli (noise patterns, manipulated scenes, gratings, words) provides good coverage of different visual features.

Of particular interest, but also a potential confound, is the choice of artificial stimuli. As deep neural networks are (often) surprisingly bad using these type of stimuli. Using stimuli like spiral gratings, phase-scrambled images, etc. etc., the dataset probes a known weakness of

current computational models. This makes NSD-synthetic an excellent diagnostic tool for understanding where and why DNNs fail to capture human visual processing.

We thank Reviewer #4 for the very positive evaluation and helpful comments, which resulted in useful clarifications in the manuscript's text and figures regarding the validity of our findings across both low- and high-level visual cortex, as well as in a new discussion and analysis related to the comparison of encoding models based on ResNet50 and MoCo deep neural networks.

Major/Medium comments:

1. The authors show that self-supervised models (MoCo) outperform task-supervised models (ResNet-50) in OOD generalization, and that vision transformers outperform CNNs. For early visual cortex (V1-V4), where the synthetic stimuli drive robust responses, these findings appear valid and represent an important discovery about differential model architectures. The high noise ceiling values in these areas (Figure 2A) support meaningful model comparisons. We do not really know what happens in other parts of cortex, or if the models would generalise better towards OOD stimuli for areas in higher Cortex (e.g. OOD stimuli that the cortex would respond to). Not sure what to do about this, you could look at OOD effects vs SNR effects. It could also really be that they are restricted to early cortex. But in any case I think it is important to reframe the conclusion a little bit to reflect that the OOD generalization findings are strongest for early visual cortex, with important caveats about interpretability in higher areas.

We agree with the Reviewer that the lower SNR over early compared to higher visual cortex could potentially limit the strength of the conclusions that one can make about OOD generalization on higher visual areas, and we do explicitly address this in the manuscript. However, because the fMRI responses of NSD-synthetic's higher visual areas are not devoid of signal, we do not believe that NSD-synthetic prevents meaningful conclusion concerning higher visual areas. Following we lay out the arguments/analyses of why we think this to be the case.

While in **Fig. 2a** we show that the noise ceiling is highest in early areas compared to intermediate, ventral, dorsal, and lateral areas, these results do not show that higher-level visual areas do not encode visual information at all. In fact, we show that the responses of intermediate, ventral, dorsal, and lateral areas also consist of stimulus-related signal based on their noise ceiling (**Fig. 2a**), and based on the results of univariate (**Fig. 2b-e**) and multivariate (**Fig. 2f**) analyses (**Results (Fig. 2), page 6, lines 115-186**):

“Fig. 2 | Noise ceiling, univariate, and multivariate analyses reveal robust visual signals in NSD-synthetic. a, Participant-wise noise ceiling signal-to-noise ratio (NCSNR), plotted on flattened cortical surfaces. White contours indicate regions based on the ‘streams’ ROI collection as provided in the NSD data release. **b,** ROI-wise participant-average univariate responses for noise, natural/manipulated scenes, contrast modulation, and phase-coherence modulation (groupings indicated by vertical white lines). **c,** ROI-wise responses for chromatic noise (Hue01–Hue16) and corresponding achromatic noise (Hue00). **d,** ROI-wise responses for single words. For visualization purposes, the images with the most peripheral words are not square cropped (Word4 Pos1, Word4 Pos5, Word6 Pos1, Word6 Pos5). **e,** ROI-wise responses for spiral gratings. **f,** RSA scores indicating the similarity of multivariate fMRI responses between participants

and ROIs. Thin white lines separate groups of the eight participants within the same ROI. Cyan boxes indicate cross-participant comparisons within the same ROI.”

Furthermore, when comparing the in-distribution (ID) and out-of-distribution (OOD) encoding accuracies within the same model, or across models, to prevent our results from being biased by vertices that do not encode stimulus-related signal (i.e., noisy vertices), we only considered vertices with noise ceiling above the threshold of 0.3 (i.e., vertices with at least 30% of variance consisting of signal). Even after this thresholding, a large portion of ventral, lateral, and dorsal cortex vertices were retained for NSD-synthetic, indicating that our OOD findings are also valid for a significant amount of high-level visual cortex (**Fig. 4c-d; Fig. 5**). To make this more explicit, we edited the scatterplots in **Fig. 4e** to highlight the fact that, when tested OOD, the encoding models fail to account for large portions of explainable variance across both lower- and higher-level visual cortex (**Results (Fig. 4), page 12, lines 268-275**):

*“We further observed that decreases in OOD performance compared to ID performance are consistent across both lower- and higher-level visual areas (see red regions in **Fig. 4d**). Importantly, these decreases in performance are not due to lack of explainable signal in NSD-synthetic’s fMRI responses, but rather to model failures. When plotting the vertex-wise explained variance scores against their noise ceilings, we found that compared to ID tests on NSD-core, the encoding models’ explained variance scores were consistently lower relative to the noise ceiling for OOD tests on NSD-synthetic, **across both lower- and higher-level visual areas (Fig. 4e).**”*

“Fig. 4 | Brain encoding models exhibit reduced performance when tested out-of-distribution on NSD-synthetic. We built AlexNet-based encoding models and tested them in-distribution (ID) on NSD-core as well as out-of-distribution (OOD) on NSD-synthetic. This figure shows results averaged across participants on flattened cortical surfaces. **a**, Model explained variance (r^2). **b**, Noise ceiling (r^2). **c**, Model explained variance normalized by the noise ceiling. To reduce false positives, for each vertex we averaged results from only those participants with a noise ceiling greater than 0.3 (vertices for which no participant has a noise ceiling above 0.3 are not shown). **d**, Difference between ID and OOD noise-ceiling-normalized explained variance. For each vertex, we averaged results from only those participants with noise ceiling greater than 0.3 for both NSD-core and NSD-synthetic. The asterisk indicates a significant difference between the ID and OOD noise-ceiling-normalized explained variance scores ($P = 10^{-7}$, paired sample t -test, one-sided, $N = 8$ participants). **e**, Scatterplots of vertex-wise explained variance scores against their noise ceilings for ID (NSD-core) and OOD

(NSD-synthetic) generalization tests, *color coded by the visual stream that each vertex belongs to*. For each vertex, we plotted results from only those participants with noise ceiling greater than 0.3 for both NSD-core and NSD-synthetic (same as in panel d). Black crosses indicate median scores across all vertices.”

Of course, even though in our encoding analyses we only considered vertices with noise ceiling above the threshold of 0.3, since noise ceilings are higher for early visual cortex it could be that our findings might also be more robust for early visual cortex. We therefore made this more explicit in the limitations paragraph of the Discussion section (**Discussion, page 25, lines 628-635**):

“The results of our analyses indicate that stimulus-related information in NSD-synthetic’s fMRI responses is primarily encoded in early to intermediate visual cortical areas (i.e., V1 to hV4). This presumably is due to the fact that NSD-synthetic stimuli consist mostly of simple visual features that well activate early to intermediate visual cortex (e.g., various forms of noise and spiral gratings), but which are not well suited for driving activity in higher visual cortical areas responsive to perceptually and semantically more complex stimuli. Because of this, *the results and conclusions obtained with NSD-synthetic might be most robust when testing the OOD generalization of computational models of lower visual cortical areas.*”

2. The paper lacks discussion of why this might occur. and why this occurs for early cortex. Is the dictionary of these networks just better? Or because contrastive learning explicitly trains models to be invariant to various image transformations, and that this generalises towards these artificial stimuli?

In this cortex it might be an idea to look at layers in MOCO in relation to effects?
Some mechanistic insight would really help.

Our guess is that MoCo outperforms ResNet50 in OOD generalization due to its augmented training diet from the self-supervised contrastive learning objective. We elaborated on this in the Discussion section (**Discussion, page 22, lines 514-528**):

“Another direction of model improvement comes from benchmarking the performance of different computational models on NSD-synthetic and isolating model properties leading to best OOD generalization (Golan et al., 2020; Ren & Bashivan, 2025). Because each model embeds a different hypothesis of visual processing, these OOD generalization tests can help adjudicate between competing hypotheses, therefore facilitating theory formation. We found that encoding models based on a self-supervised deep neural network better generalize OOD than encoding models based on its task-supervised counterpart, in line with recent work proposing self-supervision as a more plausible account of coding in visual cortex (Konkle & Alvarez, 2022; Doerig et al., 2023; Prince et al., 2024). *The improved performance of self-supervised deep neural networks might derive from their training diet, which beyond colored naturalistic images also includes augmented images such as grayscale or blurred images. Compared to the fully naturalistic colored images typically used to train task-supervised deep neural networks, these augmented images are more similar to the artificial images from NSD-synthetic, therefore resulting in encoding models with better OOD generalization performances.* Future efforts that

systematically benchmark a larger variety of encoding models (Conwell et al., 2024) might confirm this finding, and also isolate further model properties that improve OOD generalization performance.”

As suggested by the Reviewer, we also compared the ID and OOD encoding performances using individual ResNet50 and MoCo layers. This layerwise analysis revealed a complex effect landscape, where MoCo does not better generalize OOD compared to ResNet50 for all layers, but only for some layers or when all layers are jointly used to train/test the encoding models. Nevertheless, the layerwise analysis further confirmed our effect of interest in that OOD tests reveal differences between models that are not detected with ID tests. For these new analyses and results, please see the response Reviewer #1 comment 3 above.

Remarks on code availability

I perused the code. It seems complete and is intelligible.

We thank the Reviewer for checking the code.

Dear Reviewers,

Please find the detailed point-by-point response below. Reviewer comments are highlighted in gray, and the corresponding responses indented. Within the responses, extracts from the manuscript are “*quoted in italics*”, and extracts reflecting changes to the manuscript are “*quoted in italics and red font*”. Manuscript line/page/figure numbers are indicated in **bold font**. Blue underscored text denotes hyperlinks.

Reviewer #1

Remarks to the Author

The authors have responded sincerely to the previous comments, and through extensive additional analyses and clarifications, the manuscript has clearly improved, addressing many of the initial concerns. At the same time, I think some of the new analyses and findings would benefit from more careful consideration and interpretation. I highly value the contribution of this dataset and recognize its significant potential impact on the field. Precisely for this reason, I believe that ensuring the analyses and interpretations are handled as rigorously as possible will further strengthen the manuscript.

We thank Reviewer #1 for reading the revised manuscript, for the positive evaluation, and for the helpful comments which resulted in valuable new analyses and discussions, as well as reinterpretations of our results. In response to the reviewer’s comments below, we have:

- (1) Performed a new analysis where we defined NSD-core’s ID and naturalistic OOD test splits based on a vision transformer image features (instead of based on fMRI responses).
- (2) Elaborated the discussion of the results obtained when controlling for session-specific effects.
- (3) Discussed the differences between applying or not applying PCA to the stimulus features when building encoding models of the brain.
- (4) Used balanced and common candidate set sizes in the zero-shot identification analyses.
- (5) Discussed the differences in OOD generalization performance between the individual NSD-synthetic image classes.
- (6) Added further clarifications and discussions as requested throughout the manuscript.

Major Comments:

[Related to my previous comments “1. The most significant concern ...”]

My first concern in the previous review was the treatment of OOD. In response, the authors redefined the training–test split in Figure 7 and prepared a naturalistic OOD test condition, which has clearly improved the manuscript. However, it remains somewhat unclear why the naturalistic OOD condition was prepared based on fMRI activity patterns rather than on stimuli. In both encoding and decoding approaches, it is generally more common to construct OOD conditions based on stimuli (Mitchell et al., 2008; Brouwer and Heeger, 2009; Madan et al., 2024; Shirakawa et al., 2025).

The manuscript would further benefit from considering reanalyses that use stimulus-based splits when preparing the naturalistic OOD test condition. The semantic feature–based data split in Shirakawa et al. (2025) could serve as an appropriate direction. Moreover, given the potential influence of session/run confounds (discussed later), it also strengthens the conclusions if the training–test split in Figure 7 further excluded session-related overlap.

Additionally, the motivation for the analysis in Figure 7 could be explained more clearly. In the Introduction, the statement that “(L76–77) the concept of OOD can be usefully applied even within the domain of naturalistic stimuli (Fig. 7)” feels somewhat abrupt in relation to the preceding paragraph (L45–51). Referring to Shirakawa et al. (2025) and their concerns about the diversity of NSD stimuli, as well as their decoding analyses, could help clarify the rationale. Finally, the procedure described in L450–L462 (low-dimensional projection, k-means clustering, and creating split) appears very similar to the procedure in Figure 6a of Shirakawa et al. (2025), and citing that paper here would strengthen the context.

In **Fig. 7**, we defined the ID and naturalistic OOD test splits based on fMRI responses in order to be conceptually consistent with **Fig. 3**, where we concluded that NSD-synthetic is OOD with respect to NSD-core based on the corresponding fMRI responses in MDS-space. However, we agree with the Reviewer that providing additional variants of the ID and naturalistic OOD test split would be a useful addition. We therefore followed a similar procedure to the one of Shirakawa et al., 2025, and defined the ID and naturalistic OOD test splits based on the image activations of a deep neural network. This new analysis led to quantitatively similar results (i.e., decreasing model generalisation performances for ID, naturalistic OOD, and synthetic OOD test splits), indicating that this ranking between test splits is meaningful and robust (**Results, page 21, lines 476-483**):

*“To ensure that this ranking between test splits [[ID generalization > naturalistic OOD generalization > synthetic OOD generalization]] is meaningful and robust, we performed two controls. First, to ensure that the ranking is not contingent on a specific choice of deep neural network, we trained and tested the encoding models on image features from deep neural networks other than the vision transformer (**Supplementary Figs. 16-18**). Second, to ensure that the way in which the test splits are defined does not change our conclusions, we selected the images of the ID and naturalistic OOD test splits based, not on empirically observed fMRI responses, but rather on the features of a deep neural network (**Supplementary Fig. 19**). In both control analyses,*

we observed decreasing model generalisation performances for ID, naturalistic OOD, and synthetic OOD test splits.”

“Supplementary Fig. 19 | Generalization performance of encoding models to naturalistic in-distribution and out-of-distribution test splits chosen based on the image features of a deep neural network. We trained vision-transformer-based encoding models on NSD-core, and tested their generalization across three conditions: ID on NSD-core, naturalistic OOD on NSD-core, and synthetic OOD on NSD-synthetic. To select the images of NSD-core’s ID and naturalistic OOD test splits, we applied the MDS and clustering procedure described in the Results section of Fig. 7 to the first 250 principal components of the image features of a vision transformer. a, MDS embedding of trial-averaged fMRI responses (aggregated across participants) for the train images (gray dots), for the ID test images (orange dots), for the naturalistic OOD test images (yellow dots), and for the synthetic OOD test images (turquoise dots). b, Encoding models’ explained variance for the three test conditions, normalized by the noise ceiling. This figure shows results averaged across participants on flattened cortical surfaces. To reduce false positives, for each vertex we averaged results from only those participants with a noise ceiling greater than 0.3 for both NSD-core and NSD-synthetic (vertices for which no participant has a noise ceiling above 0.3 are not shown). c, Encoding models’ zero-shot identification scores of the recorded fMRI response stimulus images, for each of the three test conditions, on violin plots. The y-axis represents the correct image rank, that is, the position at which the correct image appears among the models’ choices (choosing among all 852 images from the three test splits), with lower ranks indicating a more robust image identification. The colored violin shapes indicate the distribution of observations, with wider sections corresponding to higher density of values. Small transparent dots indicate the correct image rank for single images and participants. Large black dots indicate the average correct image rank across all participants and images within each test condition, with the corresponding score reported above the dot. Horizontal dashed lines indicate chance identification scores.”

Regarding the potential influence of session-specific effects, this is not feasible to fully control in the current analysis, and here is why. In this analysis we use a clustering

algorithm to split the fMRI responses (in MDS space) for NSD-core’s naturalistic images into training, ID test, and naturalistic OOD test. Now, in this context, controlling for session specific effects (i.e., enforcing train/test splits from non-overlapping sessions) would only be possible if some of the responses from the test sessions were similarly distributed to the responses for the train sessions (thus defining the ID test split), whereas some other responses from the test sessions formed a separate cluster from the train session responses (thus defining the naturalistic OOD test split). We found the first condition to be true, but not the second: the responses for the test sessions were similarly distributed to the responses for the train sessions, but did not form separate clusters. (That this is the case makes sense, since NSD’s experimental design was intentionally homogeneous across sessions for all images presented.) As a representative example, we show the fMRI responses (in MDS space) for the train/test sessions for subject 1 (**Revision Fig. 1**). Therefore, by splitting the data into train/test sessions we are not able to find the naturalistic OOD test split necessary for this analysis.

Luckily though, we do not believe controlling for session-specific effects to be as relevant for this analysis, as it was for the other analyses. This because the main claim of this analysis is that OOD test splits can also be defined using naturalistic images from NSD-core, which is supported by the finding that model performance is lower for naturalistic OOD than for naturalistic ID tests. Now, given that both naturalistic OOD and ID conditions came from the same sessions as the training data conditions (since we did not split the data into non-overlapping train/test sessions), we can safely assume that our findings are not driven by session-specific effects (as it might be the case if only the naturalistic OOD or ID data came from the same session of the training data).

Revision Fig. 1 | Subject 1 trial-average fMRI responses in MDS space for images presented in the first 30 sessions (black dots), and for images falling completely within the last 10 sessions (red dots).

Next, we thank the Reviewer for the opportunity to clarify the motivation for the analysis in **Fig. 7**. In line with Shirakawa et al., 2025, we acknowledge that NSD-core has low stimulus diversity. Consequently, while NSD-core allows for naturalistic OOD

generalization tests (as we show in **Fig. 7**), these tests are not as strict as the ones performed on fMRI responses with higher degree of OOD, such as on responses for artificial stimuli from NSD-synthetic. (The fact that NSD-synthetic leads to lower generalization performances further supports that it allows for stricter generalization tests compared to NSD-core.) Thus, the combination of NSD-core and NSD-synthetic allows for a nuanced account of the OOD generalization performance of brain models across various degrees of OOD. We made several additions to the manuscript to reflect these considerations.

(Introduction, page 5, lines 44-51): “However, it remains difficult to assess the level of out-of-distribution (OOD) generalization of brain models built using recent large-scale visual neural datasets, that is, whether model predictions generalize outside of the stimulus distribution on which they are trained (Liu et al., 2021). This is because, despite the unprecedented size of these datasets, their stimuli typically live within the same visual distribution, which comprises only a fraction of the vast visual space that our brains process during our lifetime. *For example, through clustering and reconstruction analyses, Shirakawa and colleagues showed that NSD’s stimulus images have limited visuo-semantic diversity (Shirakawa et al., 2025).* As a result, brain models are typically tested in-distribution (ID), that is, within the visual distribution on which they are trained.”

(Results, page 21, lines 484-488): “In conclusion, these results showcase the strength of combining NSD-core and NSD-synthetic to test the OOD robustness of brain models on neural responses. The results demonstrate that the concept of OOD is not restricted to artificial stimuli but can be usefully applied even within the domain of naturalistic stimuli, *although resulting in less strict OOD generalization tests due to the lower degree of OOD of fMRI responses for naturalistic compared to artificial stimuli.*”

(Discussion, page 23, lines 555-566): “First, there is a continuum of degrees to which the test data can be OOD with respect to the train data, which in turn affects the generalization performance of computational models of the brain. We showed this for test fMRI responses for different NSD-synthetic image classes (**Fig. 6**), as well as for test fMRI responses for naturalistic and synthetic images (**Fig. 7**). Our empirical results reinforce the fact that OOD generalization performance of brain models cannot be fully described by a simple summary number, but lives in a continuum *of distances between the train and test data distributions*. Combining NSD-core and NSD-synthetic provides a way to test several degrees of OOD generalization across this continuum, *from lower degrees of OOD and thus less strict generalization tests using fMRI responses for NSD-core’s naturalistic stimuli, to higher degrees of OOD and thus stricter tests using NSD-synthetic*. We envision that this will endow researchers with greater flexibility in model testing, resulting in a more nuanced account of the OOD generalization performance of brain models.”

Finally, we also referenced the Shirakawa et al., 2025 paper when describing our method to define ID and OOD naturalistic test splits (**Results, page 21, line 518**):

“We first used a combination of MDS and clustering algorithms to divide NSD-core’s fMRI responses into train, ID test, and naturalistic OOD test splits (Shirakawa et al., 2025).”

****[Related to my previous comments “2. The encoding analysis in this paper...”]****

The second concern in my previous review was the potential confound of using the same runs/sessions for both training and test data. The authors addressed this with additional analyses such as Supplementary Figure 2, which control for session (and run) confounds by separating the last five sessions for testing. This addition is valuable and reassures me that the authors have considered the issue carefully.

One important point that could be emphasized more clearly, however, is that ID generalization drops substantially once session confounds are removed (with confounds: 45.14% in Fig. 4c; without confounds: 38.64% in Sup. Fig. 2a). Since OOD performance decreases only minimally (with confounds: 10.12%; without confounds: 9.94%), this suggests that session confounds strongly affect encoding performance.

It may strengthen the manuscript if the default encoding analyses (Fig. 4 ~ 7) are presented in a version that excludes session confounds, allowing for a clearer comparison focused on effects of interest. Explicitly acknowledging the impact of session confounds in the main text would also help readers better interpret the results. Highlighting this point would provide value not only for this dataset but also for future large-scale data collection paradigms and open-data splitting strategies.

We appreciate the Reviewer’s concern with session-specific effects, which led us to add the following text to the main Results section of the paper (**Results, page 13, lines 308-324**):

*“To ensure that the lower OOD performances are meaningful and robust, we performed three controls. First, to ensure that the lower performance on NSD-synthetic is not simply due to the fact that it was collected in a separate scan session, we trained and tested the encoding models using fMRI responses from separate NSD-core scan sessions (**Supplementary Fig. 2**). [...] In all three **control analyses**, we observed lower OOD performances for NSD-synthetic than for NSD-core. **Importantly, controlling for session-specific effects led to a considerable decrease in ID (Δ mean noise-ceiling-normalized explained variance score of 6.5%), but not OOD (Δ mean noise-ceiling-normalized explained variance score of 0.18%) generalization. Therefore, for a clearer comparison between ID and OOD generalization performances using NSD-synthetic, we recommend researchers to train and test their models on data from independent scan sessions.**”*

****[Related to my previous comments “3. In the encoding analysis,...”]****

The third concern in my previous review was that including PCA might emphasize ID-relevant information while discarding OOD-relevant features. The new analyses in Supplementary Figures 3 and 4, comparing linear regression with PCA against ridge regression without PCA across layers, are a meaningful revision.

While this is a valuable addition and certainly reduces my concern, I believe the results reveal something more important than is currently mentioned in the main text (L285). Specifically, in many layers, OOD prediction performance under ridge regression (without PCA) exceeds that under both linear regression (with PCA) and the main results (Fig. 4), a pattern not observed for ID performance. This implies that applying PCA to training features may not be the optimal approach for OOD modeling, as I pointed out in the previous review.

Explicitly acknowledging this point in the main text could further strengthen the paper, especially given its central focus on this paper is out-of-distribution modeling

We agree with the Reviewer that finding the optimal way of mapping stimulus features onto brain responses in order to maximize the robustness and generalizability of encoding models is a crucial problem for the field of computational visual neuroscience.

Our results in **Supplementary Figs. 3-4** show that, for 3 out of 4 AlexNet layers, ridge regression on the full layer space leads to higher OOD prediction accuracies than linear regression on the PCA-downsampled layer space. However, we do not believe that this is enough evidence to claim that applying PCA to training features may not be the optimal approach for OOD modeling, for two reasons. First, the highest OOD prediction accuracy actually comes from encoding models based on linear regression with PCA trained on AlexNet layer 7. Second, establishing which regression strategy leads to highest encoding accuracy would require rigorously training/testing the linear regressions using different amounts of principal components, and, more generally, a more comprehensive treatment of different types of data scenarios (which is obviously out of scope of the present paper).

Based on these considerations, we propose that the optimal regression strategy might be contingent on the data used (**Results, page 13, lines 308-326**):

“To ensure that the lower OOD performances are meaningful and robust, we performed three controls. [...] Second, to ensure that the specific regression method that one chooses to use does not change our conclusions, we trained and tested the encoding models using individual AlexNet layers, by either mapping each layer’s first 250 PCs onto fMRI responses with linear regression, or by mapping the layer’s full feature space using ridge regression (Supplementary Figs. 3-4). [...] In all three control analyses, we observed lower OOD performances for NSD-synthetic than for NSD-core. [...] Furthermore, neither of the two regression methods led to best OOD generalization performances for all AlexNet layers, suggesting that the optimal regression strategy might be contingent on the specific data scenario.”

****[Related to my previous comments “4. A further important concern...”]****

The fourth concern in my previous review was that that OOD performance was evaluated only with encoding analyses. The authors addressed this with image-identification analyses, which is a valuable addition. However, interpretation remains somewhat complicated by (i) class imbalance candidate set in Fig. 6d (e.g., 112 spiral gratings vs. 68 chromatic noise vs.

40 word stimuli vs. 8–16 stimuli in other classes), and (ii) different candidate sets across conditions in Fig. 7c.

It may be helpful to consider using balanced candidate sets (e.g., N images per class) to eliminate the possibility that imbalance drives differences in Fig. 6d. Similarly, for Fig. 7c, using a common candidate set across conditions (e.g., pooling all test images, or drawing balanced samples from ID and OOD splits, or randomly selecting N training images) could make the interpretation clearer. Even if these stricter analyses do not substantially change the results, their inclusion would demonstrate rigor and increase the impact and value of this paper.

We thank the Reviewer for these helpful comments, which led to two edits to the zero-shot identification analysis.

(i) To mitigate the risk of the identification results reflecting imbalances between the number of images in each NSD-synthetic image class, we re-ran the zero-shot identification analysis of **Fig. 6d** using the candidate image sets of 8 randomly selected images per NSD-synthetic image class (i.e., since every class has at least 8 images), for a total of 64 images. We iterated the identification algorithm 1,000 times, while each time selecting a different random sample of 8 candidate images per image class, and then averaged the results across the 1,000 iterations. Despite this algorithmic change, we obtained the same ranking of identification performance between the NSD-synthetic image classes as previously (**Fig. 6d**), and still found a significant correlation between the ranks of the correct stimulus image and the distributional distance scores (**Fig. 6e**). We added these new results to the main manuscript, as well as to the supplementary information.

“Fig. 6 | The degree of out-of-distribution is informative of brain model failures. [...] d, Encoding models’ zero-shot identification scores of the recorded fMRI response stimulus images, for each NSD-synthetic image class, on violin plots. The y-axis represents the correct image rank, that is, the position at which the correct image appears among the models’ choices (choosing among 8 randomly selected images per NSD-synthetic image class, for a total of 64 images; the final identification scores reflect the average across 1,000 analysis iterations, each with a different random selection of 8 images per image class) with lower ranks indicating a more robust image identification. The colored violin shapes indicate the distribution of observations, with wider sections corresponding to higher density of values. Small transparent dots indicate the correct image rank for

single images and participants. Large black dots indicate the average correct image rank across all participants and images within each NSD-synthetic image class, with the corresponding score reported above the dot. Horizontal dashed lines indicate chance identification scores. e, Scatterplot indicating the relationship between the degree of OOD of fMRI responses and the zero-shot image identification scores, across NSD-synthetic image classes. The x-axis represents the Euclidean distance in MDS embedding space between fMRI responses for each NSD-synthetic image class and NSD-core (i.e., the distances in panel a). The y-axis represents the average correct image rank across all participants and images within each NSD-synthetic image class (i.e., the average scores in panel d).”

(ii) We re-ran the zero-shot identification analysis of **Fig. 7c** using the same candidate image set for each of the three test splits (i.e., all 852 images from the three test splits), and obtained quantitatively similar results: best identification scores for ID testing on NSD-core, followed by naturalistic OOD testing on NSD-core, and finally by synthetic OOD testing on NSD-synthetic. We added these new results to the main manuscript, as well as to the supplementary information.

“Fig. 7 | NSD enables out-of-distribution tests of brain models across both naturalistic and synthetic stimulus images. [...] Encoding models’ zero-shot identification scores of the recorded fMRI response stimulus images, for each of the three test conditions, on violin plots. The y-axis represents the correct image rank, that is, the position at which the correct image appears among the models’ choices (choosing among all 852 images from the three test splits), with lower ranks indicating a more robust image identification. The colored violin shapes indicate the distribution of observations, with wider sections corresponding to higher density of values. Small transparent dots indicate the correct image rank for single images and participants. Large black dots indicate the average correct image rank across all participants and images within each test condition, with the corresponding score reported above the dot. Horizontal dashed lines indicate chance identification scores.”

[Related to my previous comments “5. The current OOD set seems...”]

The fifth point was in my previous review to see the performance differences across each OOD category. In the revised manuscript, Figure 6 presents these results, and the Discussion briefly notes them (L508–L513). This is an important revision, though I believe it could be highlighted more concretely. For example, although aggregate NSD-synthetic

performance (Supplementary Fig. 5 / Fig. 4) appears low, class-level performance varies substantially (40.18% for manipulated scenes vs. 13.37% for spiral gratings).

It would also be helpful to explain why the noise class shows relatively high encoding performance (29.25%), even exceeding that of some OOD classes (e.g., chromatic noise, spiral gratings). An additional exploration of which aspects of stimulus-related information are being actually captured by the encoding model could provide useful insight. In this context, rather than focusing solely on encoding performance, a more multifaceted analysis (e.g., preparing appropriate candidate sets for the image-identification analysis) could offer a promising direction.

We appreciate these ideas, which we point towards future research directions. Following on the Reviewer's suggestions, we now elaborate on the OOD generalization performance for the individual NSD-synthetic image classes in the Discussion section (**Discussion, page 22, lines 582-596**):

“Beyond indicating that further model improvement is required to accurately describe visual representations in the brain, NSD-synthetic also suggests directions for closing the gap between ID and OOD generalization performance. One direction of model improvement comes from consideration of the properties of NSD-synthetic’s stimuli. Since the stimuli consist of a multitude of diverse and parameterized images, they facilitate discovery of the specific visual features to which computational models fail to generalize. Notably, we found that the generalization performance for the different NSD-synthetic image classes varied considerably, from performances on par to the ones obtained ID (e.g., mean noise-neiling-normalized encoding accuracy of 40.18% for manipulated scenes), to substantially lower performances (e.g., 13.37% for spiral gratings). Future research efforts could explore why certain visual features are modeled better than others, for example, why fMRI responses for the noise image class are better modeled than responses for the chromatic noise class. We additionally found that the generalization performances of the NSD-synthetic image classes negatively correlated with their degree of OOD. In light of these observations, we suggest that future modeling efforts may benefit from focusing on improving model robustness for stimulus properties with a high degree of OOD (Kay, 2018; Madan et al., 2024).”

Minor concerns:

- In the updated Abstract, the authors write, “(L16) However, these datasets [NSD] lack out-of-distribution (OOD) components ... Here, we address this limitation by releasing NSD-synthetic ...,” yet later they state, “(L27) the concept of OOD is not restricted to artificial stimuli but can be usefully applied even within the domain of naturalistic stimuli.” These statements may appear contradictory; a brief clarification would help.

We edited the Abstract based on the considerations made above in response to Reviewer #1 comment 1 (i.e., where we clarified the motivation for the analysis in **Fig. 7**) (**Abstract, page 1, lines 17-36**):

“Large-scale *visual neural* datasets such as the Natural Scenes Dataset (NSD) are boosting computational neuroscience research by enabling models of the brain with performances beyond what was possible just a decade ago. *However, because the stimuli of these datasets typically live within a common naturalistic visual distribution, they do not allow for strict out-of-distribution (OOD) generalization tests* which are crucial for the development of more robust models. Here, we address this limitation by releasing NSD-synthetic, a dataset consisting of 7T fMRI responses from the same eight NSD participants for 284 synthetic images. We show that NSD-synthetic’s fMRI responses reliably encode stimulus-related information and are OOD with respect to NSD. Furthermore, we provide a proof of principle that OOD generalization tests on NSD-synthetic reveal differences between models of the brain that are not detected with the original NSD data; we demonstrate that the degree of OOD (quantified as the distance between a set of responses and the training data used for modeling) is predictive of the magnitude of model failures; and we show that *less strict OOD generalization tests can be usefully applied even within the domain of naturalistic stimuli*. These results showcase how NSD-synthetic enables OOD generalization tests that facilitate the development of more robust models of visual processing and the formulation of more accurate theories of human vision.”

- At L389, the authors state that the degree of OOD is measured by Euclidean distance between training and test fMRI responses, but in practice it is computed in MDS space (L392–396). Modifying this inconsistency would improve precision.

We thank the Reviewer for pointing out this imprecision, which we fixed (**Results, page 18, lines 444-446**):

“We operationally define the degree of OOD as the Euclidean distance between the fMRI response to a given stimulus of interest and the NSD-core train responses *in two-dimensional MDS space*.”

- At L545, the phrase “across this continuum” is unclear. The results suggest discrete stages of OOD generalization rather than a continuous spectrum; clarification would be helpful.

With “continuum” we refer to the continuous space defined by the distance between the test and train data distributions. The results highlight discrete stages of this continuum because there we aggregated the test images in the 8 NSD-synthetic image classes (or in naturalistic/synthetic test splits). However, each image of these image classes (or test splits) lives in a continuous space of distance from the train data distribution. We clarified this in the Discussion (**Discussion, page 23, lines 625-636**):

“First, there is a continuum of degrees to which the test data can be OOD with respect to the train data, which in turn affects the generalization performance of computational models of the brain. We showed this for test fMRI responses for different NSD-synthetic image classes (**Fig. 6**), as well as for test fMRI responses for naturalistic and synthetic images (**Fig. 7**). Our empirical results reinforce the fact that OOD generalization performance of brain models cannot be fully described by a simple summary number, but lives in a continuum *of distances between the train and test data distributions*. Combining NSD-core and NSD-synthetic provides a way to test

several degrees of OOD generalization across this continuum, from lower degrees of OOD and thus less strict generalization tests using fMRI responses for NSD-core's naturalistic stimuli, to higher degrees of OOD and thus stricter tests using NSD-synthetic. We envision that this will endow researchers with greater flexibility in model testing, resulting in a more nuanced account of the OOD generalization performance of brain models."

- At L292, "Fig. 7" should be corrected to "Fig. 5."

We thank the Reviewer for pointing out this typo, which we fixed.

- At L535, the phrase "'~ across the visual areas".' might be better written as "'~ across the visual areas."

As suggested, we moved the full stop inside the quotes.

Overall, this revised version is a substantial improvement over the previous manuscript. I appreciate the authors' efforts in addressing the concerns raised in the first round of review and for performing extensive additional analyses. With the further clarifications and refinements suggested above, I believe this work will provide a particularly valuable contribution to the community and will have significant impact in advancing research on large-scale visual neural datasets.

We thank the Reviewer for the positive evaluation of the revised manuscript, as well as for the new comments that we believe improved the clarity and strength of our findings.

[Reference]

Mitchell, T. M., Shinkareva, S. V., Carlson, A., Chang, K.-M., Malave, V. L., Mason, R. A., & Just, M. A. (2008). Predicting Human Brain Activity Associated with the Meanings of Nouns. *Science*, *320*(5880), 1191–1195. <https://doi.org/10.1126/science.1152876>

Brouwer, G. J., & Heeger, D. J. (2009). Decoding and Reconstructing Color from Responses in Human Visual Cortex. *Journal of Neuroscience*, *29*(44), 13992–14003. <https://doi.org/10.1523/JNEUROSCI.3577-09.2009>

Shirakawa, K., Nagano, Y., Tanaka, M., Aoki, S. C., Muraki, Y., Majima, K., & Kamitani, Y. (2025). Spurious reconstruction from brain activity. *Neural Networks*, *190*, 107515. <https://doi.org/10.1016/j.neunet.2025.107515>

Madan, S., Xiao, W., & Cao, M. (2024). Benchmarking Out-of-Distribution Generalization Capabilities of DNN-based Encoding Models for the Ventral Visual Cortex. *NeurIPS*, *2832*, 89249–89277. <https://dl.acm.org/doi/10.5555/3737916.3740748>

We thank the Reviewer for these references.

Reviewer #2

Remarks to the Author

We thank Reviewer #2 for co-reviewing the manuscripts and for the helpful comments.

Remarks on code availability

I have just skimmed but not attempted to run. It appears to have well-documented and readable code.

Those well-structured codes will be useful for reproducing the results.

We thank the Reviewer for checking the code.

Reviewer #3

Remarks to the Author

The authors have amply addressed my already minor comments. They have also gone above and beyond in their responses to other reviewers' comments, substantially extending the analyses in the paper. It's good to see that the new analyses and framing treat being "out of distribution" as a continuum, and as something separable from whether an image is natural/artificial. I continue to find the paper a valuable contribution to datasets and model evaluations in visual neuroscience.

We thank Reviewer #3 for reading the revised manuscript and for the positive evaluation, as well as for the comments of the first review round that resulted in valuable edits and additions to the manuscript.

Remarks on code availability

I haven't tried to run the code, but have read the ReadMe and skimmed through the code files. They appear well-organised and well-documented, and to provide everything needed to reproduce analyses underlying each figure in the paper.

We thank the Reviewer for checking the code.

Reviewer #4

Remarks to the Author

The authors have replied extensively to mine, but also the concerns of the other reviewers. I in particular appreciate the plimitation expressed in Discussion, page 25, lines 628-635 and the clear ideas expressed in lines 514-528.

I have no further comments.

We thank Reviewer #4 for reading the revised manuscript and for the positive evaluation, as well as for the comments of the first review round that resulted in valuable additional analyses and discussions.

Remarks on code availability

The code seems to be complete and also updated to the last version. I have sampled some files. Looks good.

We thank the Reviewer for checking the code.

Dear Reviewers,

Please find the detailed point-by-point response below. Reviewer comments are highlighted in gray, and the corresponding responses indented. Within the responses, extracts from the manuscript are “*quoted in italics*”, and extracts reflecting changes to the manuscript are “*quoted in italics and red font*”. Manuscript line/page/figure numbers are indicated in **bold font**.

Reviewer #1

Remarks to the Author

The authors have responded sincerely to the previous comments, and through clarifications additional analyses, the manuscript has surely improved. I have only one remaining concern regarding the expression “less strict OOD generalization tests” used to describe the naturalistic OOD split. As currently written, this phrasing may unintentionally undervalue the potential of NSD-core.

We thank Reviewer #1 for the positive evaluation, and for the dedication in reviewing the manuscript which resulted in helpful revisions that clarified the manuscript text.

I understood that preparing OOD test splits that additionally control for session-specific effects is challenging in the present analysis. Considering that the main purpose of this study is a proof-of-principle demonstration using NSD-synthetic, I will not request further additional analyses. However, I would encourage the authors to refine the manuscript by more clearly acknowledging the limitations of the current naturalistic OOD split and by avoiding statements that might be interpreted as fundamental limitations of NSD-core itself.

In particular, referring to the NSD-core-based OOD evaluation as a “less strict OOD generalization test” could be read as implying that NSD-core inherently permits only weak OOD tests. In practice, with more carefully designed train-test splits, stricter naturalistic OOD evaluations are possible even within NSD-core. For example, following the conventional stimulus-based OOD splitting approaches (Mitchell et al., 2008; Brouwer & Heeger, 2009; Madan et al., 2024; Shirakawa et al., 2025), we could construct a naturalistic OOD test set consisting of a specific semantic category (e.g., “zebra” images) presented only in the last few sessions. The training set could then be built from earlier sessions while excluding all zebra-containing stimuli. This would allow direct comparison between naturalistic OOD tests (inherent to NSD-core) and synthetic OOD tests (inherent to NSD-synthetic) while simultaneously removing session confounds existing in the current naturalistic OOD.

A similar logic can also be applied to the fMRI-based splits. For instance, given the distribution shown in Revision Fig. 1, we could design the red samples at the right side as a test set and construct the training set from the remaining black points while excluding the corresponding region of the MDS space.

I am not requesting that the authors implement such additional analyses. Rather, I suggest that the manuscript be revised to (i) clarify that the “less strict” nature of the current NSD-core OOD tests reflects the specific data split used in this study, not a fundamental limitation of the dataset itself, and (ii) acknowledge more explicitly that stricter OOD tests within NSD-core are, in principle, possible with suitably designed splits.

We addressed the Reviewer’s concern by (i) removing the statements that NSD-core allows for less strict OOD generalization tests compared to NSD-synthetic, and by (ii) acknowledging that more constrained train-test splits on NSD-core might allow for stricter OOD generalization tests.

Specifically, the following revised sections may merit careful reconsideration:

L20-L22:

“because the stimuli of these datasets typically live within a common naturalistic visual distribution, they do not allow for strict OOD generalization tests”

→ Suggest clarifying that strict tests are possible but require more deliberate split design.

We edited this segment (**Abstract, page 1, lines 15-18**):

*“because the stimuli of these datasets typically live within a common naturalistic visual distribution, **they make it challenging to implement** out-of-distribution (OOD) generalization tests crucial for the development of more robust brain models.”*

L31-33

“less strict OOD generalization tests can be usefully applied even within the domain of naturalistic stimuli.”

We removed this segment since by Editorial request we had to reduce the length of the abstract.

L555-556

“although resulting in less strict OOD generalization tests due to the lower degree of OOD of fMRI responses for naturalistic compared to artificial stimuli.”

We removed the statement of NSD-core being a less strict testbed of OOD tests, and made a minor edit to the preceding sentence (**Results, page 14, lines 369-371**):

*“The results also demonstrate that the concept of OOD is not restricted to artificial stimuli, but **that through carefully-designed train-test splits** it can be usefully applied even within the domain of naturalistic stimuli.”*

L633- L635

“from lower degrees of OOD and thus less strict generalization tests using fMRI responses for NSD-core’s naturalistic stimuli, to higher degrees of OOD and thus stricter tests using NSD-synthetic”

We made an addition to this segment (**Discussion, page 16, lines 438-446**):

“Our empirical results reinforce the fact that OOD generalization performance of brain models cannot be fully described by a simple summary number, but lives in a continuum of distances between the train and test data distributions. Combining NSD-core and NSD-synthetic provides a way to test several degrees of OOD generalization across this continuum, from lower degrees of OOD and thus less strict generalization tests using fMRI responses for NSD-core’s naturalistic stimuli, to higher degrees of OOD and thus stricter tests using NSD-synthetic. (However, stricter OOD generalization tests are in principle also possible with NSD-core’s naturalistic stimuli, through a more constrained division of train-test splits than the one employed in this work.)”

This revisions have considerably strengthened the manuscript. With the minor clarifications suggested above, I am confident that the manuscript will be even clearer and more informative, and that it will serve as a valuable resource for future studies on large-scale visual neural datasets.

We thank the Reviewer for the positive evaluation of the revised manuscript, as well as for the new comments, which helped improve the clarity of the manuscript.

Reviewer #2

Remarks to the Author

We thank Reviewer #2 for co-reviewing the manuscripts and for the helpful comments.